# A Unified Comparative Study with Generalized Conformity Scores for Multi-Output Conformal Regression

**Victor Dheur** [1]  **Matteo Fontana** [2]  **Yorick Estievenart** [1]  **Naomi Desobry** [1]  **Souhaib Ben Taieb** [1 3]

## Abstract

Conformal prediction provides a powerful framework for constructing distribution-free prediction regions with finite-sample coverage guarantees. While extensively studied in univariate settings, its extension to multi-output problems presents additional challenges, including complex output dependencies and high computational costs, and remains relatively underexplored. In this work, we present a unified comparative study of nine conformal methods with different multivariate base models for constructing multivariate prediction regions within the same framework. This study highlights their key properties while also exploring the connections between them. Additionally, we introduce two novel classes of conformity scores for multi-output regression that generalize their univariate counterparts. These scores ensure asymptotic conditional coverage while maintaining exact finite-sample marginal coverage. One class is compatible with any generative model, offering broad applicability, while the other is computationally efficient, leveraging the properties of invertible generative models. Finally, we conduct a comprehensive empirical evaluation across 13 tabular datasets, comparing all the multi-output conformal methods explored in this work. To ensure a fair and consistent comparison, all methods are implemented within a unified code base[1].

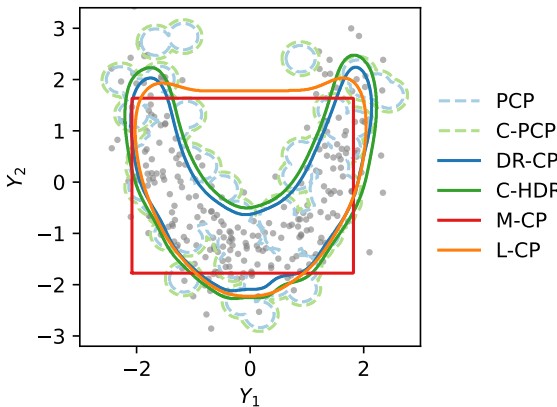

Figure 1: Examples of bivariate prediction regions with an 80% coverage level for a toy example.

[1]Department of Computer Science, University of Mons, Mons, Belgium [2]Department of Computer Science, Royal Holloway, University of London, Egham, United Kingdom [3]Department of Statistics and Data Science, Mohamed bin Zayed University of Artificial Intelligence, Abu Dhabi, United Arab Emirates. Correspondence to: Victor Dheur <victor.dheur@umons.ac.be>.

*Proceedings of the $42^{nd}$ International Conference on Machine Learning*, Vancouver, Canada. PMLR 267, 2025. Copyright 2025 by the author(s).

[1]`https://github.com/Vekteur/multi-output-conformal-regression`

## 1. Introduction

Quantifying uncertainty in model predictions is crucial in many real-world applications, often involving prediction problems with multiple output variables and complex statistical dependencies. For example, in medical diagnostics, the progression of a disease can be studied by analysing multiple health indicators that exhibit nonlinear dependencies, such as blood pressure and cholesterol levels of a patient (Rajkomar et al., 2018). Although modern probabilistic AI models can model complex relationships between variables, they may produce unreliable or overly confident predictions (Nalisnick et al., 2018).

Conformal prediction (CP) offers a robust framework for improving model reliability by generating distribution-free prediction regions with a finite-sample coverage guarantee (Vovk et al., 1999). Although substantial research has focused on univariate prediction problems (Romano et al., 2019; Sesia and Romano, 2021; Rossellini et al., 2024), multivariate settings have received less attention. Among existing work, Zhou et al. (2024) achieves marginal coverage by combining univariate prediction regions, but fails to capture dependencies between variables. Other methods, such as density-based approaches (Izbicki et al., 2022) or sample-based techniques (Wang et al., 2023b; Plassier et al., 2025), suffer from high computational costs. An alterna-

tive method (Sadinle et al., 2019) optimises the size of the region, but does not achieve asymptotic conditional coverage. For a toy bivariate example, Figure 1 illustrates the diversity of prediction regions obtained using a selection of conformal methods considered in this paper.

Our *first contribution* is a unified comparative study of nine conformal methods with different multivariate base models for constructing multivariate prediction regions within the same framework. This study highlights their key properties while also exploring the connections between them. We examine different conformity scores with different multivariate base predictors, discussing prediction regions derived from the marginal distributions of individual output variables, their joint PDF, or sampling procedures (e.g., generative models).

Our *second contribution* introduces two novel classes of conformity scores for multi-output regression that generalize their univariate counterparts. These scores ensure asymptotic conditional coverage while maintaining exact finite-sample marginal coverage.

The first, CDF-based scores, leverage the cumulative distribution function (CDF) of any conformity score to achieve asymptotic conditional coverage. This approach generalizes the univariate HPD-split score, based on univariate highest-density region from Izbicki et al., 2020, to multivariate prediction regions derived from any conformity score. Additionally, we propose a specific instance of CDF-based scores that builds on PCP from Wang et al., 2023b. This method avoids the estimation of a predictive density, instead relying solely on samples from any generative model.

The second, latent-based scores is inspired by Feldman et al., 2023 and can be interpreted as an extension of distributional conformal prediction (Chernozhukov et al., 2021) to multivariate outputs. Compared to Feldman et al., 2023, it does not require directional quantile regression, and the conformalization is performed directly in the latent space, eliminating the need to construct a grid. This enhances both computational efficiency and scalability.

Finally, as our *third contribution*, we conduct a large-scale empirical study comparing the different multi-output conformal methods across 13 tabular datasets with multivariate outputs, evaluating several performance metrics. We consider a variety of multivariate regression models, namely Multivariate Quantile Function Forecaster (Kan et al., 2022), Distributional Random Forests (Cevid et al., 2022), and a multivariate Gaussian Mixture Model parameterized by a hypernetwork (Ha et al., 2022; Bishop, 1994).

## 2. Background

Consider a multivariate regression problem where the objective is to predict a $d$-dimensional response vector $y \in \mathcal{Y} = \mathbb{R}^d$ based on a feature vector $x \in \mathcal{X} \subseteq \mathbb{R}^p$. We assume there exists a true joint distribution $F_{XY}$ over $\mathcal{X} \times \mathcal{Y}$, and we have access to a dataset $\mathcal{D} = \{(X^{(j)}, Y^{(j)})\}_{j=1}^n$ where $(X^{(j)}, Y^{(j)}) \overset{\text{i.i.d.}}{\sim} F_{XY}$. Given a feature vector $x$, we denote the conditional distribution of $Y$ given $X = x$ as $F_{Y|X=x}$, and the associated probability density function (PDF) as $f_{Y|X=x}$.

Using the dataset $\mathcal{D}$, for any $x \in \mathcal{X}$, CP allows us to transform base predictors, denoted $\hat{h}$, into calibrated, distribution-free prediction regions $\hat{R}(x) \subseteq \mathcal{Y}$ for the true output $y$ with finite-sample coverage guarantees.

### 2.1. Split-conformal prediction

Split-conformal prediction (SCP, Papadopoulos et al., 2002) is a computationally efficient variant of conformal prediction that divides the dataset $\mathcal{D}$ into two disjoint subsets: a training set $\mathcal{D}_{\text{train}}$ and a calibration set $\mathcal{D}_{\text{cal}}$. A model is first trained on $\mathcal{D}_{\text{train}}$ to obtain a base predictor $\hat{h}$. Based on $\hat{h}$, a conformity score (function) $s : \mathcal{X} \times \mathcal{Y} \to \mathbb{R}$ is defined, where lower scores indicate a better fit between the feature vector $x$ and the response $y$. The calibration scores $\mathcal{S} = \{s(x, y)\}_{(x,y) \in \mathcal{D}_{\text{cal}}} \cup \{+\infty\}$ are then computed, from which the $(1 - \alpha)$ empirical quantile is calculated as:

$$\hat{q} = \text{Quantile}\left(\mathcal{S}; \frac{k_\alpha}{|\mathcal{D}_{\text{cal}}| + 1}\right), \quad (1)$$

where $k_\alpha = \lceil (|\mathcal{D}_{\text{cal}}| + 1)(1 - \alpha) \rceil$. This quantile serves as the threshold for constructing prediction regions. For an input $x$, the (random) prediction region is given by:

$$\hat{R}(x) = \{y \in \mathcal{Y} : s(x, y) \leq \hat{q}\}. \quad (2)$$

If the random pair $(X, Y)$ is exchangeable with $\mathcal{D}_{\text{cal}}$, SCP guarantees marginal coverage:

$$\mathbb{P}_{X,Y,\mathcal{D}_{\text{cal}}}(Y \in \hat{R}(X)) = \mathbb{P}(s(X, Y) \leq \hat{q}) \geq 1 - \alpha, \quad (3)$$

where the probability is taken over $(X, Y)$ and $\mathcal{D}_{\text{cal}}$. Assuming no ties in scores, the marginal coverage is exactly $\frac{k_\alpha}{|\mathcal{D}_{\text{cal}}|+1}$, yielding $\mathbb{P}(Y \in \hat{R}(X)) \leq 1 - \alpha + \frac{1}{|\mathcal{D}_{\text{cal}}|+1}$.

Ideally, the prediction region should achieve *conditional coverage* at the level $1 - \alpha$, i.e.:

$$\mathbb{P}(Y \in \hat{R}(X) \mid X) \geq 1 - \alpha. \quad (4)$$

holds almost surely. This is a stronger requirement than marginal coverage in (3). However, as Barber et al. (2019) demonstrate, achieving conditional coverage is generally impossible without making additional assumptions about the underlying data-generating process.

## 2.2. Multi-output conformal methods

Many conformal prediction methods have been proposed in the literature and implemented within the SCP framework for various base predictors and conformity scores, with a specific focus on univariate prediction problems. In this section, we survey several conformal methods for constructing multivariate prediction regions, using different multivariate base predictors and corresponding conformity scores. Specifically, we discuss density-based, and sample-based methods, which are based on their joint PDF, or a sampling procedure (e.g., a generative model), respectively. In the following, we describe the conformity scores $s$ for different methods. The methods `M-CP` and `CopulaCPTS`, which produce hyperrectangular regions, are detailed in Appendix B. Once a conformity score is defined, the corresponding prediction region $\hat{R}$ can be computed using (2). We detail this relationship for each method in Appendix C. Furthermore, in Section 5, we analyze the properties and relationships between these methods and provide illustrative examples of the resulting prediction regions.

**DR-CP.** Given a predictive density $\hat{f}_{Y|X=x}$, a natural conformity score is the negative density:

$$s_{\text{DR-CP}}(x, y) = -\hat{f}(y \mid x). \tag{5}$$

The corresponding prediction region is a density superlevel set, $\hat{R}_{\text{DR-CP}}(x) = \{y \in \mathcal{Y} : \hat{f}(y \mid x) \geq -\hat{q}\}$. Sadinle et al. (2019) use this conformity score in the context of classification.

**C-HDR.** Izbicki et al., 2022 proposed the HPD-split method, which defines a conformity score based on the Highest Predictive Density (HPD):

$$\text{HPD}_{\hat{f}}(y \mid x) = \int_{\{y' \mid \hat{f}(y' \mid x) \geq \hat{f}(y \mid x)\}} \hat{f}(y' \mid x) \, dy' \tag{6}$$

$$= \mathbb{P}\left(\hat{f}(\hat{Y} \mid x) \geq \hat{f}(y \mid x) \mid X = x\right), \tag{7}$$

where $\hat{Y} \sim \hat{f}_{Y|X=x}$. The corresponding prediction region is a highest density region (HDR, Hyndman, 1996) with respect to $\hat{f}$ at level $\hat{q}$:

$$\hat{R}_{\text{C-HDR}}(x) = \{y \in \mathcal{Y} : \hat{f}(y \mid x) \geq t_{\hat{q}}\}, \tag{8}$$

$$\text{where } t_{\hat{q}} = \sup\{t : \mathbb{P}(\hat{f}(\hat{Y} \mid x) \geq t \mid X = x) \geq \hat{q}\}.$$

Compared to `DR-CP`, where the threshold $-\hat{q}$ is independent of $x$, `C-HDR` allows the threshold $t_{\hat{q}}$ to vary with $x$. To compute the HPD in (6), Izbicki et al., 2022 use numerical integration, whereas in our experiments, we approximate (7) using Monte Carlo sampling, as described in (13).

In the context of classification, Adaptive Prediction Sets (Romano et al., 2020) follows a similar principle by constructing a "highest mass region", which corresponds to a

superlevel set of the probability mass function with probability content at least $\hat{q}$.

**PCP.** Let $\tilde{Y}^{(1)}, \tilde{Y}^{(2)}, \ldots, \tilde{Y}^{(L)}$ denote a sample with $L$ points from the (estimated) conditional distribution $\hat{F}_{Y|X=x}$. Probabilistic Conformal Prediction (PCP, Wang et al., 2023b) defines a conformity score as the closest distance to $y$:

$$s_{\text{PCP}}(x, y) = \min_{l \in [L]} \|y - \tilde{Y}^{(l)}\|_2, \tag{9}$$

$$\text{where } \tilde{Y}^{(l)} \sim \hat{F}_{Y|X=x}, \quad l \in [L]. \tag{10}$$

The corresponding region is a union of $L$ balls centered at each sampled point $\tilde{Y}^{(l)}$, i.e. $\hat{R}_{\text{PCP}}(x) = \bigcup_{l \in [L]}\{y \in \mathcal{Y} : \|y - \tilde{Y}^{(l)}\|_2 \leq \hat{q}\}$.

**HD-PCP.** When a predictive density is available alongside a sample of $L$ points, Wang et al., 2023b proposed an extension to `PCP`, called `HD-PCP`. This method uses the same conformity score as in (9), but only retains the $\lfloor (1 - \alpha)L \rfloor$ samples with the highest density, ensuring that the prediction region is concentrated on high-density points.

**ST-DQR.** Motivated by the limitation that existing multivariate quantile regression methods do not allow the construction of regions with arbitrary shapes, Feldman et al., 2023 proposed to construct convex regions in a latent space $\mathcal{Z}$ using directional quantile regression (Paindaveine and Šiman, 2011). These regions are then mapped to the output space $\mathcal{Y}$ using a conditional variational autoencoder (CVAE), allowing a non-linear mapping between the two spaces. Specifically, they apply a conformalization step by creating a grid of points within the region in $\mathcal{Z}$, map the points to the output space $\mathcal{Y}$, and construct $d$-balls around the mapped samples, similarly to `PCP`.

## 3. Generalized Conformity Scores for Multi-Output Regression

In this section, we introduce two new classes of conformity scores: *CDF-based* and *latent-based* scores. These scores generalize existing conformity scores for univariate regression to accommodate any conformity score for multivariate outputs. The former generalizes HPD-split (Izbicki et al., 2020) to any conformity score, allowing to apply this method to multivariate outputs. We further propose a specific instance that builds on PCP (Wang et al., 2023b). The latter is inspired by Feldman et al., 2023 and can be interpreted as an extension of distributional conformal prediction (Chernozhukov et al., 2021) for multivariate outputs. Section 5 will present a comparative study of the conformity scores introduced in Section 2.2 alongside those introduced in this section.

## 3.1. CDF-based conformity scores

Consider a conformity score $s_W$, and define the random variable $W = s_W(X, Y)$ for a random pair $(X, Y)$. For an observation $(x, y)$, we introduce a new conformity score based on the conditional CDF of $W$ given $X = x$, evaluated at $s_W(x, y)$. Specifically, the score is given by

$$s_{\text{CDF}}(x, y) = \mathbb{P}(s_W(X, Y) \leq s_W(x, y) \mid X = x) \quad (11)$$
$$= F_{W|X=x}(s_W(x, y)). \quad (12)$$

This new conformity score measures the rank of $s_W(x, y)$ relative to the conditional distribution of $W$ given $X = x$.

This method applies to any conformity score $s_W$ and generalizes the (oracle) HPD-split introduced in Izbicki et al., 2020 in the context of univariate regression. Specifically, when $s_W(x, y) = s_{\text{DR-CP}}(x, y)$ is used in (12), we recover the C-HDR method. Additionally, by the probability integral transform, $s_{\text{CDF}}(X, Y) \mid X = x \sim \mathcal{U}(0, 1)$ for $x \in \mathcal{X}$, meaning that the conformity score's distribution is independent of $x$. This property ensures that conditional coverage is achieved as $|\mathcal{D}_{\text{cal}}| \to \infty$ (see Appendix E.2, Lemma 2). A similar observation was made by Izbicki et al., 2020 for C-HDR.

However, in practice, since the distribution of $Y \mid X = x$ is unknown, we approximate $s_{\text{CDF}}$ using Monte Carlo sampling:

$$s_{\text{ECDF}}(x, y) = \frac{1}{K} \sum_{k \in [K]} \mathbb{I}\left( s_W(x, \hat{Y}^{(k)}) \leq s_W(x, y) \right),$$
$$\text{where } \hat{Y}^{(k)} \sim \hat{F}_{Y|X=x}, \ k \in [K]. \quad (13)$$

Dheur et al., 2024 considered a particular case of this empirical CDF-based approach with the $s_{\text{DR-CP}}$ score for a bivariate prediction problem involving temporal point processes, where the HDR is estimated via Monte Carlo sampling.

**C-PCP.** We introduce a special case of our new score, called C-PCP (CDF-based Probabilistic Conformal Prediction), by setting $s_W(x, y) = s_{\text{PCP}}(x, y)$ in (13), which gives:

$$s_{\text{C-PCP}}(x, y) =$$
$$\frac{1}{K} \sum_{k \in [K]} \mathbb{I}\left( \min_{l \in [L]} \|\hat{Y}^{(k)} - \tilde{Y}^{(l)}\| \leq \min_{l \in [L]} \|y - \tilde{Y}^{(l)}\| \right).$$

Compared to the methods in Izbicki et al., 2020 and Dheur et al., 2024, this score has the advantage of not requiring the estimation of a predictive density, relying instead on samples from the conditional distribution. Consequently, this score can be applied with any generative model that does not have an explicit density, while still retaining the desirable properties of our CDF-based score.

Interestingly, C-PCP shares similarities with the recently proposed CP²-PCP method by Plassier et al., 2025. For a given $x \in \mathcal{X}$, both methods adapt the radius of the balls based on a second sample from the conditional distribution composed of $K$ points, requiring a total of $L + K$ samples. A detailed discussion can be found in Appendix I.

## 3.2. Latent-based conformity scores

Inspired by Feldman et al. (2023), we propose a latent-based conformity score with key distinctions. First, our method does not require the use of directional quantile regression. Additionally, the conformalization step is performed in the latent space, eliminating the need to construct a grid, which improves both computational efficiency and scalability.

Our base predictor is a conditional invertible generative model $\hat{Q} : \mathcal{Z} \times \mathcal{X} \to \mathcal{Y}$, which maps a latent random variable $Z \in \mathcal{Z}$ (e.g., drawn from a standard multivariate normal distribution) to the output space $\mathcal{Y}$, conditional on $X \in \mathcal{X}$ (e.g., using normalizing flows). The model is both conditional and invertible, meaning that

$$\hat{Q}(\hat{Q}^{-1}(y; x); x) = y, \forall x \in \mathcal{X}, y \in \mathcal{Y}.$$

We propose the following conformity score, called L-CP (Latent-based Conformal Prediction), defined as:

$$s_{\text{L-CP}}(x, y) = d_{\mathcal{Z}}(\hat{Q}^{-1}(y; x)), \quad (14)$$

where $d_{\mathcal{Z}} : \mathcal{Z} \to \mathbb{R}$ is a conformity function in the latent space $\mathcal{Z}$, independent of $x$. In our experiments, we use $Z \sim \mathcal{N}(0, I_d)$ and $d_{\mathcal{Z}}(z) = \|z\|$.

The corresponding prediction region is obtained by mapping a region in the latent space, $R_{\mathcal{Z}}(\hat{q}) = \{z \in \mathcal{Z} : d_{\mathcal{Z}}(z) \leq \hat{q}\}$, to a region in the output space, $\hat{R}_{\text{L-CP}}(x) = \{\hat{Q}(z; x) : z \in R_{\mathcal{Z}}(\hat{q})\}$.

L-CP generalizes Distributional Conformal Prediction (Chernozhukov et al., 2021), which is a special case when $Y$ is univariate ($d = 1$), $Z \sim \mathcal{U}(0, 1)$, $d_{\mathcal{Z}}(z) = |z - \frac{1}{2}|$, and $\hat{Q}(\cdot; x)$ is the quantile function of $Y$ given $x$.

Concurrent work by Fang et al., 2025 introduces CONTRA, sharing the same algorithm as our latent-based methods. While related, the papers diverge in their primary focus. Fang et al., 2025 emphasizes the smaller prediction regions achieved by CONTRA, whereas our work concentrates on the computational complexity and conditional coverage guarantees of the latent-based methods while obtaining region sizes that are small but not smaller than density-based methods.

## 4. Related Work

Conformal Prediction (CP), introduced by Vovk et al., 1999, forms the foundation of our work by providing prediction

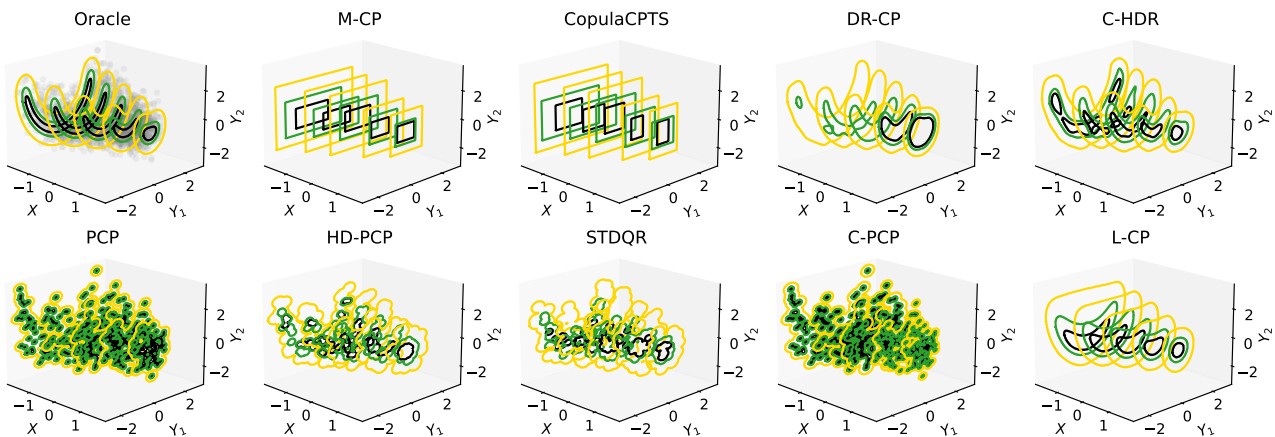

Figure 2: Prediction regions for a bivariate unimodal dataset, conditional on a unidimensional input. The black, green, and yellow contours represent regions with nominal coverage levels of 20%, 40%, and 80%, respectively.

regions with finite-sample coverage guarantees. CP methods are well established for regression with univariate outputs (Papadopoulos et al., 2008; Lei and Wasserman, 2014; Romano et al., 2019; Sesia and Romano, 2021) and classification (Romano et al., 2020; Angelopoulos et al., 2020). In the multi-output regression setting, we need to capture dependencies between output dimensions, represent more complex prediction regions and handle a larger computational demand.

To address multivariate prediction challenges, optimal transport methods such as cyclically monotone mappings (Carlier et al., 2016) define multivariate quantile regions with desirable properties such as existence and uniqueness of mappings. Hallin and Šiman (2017), Hallin et al. (2021), and Barrio et al. (2024) have proposed extensions of these approaches. Neural network-based techniques leverage normalizing flows (Kan et al., 2022; Huang et al., 2020) or variational autoencoders (Feldman et al., 2023) to learn flexible quantile regions. Additionally, highest density regions (HDRs) (Hyndman, 1996) handle multimodality and have been applied in various contexts (Camehl et al., 2024; Izbicki et al., 2022; Dheur et al., 2024). Recently, Wang et al., 2023b proposed constructing prediction regions as hyperballs centered on generated samples, with extensions by Plassier et al., 2025 improving conditional validity. Other methods use copulas (Messoudi et al., 2021a; Sun and Yu, 2024b) to model the dependency between variables.Appendix A provides a more detailed discussion on related work.

## 5. Comparison of Multi-Output Conformal Methods

In this section, we present a unified comparison of the conformity scores introduced in Section 2.2 and the generalized

scores proposed in Section 3.1.

### 5.1. Illustrative examples

We provide illustrative examples of bivariate prediction regions for different conformal methods on simulated data, covering both unimodal (Figure 2) and bimodal distributions (Figure 10 in Appendix D.2). The data-generating processes are given in Appendix D.2. Additionally, we present bivariate prediction regions for a real-world application, predicting a taxi passenger's drop-off location based on the passenger's information (Figures 8 and 9 in Appendix D.1).

In both Figures 2 and 10, the black, green, and yellow contours represent prediction regions with nominal coverage levels of 20%, 40%, and 80%, respectively. The top-left panel illustrates the density level sets of the oracle distribution $F_{Y|X}$. The remaining panels display the prediction regions generated by various conformal methods, all utilizing the MQF[2] base predictor, as explained in Appendix F.2.

We observe the following for the unimodal case in Figure 2. M-CP and CopulaCPTS capture heteroscedasticity but produce rectangular prediction regions, which do not align with the circular level sets of the oracle conditional distribution, resulting in a lack of sharpness. DR-CP fails to maintain conditional coverage, and for $X = 1$, the absence of black and green contours indicates that the predictive density does not reach the threshold $-\hat{q}$ defined in (5) for coverage levels of 0.2 and 0.4. C-HDR generates prediction regions that closely resemble the oracle level sets. PCP generates highly discontinuous regions, especially at lower coverage levels, where the regions appear as balls centered on individual samples. In contrast, HD-PCP and STDQR yield smoother, more continuous regions but require the estimation of a predictive PDF or the identification of a map from the latent space to the output space, respectively.

Table 1: Properties of different multivariate conformal methods. (*) `M-CP` achieves ACC under certain assumptions (Appendix E.2.3). (**) `STDQR` and `L-CP` require a conditional invertible generative model $\hat{Q} : \mathcal{Z} \times \mathcal{X} \rightarrow \mathcal{Y}$. (†) `CopulaCPTS` has a pre-training cost of $O(C)$.

| Method | Type of region | Asymptotic conditional coverage | Computational complexity | Predictive density not required | Sampling procedure not required |
|---|---|---|---|---|---|
| `M-CP` | Hyperrectangle | ✗(*) | $O(dM)$ | ✓ | ✓ |
| `CopulaCPTS` | Hyperrectangle | ✗ | $O(dM)^{\dagger}$ | ✓ | ✓ |
| `DR-CP` | Density superlevel set | ✗ | $O(D)$ | ✗ | ✓ |
| `C-HDR` | Density superlevel set | $K \rightarrow \infty$ | $O(K(D+S))$ | ✗ | ✗ |
| `PCP` | Union of $d$-balls | ✗ | $O(LS)$ | ✓ | ✗ |
| `HD-PCP` | Union of $d$-balls | ✗ | $O(L(D+S))$ | ✗ | ✗ |
| `STDQR` | Union of $d$-balls | ✗ | $O(LS)$ | ✓(**) | ✗ |
| `C-PCP` | Union of $d$-balls | $K \rightarrow \infty$ | $O((K+L)S)$ | ✓ | ✗ |
| `L-CP` | Quantile region | ✓ | $O(Q)$ | ✓(**) | ✗ |

For our methods, unlike `PCP`, `C-PCP` adjusts the radius of the prediction regions to improve conditional coverage. This is evident in the example, where the radius of the balls for $X = -1$ is smaller than for $X = 1$, as indicated by the tighter regions around the samples. `L-CP` generates prediction regions that closely align with the oracle level sets, demonstrating good conditional coverage.

For the bimodal distribution in Figure 10 (Appendix D.2), the prediction regions generated by `M-CP` and `L-CP` are connected, failing to capture the bimodal nature of the distribution. For the real-world application, Figures 8 and 9 (Appendix D.1) illustrate predictions under low and high uncertainty, respectively. Our methods, `L-CP` and `C-PCP`, alongside `M-CP` and `C-HDR`, demonstrate the best adaptability to outputs with varying levels of uncertainty.

### 5.2. Properties

In this section, we compare conformal methods based on several key properties. In the following, we use $\stackrel{\text{d.}}{=}$ to denote equality in distribution and $\stackrel{\text{a.s.}}{=}$ to denote almost sure equality.

**Marginal coverage.** All the conformal methods presented achieve the classical finite-sample *marginal coverage*. But, as noted by Wang et al., 2023b (Theorem 1), the marginal coverage of methods such as `C-HDR`, `PCP`, `HD-PCP`, and `C-PCP` also depends on the randomness of the generated samples. In Appendix E.1, we demonstrate that the marginal coverage, conditional on the calibration dataset $\mathcal{D}_{\text{cal}}$ and the samples drawn from it, follows a beta distribution, using standard arguments. `CopulaCPTS` is the only method that does not enter into the standard split-conformal algorithm and who does not satisfy the above property.

**Asymptotic conditional coverage (ACC).** We examine the *asymptotic conditional coverage* property, which corresponds to conditional coverage as defined in (4) under

the assumptions that $|D_{\text{cal}}| \rightarrow \infty$ and the base predictor corresponds to the oracle distribution $F_{Y|X}$.

While the assumption of oracle base predictor is strong, it is crucial to demonstrate that the conformal procedure preserves the performance of the base model. Specifically, given $x \in \mathcal{X}$, for `M-CP` and `CopulaCPTS`, we assume $\hat{l}_i(x) = Q_{Y_i|X=x}(\alpha_l)$ and $\hat{u}_i(x) = Q_{Y_i|X=x}(\alpha_u)$ with $i = 1, \ldots, d$; for `DR-CP`, `C-HDR`, and `HD-PCP`, $\hat{f}_{Y|X=x} = f_{Y|X=x}$; for `L-CP`, $\hat{Q}(Z; x) \stackrel{\text{d.}}{=} Y|X = x$; and for `PCP` and `C-PCP`, $\hat{F}_{Y|X=x} = F_{Y|X=x}$.

Our empirical results (Section 6) demonstrate that methods achieving ACC under these assumptions also exhibit superior approximate conditional coverage across diverse datasets and base predictors. `L-CP` is the only method that achieves ACC without additional assumptions. `C-HDR` and `C-PCP` achieve ACC with $K \rightarrow \infty$. Finally, `M-CP` achieves ACC under specific assumptions. Assuming that $Y_1, \ldots, Y_d$ are conditionally independent given $X$, `M-CP` achieves ACC if $\alpha_u - \alpha_l = \sqrt[d]{1 - \alpha}$. Furthermore, under the unrealistic assumption that $Y_1 \mid X \stackrel{\text{a.s.}}{=} \ldots \stackrel{\text{a.s.}}{=} Y_d \mid X$, `M-CP` achieves ACC if $\alpha_u - \alpha_l = 1 - \alpha$. The true dependence typically lies between these two extremes. We provide detailed proofs of these statements in Appendix E.2.

As discussed in Section 5.1, `DR-CP` fails to achieve ACC. Likewise, `PCP`, `HD-PCP` and `STDQR` do not achieve ACC, as they are constrained to producing regions with upper bounded volume for any $x \in \mathcal{X}$. Assuming each ball has a volume of $V$, `PCP` generates regions with a total volume of at most $LV$. For a given instance $x \in \mathcal{X}$ with high uncertainty, it may be impossible to capture sufficient probability mass to achieve conditional coverage.

**Region size.** Among the methods that achieve ACC, `C-HDR` is expected to perform best, as it converges to the highest

density regions, which correspond to the smallest volume regions (Hyndman, 1996). Prediction regions from `C-PCP` are expected to have a larger volume since they are constrained to a union of $L$ $d$-balls. Similarly, prediction regions from `L-CP` are less flexible than those from `C-HDR`, as they are connected when the region $R_Z(\lambda)$ in the latent space is connected for all $\lambda \in \mathbb{R}$ and $\hat{Q}$ is continuous. This constraint may be desirable when more interpretable regions are preferred (Sesia and Romano, 2021).

Among the remaining methods, `DR-CP` minimizes the mean region size $\mathbb{E}[|\hat{R}(X)|]$ under the oracle PDF as $|\mathcal{D}_{\text{cal}}| \to \infty$, as shown in Theorem 1 by Sadinle et al., 2019. In contrast, `M-CP` and `CopulaCPTS` are expected to yield larger prediction regions, as they do not explicitly account for dependencies between outputs. While `PCP`, `HD-PCP`, and `C-PCP` can capture multimodality, they are susceptible to the randomness of the sampling procedure, as evidenced by the shape of the regions in Figure 2. Furthermore, since they rely on a finite union of $L$ $d$-balls, they are subject to the curse of dimensionality in high-dimensional spaces, where data sparsity necessitates larger balls to maintain marginal coverage.

A potential weakness of the mean region size is that it can be disproportionately skewed by inputs with high uncertainty. To mitigate this sensitivity, we also report the median region size as a more robust alternative.

**Computational complexity.** Table 1 reports the computational complexity of each conformity score. For `M-CP` and `CopulaCPTS`, let $M$ represent the compute time of the univariate conformity score for a single dimension and $C$ the optimization time for CopulaCPTS. Let $D$, $S$, and $Q$ denote the time required for density evaluation, sampling, and calculating the inverse of the quantile function $\hat{Q}^{-1}$, respectively. In many cases, $M$ and $C$ are relatively low, while $D$, $S$, and $Q$ are comparable. `C-HDR`, `PCP`, `HD-PCP`, `STDQR` and `C-PCP` are significantly slower than `M-CP`, `L-CP`, and `DR-CP` since they need to generate a large number of samples to compute the conformity score (we used $K = L = 100$ in our experiments).

**Base predictor.** Some conformal methods stand out because they do not need to evaluate the predictive density $\hat{f}$ or generate samples. `M-CP` and `CopulaCPTS` only require a univariate model for each dimension, without needing a model for the joint distribution of $Y$. `DR-CP` does not require sampling from the model, which is beneficial when using normalizing flows that are slower to invert (e.g., Masked Autoregressive Flows (MAF, Papamakarios, Pavlakou, et al., 2017) or Convex Potential Flows (Huang et al., 2020)). `PCP` and `C-PCP` do not require evaluating the predictive density $\hat{f}$, making them compatible with any generative model, including diffusion models and GANs. `L-CP` and `STDQR` do not require predictive density evaluation but require the model to be invertible. We summarize the different properties in Table 1.

### 5.3. Connection between sample-based and density-based methods

Interestingly, the sample-based methods (`PCP`, `HD-PCP`, `C-PCP`) can be viewed as special cases of density-based methods (`DR-CP`, `C-HDR`). Let us assume a common predictive PDF $\hat{f}$ is used for the base predictor of these conformal methods. While `PCP` and `C-PCP` do not require a PDF, we assume that $\hat{f}_{Y|X=x}$ and $\hat{F}_{Y|X=x}$ correspond to the same distribution. Let $\tilde{Y}^{(l)} \sim \hat{F}_{Y|X=x}$ for $l \in [L]$, and $f_{\mathbb{S}}(\cdot; \tilde{Y}^{(l)})$ be a PDF with spherical level sets, centered at $\tilde{Y}^{(l)}$, such as a standard multivariate Gaussian $\mathcal{N}(\cdot; \tilde{Y}^{(l)}, I_d)$. For $x \in \mathcal{X}$, we define a new PDF $\hat{f}_{\max}(y \mid x) = \max_{l \in [L]} f_{\mathbb{S}}(y; \tilde{Y}^{(l)})/C$, where $C$ is a normalizing constant ensuring that $\hat{f}_{\max}(\cdot \mid x)$ integrates to 1. The following proposition establishes the relationship between these methods.

*Proposition* 1. `PCP` is equivalent to `DR-CP` with $\hat{f} = \hat{f}_{\max}$. Similarly, `HD-PCP` is equivalent to `DR-CP` with $\hat{f} = \hat{f}_{\max}$ where only $\lfloor (1-\alpha)L \rfloor$ samples with the highest density among $\{\tilde{Y}^{(l)}\}_{l \in [L]}$ are kept. Finally, `C-PCP` is equivalent to `C-HDR` with $\hat{f} = \hat{f}_{\max}$.

We provide a proof in Appendix E.3. Although these sample-based methods are special cases of density-based approaches, the key advantage of `PCP` and `C-PCP` is that they rely solely on a sampling procedure, without requiring a predictive density $\hat{f}$ as base predictor. Figure 3 summarizes the connections between the main conformal methods.

An interesting practical takeaway is that `DR-CP` and `C-HDR` are linked in the same way as `PCP` and `C-PCP`. Since `DR-CP` under the oracle has the smallest expected region size while `C-HDR` empirically has a smaller median region size, similar observations are expected for `PCP` and `C-PCP`. This is verified empirically: `PCP` has a smaller mean region size across all base predictors, while `C-PCP` has a smaller median region size.

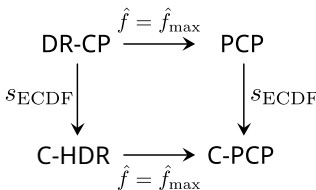

Figure 3: Connections between different methods.

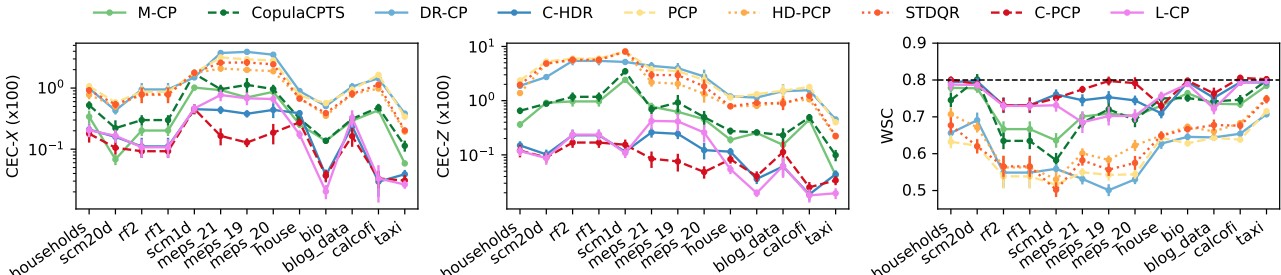

Figure 4: Conditional coverage metrics across datasets sorted by size. CEC-X and CEC-Z should be minimized while WSC should approach $1 - \alpha$.

# 6. A Large-Scale Study of Multi-Output Conformal Methods

In this section, we present a large-scale study of multi-output conformal methods using 13 tabular datasets from previous studies (Tsoumakas et al., 2011; Feldman et al., 2023; Wang et al., 2023b; Barrio et al., 2024; Camehl et al., 2024). To ensure sufficient data for training, calibration, and testing, we include only datasets with at least 2,000 instances. The selected datasets contain between 7,207 and 50,000 data points, with the number of input features $p$ ranging from 1 to 279 and the number of output variables $d$ ranging from 2 to 16.

We consider three base predictors: the Multivariate Quantile Function Forecaster (MQF$^2$), a normalizing flow (Kan et al., 2022), Distributional Random Forests (Cevid et al., 2022), and a multivariate Gaussian mixture model (Bishop, 1994). We present results for MQF$^2$ in the main text, while similar results for the other models are provided in Appendix G. We compare the methods using several metrics, including conditional coverage (WSC, CEC-X, and CEC-V), marginal coverage (MC), region size, and computational time. A detailed description of the experimental setup is provided in Appendix F.

**Conditional coverage.** Figure 4 presents the results for all datasets, ordered by increasing dataset size. On most datasets, C-PCP, L-CP, and C-HDR obtain the best conditional coverage. In contrast, HD-PCP, STDQR, PCP, and DR-CP are the least conditionally calibrated. Finally, M-CP

and CopulaCPTS attain intermediate conditional coverage, with M-CP performing slightly better. These results align with our analysis in Section 5.2, where we showed that C-PCP, L-CP, and C-HDR achieve ACC, while HD-PCP, STDQR, PCP, and DR-CP do not, and M-CP achieves it only under specific conditions. Finally, Figure 12 shows that all methods achieve marginal coverage, as expected.

**Region size.** Figure 5 presents a critical difference (CD) diagram (Demšar, 2006) comparing the median region size of all methods across datasets. Higher-ranked methods (further right) perform better. Thick horizontal lines indicate models with no statistically significant difference at the 0.05 level (see Appendix F.5 for details).

Among the methods that achieve ACC, C-HDR yields the smallest median region size, as expected, since its regions converge to the highest density regions (Izbicki et al., 2022). C-PCP and L-CP produce slightly larger regions, though the difference is not significant for these datasets. Among the remaining methods, DR-CP yields the smallest median region. In contrast, M-CP and CopulaCPTS generate larger regions, which is expected given their less flexible hyper-rectangular shape. PCP tends to obtain the largest region sizes as it includes samples from low-density areas, whereas STDQR and HD-PCP mitigate this by removing samples from low-density areas, resulting in more compact regions. Finally, Figure 13 in Appendix G provides results for the mean

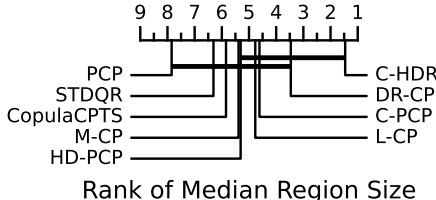

Figure 5: CD diagrams with the base predictor MQF$^2$ based on 10 runs per dataset and method.

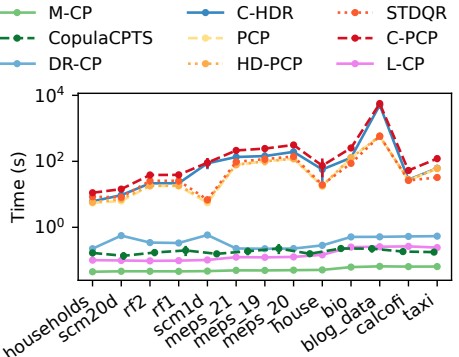

Figure 6: Total time in seconds for calibration and test.

region size, where `DR-CP` consistently performs best, under the oracle setting, it minimizes the expected region size, as explained in Section 5.2.

**Computation time.** Figure 6 shows the total computation time for each method. `M-CP` and `CopulaCPTS` have the shortest computation times, as they do not require learning a complex model for the output joint distribution. `L-CP` and `DR-CP` follow, benefiting from the absence of per-instance sampling. In contrast, sampling-based methods typically require 100 to 200 times more computation time.

**Application to image dataset.** Finally, we extend our comparison to a regression problem where the output is an image, which has a higher dimensionality than the previously considered tabular datasets. Specifically, we use the CIFAR-10 dataset (Krizhevsky et al., 2014), consisting of 32×32 RGB images, each labeled with one of 10 possible classes. We train the base predictor Glow (Kingma and Dhariwal, 2018), conditioned on the image label, where the output space is $\mathcal{Y} = [0, 1]^{3 \times 32 \times 32}$ ($d = 3072$) and the input space is $\mathcal{X} = \{0, \ldots, 9\}$ ($p = 1$). The results, detailed in Appendix J, lead to similar conclusions regarding conditional coverage, region size, and computational time.

# 7. Conclusion

We studied the problem of constructing conformal prediction regions for multi-output regression, which remains relatively underexplored compared to the univariate case. We presented a unified comparative study of several conformal methods along with their associated conformity scores, highlighting their properties and interconnections. In addition, we introduced two new classes of conformity scores: CDF-based scores, including a variant compatible with generative models, and latent-based scores, which exploit invertible generative models for improved computational efficiency. Both classes generalize existing conformity scores from the univariate setting.

The choice of conformity score directly influences the geometry and flexibility of the resulting prediction regions. In the univariate setting, the most flexible regions are typically unions of intervals. In contrast, the multivariate case allows for a wider variety of geometries, ranging from hyperrectangles and ellipsoids to highly flexible, nonconvex regions that can be disconnected and capture distributional bimodality. A simple and computationally efficient approach is to construct separate univariate prediction regions for each output dimension and apply a correction for joint coverage. However, these methods do not capture dependencies between output dimensions and typically result in rigid, (unions of) hyperrectangular regions with limited flexibility. In contrast, more flexible methods account for correlations and dependencies across outputs by incorporating the covariance

structure, modeling the joint density, or leveraging generative models. These approaches produce more expressive prediction regions but are generally more computationally demanding.

While conformal prediction (CP) always guarantees marginal coverage, conformity scores whose thresholds do not vary instance-wise fail to achieve the desirable property of asymptotic conditional coverage (ACC). In contrast, our proposed scores enable ACC but require estimating the conditional distribution of the conformity score—an inherently challenging task in low-data regimes. Similarly, CP methods based on generative models introduce additional sampling variability. Finally, our large-scale empirical study systematically compares these conformal methods across multiple multi-output regression datasets, using various evaluation metrics, including conditional coverage and prediction region volume.

Future work will focus on replacing region prediction with a recalibrated multivariate distribution, equipped with an explicit density function and conformal coverage guarantees. We also plan to extend our approach to more complex outputs, including semi-structured and unstructured data (e.g., images, text, and graphs), and to further investigate the theoretical connection between multivariate calibration and conformal prediction

## Impact statement

This work contributes to the development of statistically reliable and interpretable machine learning algorithms. Enhancing trust and transparency in predictive modeling helps designing more practical and accessible models for real-world applications.

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

## A. Related Work

Our research builds on a broad body of literature that spans several closely related themes. This supplementary section provides a concise overview of these topics.

In the realm of multivariate functional data, Diquigiovanni et al., 2021a introduced conformal predictors that create adaptive, finite-sample valid prediction bands, with extensions into time series applications, particularly in energy markets (Diquigiovanni et al., 2024). In image processing, recent applications (Horwitz and Hoshen, 2022; Teneggi et al., 2023) apply CP in a pixel-wise manner, resulting in hyperrectangular regions that may not capture pixel dependencies effectively.

For multi-step-ahead or multi-horizon forecasting, predictions can be made across multiple outputs simultaneously rather than sequentially, aligning with a multi-output forecasting framework. Stankeviciute et al., 2021 explored multi-horizon time series forecasting using recurrent neural networks (RNNs), incorporating univariate conformal techniques with nominal coverage adjustments via Bonferroni correction. Similarly, English et al., 2024 adapted the Amplitude-Modulated L-inf norm method from Diquigiovanni et al., 2021b for multi-output, multi-step forecasting.

In multi-target regression, Messoudi et al., 2021b applied copula functions in deep neural networks to provide multivariate predictions with guaranteed coverage. Their findings suggest that simple parametric copulas can work for certain datasets, but more complex copulas may be required for well-calibrated predictions, which introduces challenges, as complex copulas typically require significant calibration data. Building on this, Sun and Yu, 2024b proposed a copula-based method for multi-step time series forecasting, optimizing the calibration and efficiency of confidence intervals. However, this method requires two calibration phases and is primarily feasible with large calibration datasets. Moreover, its validity relies on the empirical copula, limiting applicability to other learnable copula classes. One very recent advancement on the subject, following ideas expressed by Messoudi et al., 2021b in their conclusions, is Park et al., 2024, where the dependence structure between marginal distributions is recovered via the use of vine copulas.

Another set of methodologies that tackle multi-output problems are based on multiplicity-correction approaches for multiple testing. Timans et al., 2025 improves over Bonferroni correction using permutation tests, and obtain a tighter and globally valid prediction. Methods based on multiplicity correction such as controlling the Family-Wise Error Rate (FWER) are valuable for providing error control guarantees across multiple outputs. In contrast, the methods we survey and propose aim for potentially tighter prediction regions by directly modeling the multivariate structure.

In the context of conformal prediction, the flexibility in configuring the prediction region is a degree of freedom for the modeler. To overcome the limitations of traditional hyper-rectangular prediction regions, Messoudi et al., 2022 introduced ellipsoidal uncertainty sets that enable instance-specific adaptation of confidence regions. Johnstone and Ndiaye, 2022 advanced multi-output regression by developing efficient techniques for approximating conformal prediction sets without retraining the model, although their approach relies heavily on the predictive model being a linear function of $Y$. Sun and Yu, 2024b constructed ellipsoidal prediction regions for time series, capable of modeling dependencies between outputs, though this method does not handle multimodality. Our work closely connects with the multivariate conformal prediction literature, where multi-horizon prediction is viewed as a prediction across multiple outputs.

Overall, as this study demonstrates, the flexibility of conformal prediction allows for coherent handling of diverse data types. Multi-output problems represent one facet of a broader taxonomy, as explored by Zhou et al., 2025, who discuss further developments in multi-output conformal prediction.

## B. Additional multi-output conformal methods

In this section, we describe the prediction regions $\hat{R}$ for `M-CP` and `CopulaCPTS`, which both produce hyperrectangular regions.

**M-CP.**  Zhou et al., 2024 applied a univariate conformal method to each output $i \in [d]$ of the multivariate response. Specifically, given a conformity score $s_i$ for the $i$-th dimension, joint coverage across all dimensions can be achieved using the following conformity score:

$$s_{\text{M-CP}}(x, y) = \max_{i \in [d]} s_i(x, y_i). \tag{15}$$

A similar score has been considered by Diquigiovanni et al. (2021b) in the context of functional regression.

In this work, we use Conformalized Quantile Regression (CQR, Romano et al., 2019) for each output $i \in [d]$, where the

conformity score is given by:

$$s_i(x, y_i) = \max\{\hat{l}_i(x) - y_i, y_i - \hat{u}_i(x)\},\tag{16}$$

with $\hat{l}_i(x)$ and $\hat{u}_i(x)$ representing the lower and upper conditional quantiles of $Y_i|X = x$ at levels $\alpha_l$ and $\alpha_u$, respectively. In our experiments, we consider equal-tailed prediction intervals, where $\alpha_l = \frac{\alpha}{2}$, $\alpha_u = 1 - \frac{\alpha}{2}$, and $\alpha$ denotes the miscoverage level. The corresponding prediction region $\hat{R}_{\text{M-CP}}(x) = \bigtimes_{i=1}^{d}[\hat{l}_i(x) - \hat{q}, \hat{u}_i(x) + \hat{q}]$ is a hyperrectangle.

**CopulaCPTS.** CopulaCPTS (Sun and Yu, 2024a) is originally designed for time-series but the calibration procedure is valid for any multi-dimensional variable. It models the joint probability of uncertainty for each output with a copula. The calibration set is divided into two subsets $\mathcal{D}_{\text{cal}-1}$ and $\mathcal{D}_{\text{cal}-2}$. $\mathcal{D}_{\text{cal}-1}$ serves for the estimation of a CDF on the conformity score of each output and $\mathcal{D}_{\text{cal}-2}$ allows to calibrate the copula. CopulaCPTS can combine any univariate or multivariate conformity scores. In this paper, we use the CQR score $s_i$ (16) for each dimension $i \in [d]$.

Denote $\hat{F}_i$ the empirical CDF of the conformity scores $\{s_i(x, y_i)\}_{(x,y)\in\mathcal{D}_{\text{cal}-1}}$ for $i \in [d]$, and $\hat{F}_i^{-1}$ the corresponding empirical quantile function. In practice, to minimize region sizes while achieving marginal validity, CopulaCPTS computes the optimal $s_1^*, \ldots, s_d^*$ that minimize the following loss using stochastic gradient descent:

$$\mathcal{L}(\hat{s}_1, \ldots, \hat{s}_d) = \frac{1}{|\mathcal{D}_{\text{cal}-2}|} \sum_{(x,y)\in\mathcal{D}_{cal-2}} \prod_{i=1}^{d} \mathbb{1}\left[\hat{F}_i(s_i(x, y_i)) < \hat{F}_i^{-1}(\hat{s}_i)\right] - (1 - \alpha).\tag{17}$$

Then, the prediction region is defined as:

$$\hat{R}_{\text{CopulaCPTS}}(x) = \{y \in \mathcal{Y} : \forall i \in [d], s_i(x, y_i) < s_i^*\}\tag{18}$$

Sun and Yu, 2024a proved that this prediction set achieves marginal coverage. However, since CopulaCPTS does not follow the SCP algorithm, it does not achieve properties on the marginal coverage from Appendix E.1.

## C. Relationship between conformity scores and regions

Section 2.2 and Section 3 in the main text presented several multi-output conformal methods through their conformity score $s$. As explained in Section 2.1, their corresponding prediction region $\hat{R}$ can be computed as follows:

$$\hat{R}(x) = \{y \in \mathcal{Y} : s(x, y) \le \hat{q}\}.$$

In this section, we explicitly derive the prediction region associated with these methods.

**M-CP.** Following Zhou et al., 2024, the prediction region $\hat{R}_{\text{M-CP}}$ can be derived from $s_{\text{M-CP}}$ as follows:

$$s_{\text{M-CP}}(x, y) \le \hat{q} \iff \max_{i\in[d]} s_i(x, y_i) \le \hat{q}\tag{19}$$

$$\iff \forall i \in [d], s_i(x, y_i) \le \hat{q}\tag{20}$$

$$\iff \forall i \in [d], \max\{\hat{l}_i(x) - y_i, y_i - \hat{u}_i(x)\} \le \hat{q}\tag{21}$$

$$\iff \forall i \in [d], \hat{l}_i(x) - y_i \le \hat{q} \wedge y_i - \hat{u}_i(x) \le \hat{q}\tag{22}$$

$$\iff \forall i \in [d], \hat{l}_i(x) - \hat{q} \le y_i \wedge y_i \le \hat{u}_i(x) + \hat{q}\tag{23}$$

$$\iff \forall i \in [d], y_i \in [\hat{l}_i(x) - \hat{q}, \hat{u}_i(x) + \hat{q}]\tag{24}$$

$$\iff y \in \bigtimes_{i=1}^{d}[\hat{l}_i(x) - \hat{q}, \hat{u}_i(x) + \hat{q}]\tag{25}$$

$$\iff y \in \hat{R}_{\text{M-CP}}(x).\tag{26}$$

**DR-CP** The equivalence is straightforward.

**C-HDR.** Given $\hat{Y} \sim \hat{f}_{Y|X=x}$ and $U = \hat{f}(\hat{Y} \mid X = x)$, for any $y \in \mathcal{Y}$, we can write

$$s_{\text{C-HDR}}(x, y) \tag{27}$$
$$= \text{HPD}_{\hat{f}}(y \mid x) \tag{28}$$
$$= \mathbb{P}(\hat{f}(\hat{Y} \mid x) \geq \hat{f}(y \mid x) \mid X = x) \tag{29}$$
$$= \mathbb{P}(U \geq \hat{f}(y \mid x) \mid X = x) \tag{30}$$
$$= 1 - \mathbb{P}(U \leq \hat{f}(y \mid x) \mid X = x) \tag{31}$$
$$= 1 - F_{U|X=x}(\hat{f}(y \mid x)), \tag{32}$$

where $F_{U|X=x}$ is the conditional CDF of $U$ given $X = x$.

Recall that the prediction region for C-HDR is given by

$$\hat{R}_{\text{C-HDR}}(x) = \{y \in \mathcal{Y} : \hat{f}(y \mid x) \geq t_{\hat{q}}\}, \quad \text{where } t_{\hat{q}} = \sup\{t : \mathbb{P}(\hat{f}(\hat{Y} \mid x) \geq t \mid X = x) \geq \hat{q}\}. \tag{33}$$

The threshold $t_{\hat{q}}$ in (33) can be equivalently written as follows:

$$t_{\hat{q}} = \sup\{t : \mathbb{P}(\hat{f}(\hat{Y} \mid x) \geq t \mid X = x) \geq \hat{q}\} \tag{34}$$
$$= \sup\{t : \mathbb{P}(U \geq t \mid X = x) \geq \hat{q}\} \tag{35}$$
$$= \sup\{t : 1 - P(U \leq t \mid X = x) \geq \hat{q}\} \tag{36}$$
$$= \sup\{t : 1 - \hat{q} \geq F_{U|X=x}(t)\} \tag{37}$$
$$= F_{U|X=x}^{-1}(1 - \hat{q}), \tag{38}$$

where we use the definition of the upper quantile function in the last step.

Using (27), (33), and (38), we can write

$$s_{\text{C-HDR}}(x, y) \leq \hat{q} \iff \text{HPD}_{\hat{f}}(y \mid x) \leq \hat{q} \tag{39}$$
$$\iff 1 - F_{U|X=x}(\hat{f}(y \mid x)) \leq \hat{q} \tag{40}$$
$$\iff F_{U|X=x}(\hat{f}(y \mid x)) \geq 1 - \hat{q} \tag{41}$$
$$\iff \hat{f}(y \mid x) \geq F_{U|X=x}^{-1}(1 - \hat{q}) \tag{42}$$
$$\iff \hat{f}(y \mid x) \geq t_{\hat{q}} \tag{43}$$
$$\iff y \in \hat{R}_{\text{C-HDR}}(x). \tag{44}$$

**PCP.** Let $B(\mu, r)$ represent a ball with center $\mu$ and radius $r$. Following Wang et al., 2023b, we show that, for any $x \in \mathcal{X}$, $\hat{R}_{\text{PCP}}(x)$ corresponds to a union of balls:

$$s_{\text{PCP}}(x, y) \leq \hat{q} \iff \min_{l \in [L]} \|y - \tilde{Y}^{(l)}\|_2 \leq \hat{q} \tag{45}$$
$$\iff \exists l \in [L], \|y - \tilde{Y}^{(l)}\|_2 \leq \hat{q} \tag{46}$$
$$\iff \exists l \in [L], y \in B(\tilde{Y}^{(l)}, \hat{q}) \tag{47}$$
$$\iff y \in \bigcup_{l \in [L]} B(\tilde{Y}^{(l)}, \hat{q}) \tag{48}$$
$$\iff y \in \hat{R}_{\text{PCP}}(x), \tag{49}$$

where $\tilde{Y}^{(l)} \sim \hat{F}_{Y|X=x}, l \in [L]$.

**HD-PCP.** For HD-PCP, the reasoning is similar to PCP with the difference that only the $\lfloor (1 - \alpha)L \rfloor$ samples with the highest density are kept.

**CDF-based conformity scores.** We note that the region $\hat{R}_{\text{CDF}}(x)$ has a similar form to $\hat{R}_W(x) = \{y \in \mathcal{Y} : s_W(x, y) \leq \hat{q}\}$, except that the threshold on $s_W(x, y)$ is different and depends on $x$. In fact, we can write

$$\hat{R}_{\text{CDF}}(x) = \{y \in \mathcal{Y} : s_{\text{CDF}}(x, y) \leq \hat{q}\} \tag{50}$$
$$= \{y \in \mathcal{Y} : F_{W|X=x}(s_W(x, y)) \leq \hat{q}\} \tag{51}$$
$$= \{y \in \mathcal{Y} : s_W(x, y) \leq F_{W|X=x}^{-1}(\hat{q})\}. \tag{52}$$

In the special case where $s_W = s_{\text{PCP}}$, since PCP always generates predictions as a union of balls, we can conclude that C-PCP will do the same.

**Latent-based conformity scores.** Since $\hat{Q}(\cdot; x)$ is bijective, for every $y \in \mathcal{Y}$, there exists a unique $z \in \mathcal{Z}$ such that $y = \hat{Q}(z; x)$. Therefore, the condition $d_{\mathcal{Z}}(\hat{Q}^{-1}(y; x)) \leq \hat{q}$ is equivalent to $d_{\mathcal{Z}}(z) \leq \hat{q}$, where $z = \hat{Q}^{-1}(y; x)$. This gives the prediction region:

$$\hat{R}_{\text{L-CP}}(x) = \{y \in \mathcal{Y} : d_{\mathcal{Z}}(\hat{Q}^{-1}(y; x)) \leq \hat{q}\} \tag{53}$$
$$= \{\hat{Q}(z; x) : z \in \mathcal{Z} \text{ and } d_{\mathcal{Z}}(z) \leq \hat{q}\}. \tag{54}$$

# D. Additional illustrative examples

## D.1. A real-world application

Following Wang et al., 2023a, we apply the multi-output conformal methods to the taxi dataset, where the goal is to predict the drop-off location of a New York taxi passenger based on the passenger's information.

Figures 7(a) and 8(a) displays five randomly selected samples from the dataset, showing the pick-up (red pin) and drop-off (blue pin) locations of taxi passengers. The remaining panels show a specific input-output pair $(x, y)$ and the corresponding prediction regions generated by the conformal methods discussed in this paper. The coverage level $1 - \alpha$ for these regions is set to 0.8, with $\text{MQF}^2$ as the base predictor, as introduced in Section F.2.

Figure 7 corresponds to the same data and predictions regions as Figure 8 except that it is zoomed for better comparison with Figure 9. Each region is labeled with its size, calculated using the estimator from Section F.4, displayed in the bottom left corner. Notably, C-PCP generates regions similar in shape to PCP but with an input-adaptive radius, resulting in smaller region sizes (8.2 compared to 8.67) in this case. Additionally, HD-PCP produces more clustered regions, while PCP and C-PCP show more dispersed regions.

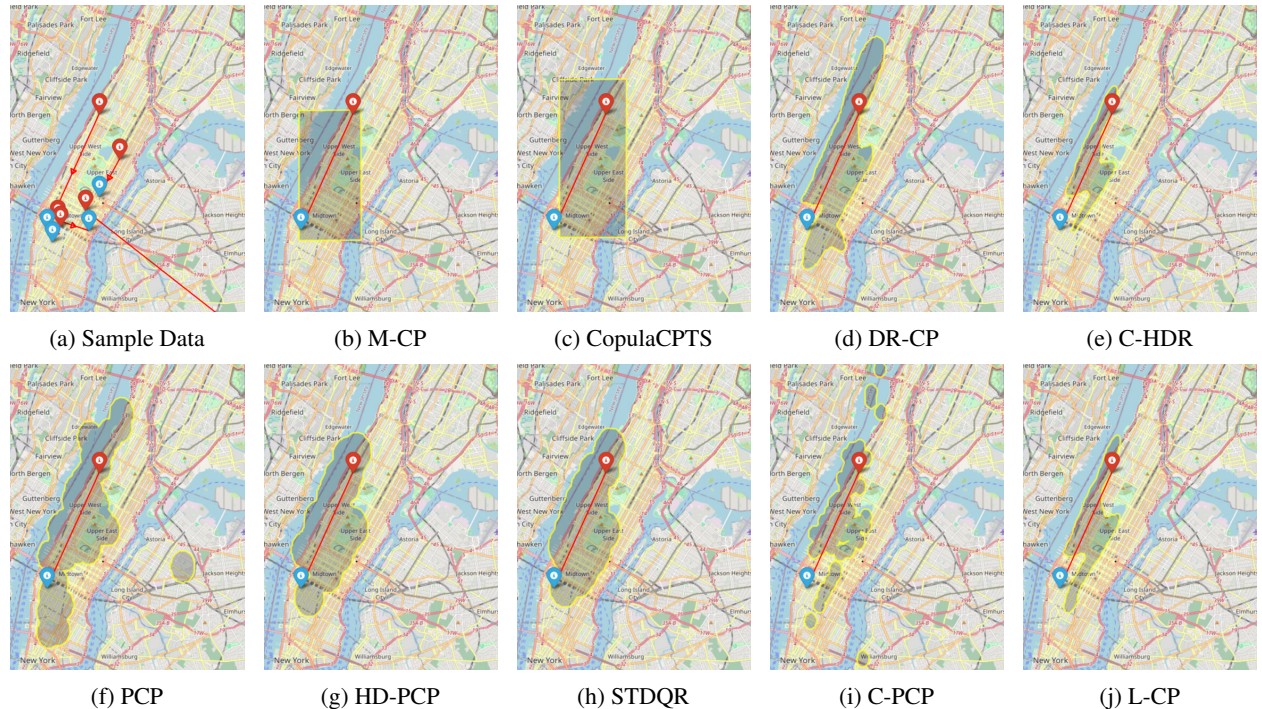

Figure 7: Conformal methods applied on the NYC Taxi dataset for an input with low uncertainty.

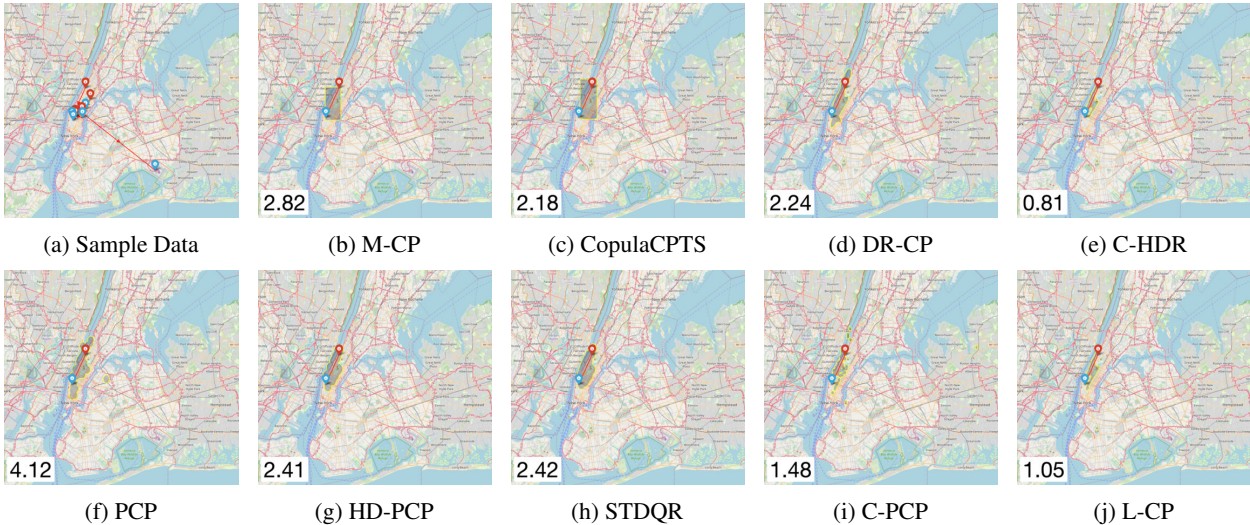

Figure 8: Conformal methods applied on the NYC Taxi dataset for an input with low uncertainty.

Figure 9 presents the same example for an input-output pair where the input is associated with higher uncertainty, resulting in larger region sizes. As in the first figure, the shapes of the regions (e.g., unions of hyperrectangles, quantile regions, etc.) remain consistent but expand to cover a larger area. Conformal methods with the best region sizes differ between the two figures, with C-HDR producing the smallest region in the first figure and DR-CP in the second. In this case, C-PCP selects a larger radius than PCP, resulting in larger regions than PCP. The observation that PCP and C-PCP produce more dispersed regions, while HD-PCP generates more clustered regions, also holds true for this higher uncertainty case.

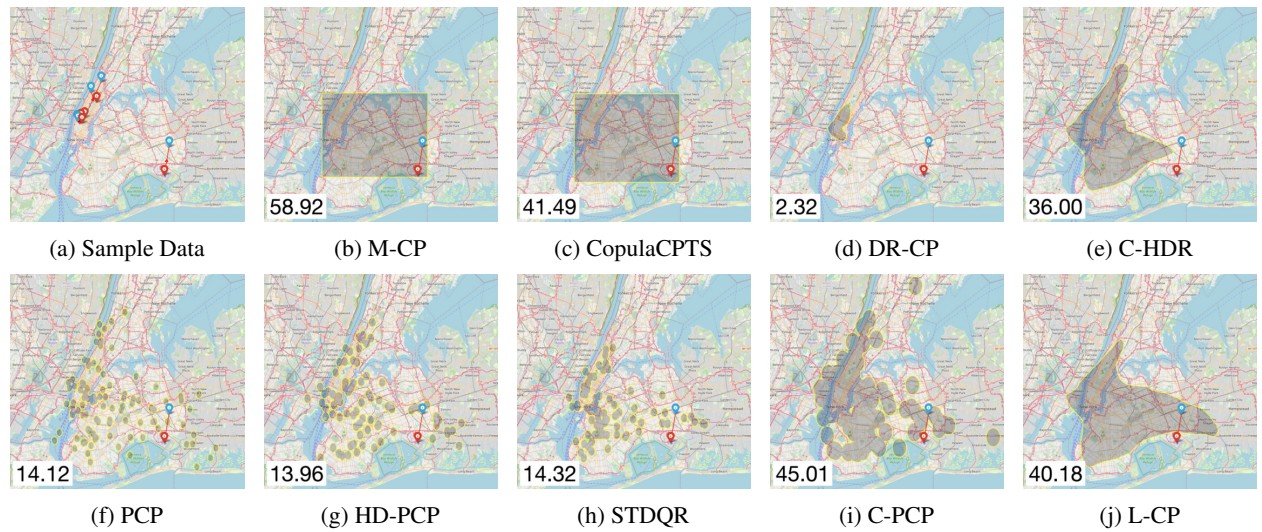

Figure 9: Conformal methods applied on the NYC Taxi dataset for an input with high uncertainty.

## D.2. Toy examples

We define two data-generating processes to evaluate the models compared to a known distribution: a unimodal heteroscedastic distribution and a bimodal heteroscedastic distribution. The input variable $X \in \mathbb{R}$ is unidimensional ($p = 1$) and the output variable $Y \in \mathbb{R}^2$ is bidimensional ($d = 2$). The variables $X$ and $Y$ are scaled linearly such that the mean and variances on each dimension are 0 and 1. The figures are inspired by Barrio et al., 2024.

**Unimodal heteroscedastic process.** The first process is illustrated in Figure 2 in the main text. The data generating process is as follows:

$$X \sim \mathcal{U}(0,1), \tag{55}$$

$$Y \mid X = x \sim \frac{1}{k} \sum_{j=1}^{k} \mathcal{N}\big((1.3 - x)\boldsymbol{\mu}^{(j)}(x),\, \sigma^2 I_2\big), \tag{56}$$

$$\tag{57}$$

where $k = 200$, $\sigma = 0.2$, $I_2$ is the $2 \times 2$ identity, and, for $j = 1, \ldots, k$,

$$\boldsymbol{\mu}_1^{(j)} = \cos \alpha_j \tag{58}$$

$$\boldsymbol{\mu}_2^{(j)} = (0.5 - \sin \alpha_j) \tag{59}$$

$$\alpha_j = \frac{(j-1)\,\pi}{k-1} \tag{60}$$

Detailed metrics for this dataset, supporting Section 5.1, are provided in Table 2.

| Method | MC | Median Size | CEC-$X$ ($\times 100$) | CEC-$Z$ ($\times 100$) | WSC | Test time |
|---|---|---|---|---|---|---|
| M-CP | $\mathbf{0.805_{0.0039}}$ | $8.47_{0.14}$ | $0.110_{0.023}$ | $0.0812_{0.018}$ | $\mathbf{0.803_{0.011}}$ | $\mathbf{0.0336_{0.022}}$ |
| CopulaCPTS | $\mathbf{0.815_{0.011}}$ | $8.78_{0.25}$ | $0.191_{0.049}$ | $0.163_{0.047}$ | $\mathbf{0.814_{0.015}}$ | $1.04_{0.021}$ |
| DR-CP | $\mathbf{0.808_{0.0042}}$ | $7.03_{0.071}$ | $0.613_{0.045}$ | $0.560_{0.042}$ | $0.710_{0.016}$ | $\mathbf{0.0209_{0.00047}}$ |
| C-HDR | $\mathbf{0.810_{0.0038}}$ | $\mathbf{6.80_{0.059}}$ | $0.0637_{0.016}$ | $0.0825_{0.012}$ | $\mathbf{0.798_{0.0070}}$ | $3.52_{0.085}$ |
| PCP | $\mathbf{0.805_{0.0039}}$ | $9.16_{0.089}$ | $0.668_{0.052}$ | $0.587_{0.046}$ | $0.713_{0.0080}$ | $1.69_{0.021}$ |
| HD-PCP | $\mathbf{0.804_{0.0037}}$ | $7.44_{0.056}$ | $0.287_{0.031}$ | $0.256_{0.034}$ | $0.758_{0.013}$ | $3.38_{0.043}$ |
| STDQR | $\mathbf{0.806_{0.0027}}$ | $7.87_{0.070}$ | $0.343_{0.025}$ | $0.305_{0.027}$ | $0.746_{0.011}$ | $1.77_{0.022}$ |
| C-PCP | $\mathbf{0.808_{0.0049}}$ | $9.14_{0.12}$ | $\mathbf{0.0464_{0.013}}$ | $\mathbf{0.0484_{0.013}}$ | $0.822_{0.0085}$ | $3.44_{0.056}$ |
| L-CP | $\mathbf{0.803_{0.0039}}$ | $8.24_{0.11}$ | $\mathbf{0.0544_{0.0073}}$ | $0.0654_{0.012}$ | $\mathbf{0.811_{0.011}}$ | $0.0217_{0.00052}$ |

Table 2: Detailed metrics for the unimodal heteroscedastic process from Figure 2.

**Bimodal heteroscedastic process.** Figure 10, similar to Figure 2 but with a bimodal distribution for the output, is introduced in Section 5.1.

The data generating process is as follows:

$$X \sim \mathcal{U}(0.5, 2), \tag{61}$$
$$Y \mid X = x \sim 0.5 \cdot \mathcal{N}(4, x I_d) + 0.5 \cdot \mathcal{N}(-4, I_d/x). \tag{62}$$

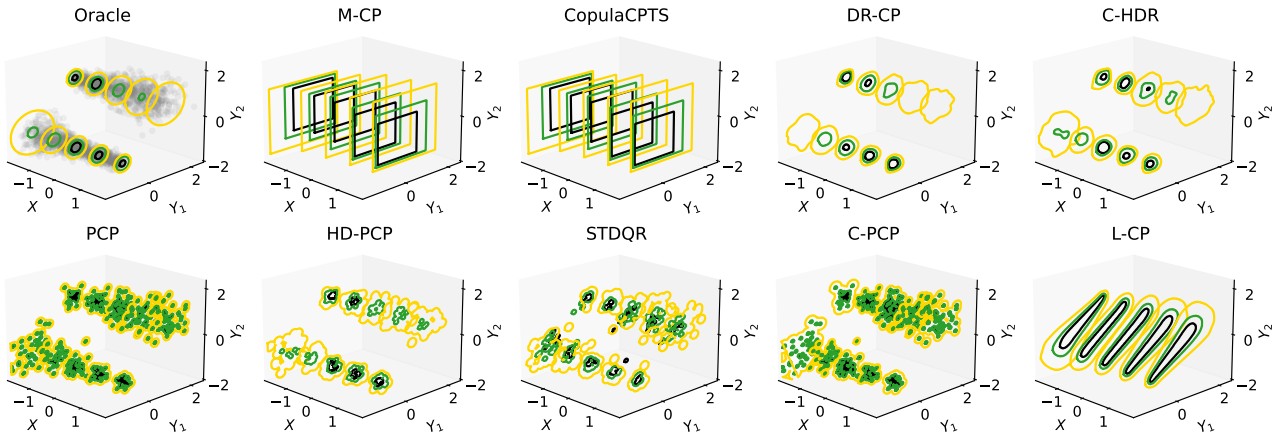

Figure 10: Examples of prediction regions on a bivariate bimodal dataset, conditional on a unidimensional input.

# E. Proofs

## E.1. Distribution of the marginal coverage conditional on calibration data

In contrast to M-CP, L-CP, and DR-CP, the methods C-HDR, PCP, HD-PCP, and C-PCP rely on a non-deterministic conformity score. For each calibration and test point, C-HDR, PCP, HD-PCP, and C-PCP require sampling $K$, $L$, $L$, and $L + K$ points, respectively.

Let $\mathcal{D}_{\text{cal}} = \{(X^{(j)}, Y^{(j)})\}_{j \in [|\mathcal{D}_{\text{cal}}|]}$ represent the calibration dataset and $(X, Y)$ be the test instance. Let $\mathcal{S}_{\text{cal}} = \{\mathcal{S}_{\text{cal}}^{(j)}\}_{j \in [|\mathcal{D}_{\text{cal}}|]}$ represent samples from the calibration dataset where $\mathcal{S}_{\text{cal}}^{(j)}$ is generated based on input $X^{(j)}$ and $\mathcal{S}_{\text{test}}$ the samples generated based on input $X$. Despite the added sampling uncertainty, these methods still provide a marginal coverage guarantee:

$$\mathbb{P}_{X, Y, \mathcal{S}_{\text{test}}, \mathcal{D}_{\text{cal}}, \mathcal{S}_{\text{cal}}}(Y \in \hat{R}(X)) \geq 1 - \alpha. \tag{63}$$

Compared to (3), the probability is additionally on $\mathcal{S}_{\text{cal}}$ and $\mathcal{S}_{\text{test}}$. This result, specifically for PCP and HD-PCP, was demonstrated by Wang et al., 2023b.

In Lemma 1, we further show that the marginal coverage conditional on the calibration dataset $\mathcal{D}_{\text{cal}}$ and the samples $\mathcal{S}_{\text{cal}}$ follows a beta distribution, using standard arguments. Assuming no ties among the scores, this lemma applies to any conformity score $s$.

**Lemma 1.** Assuming no ties among the scores and i.i.d. inputs, outputs and samples, the distribution of the coverage, conditional on the calibration dataset and its samples, is given by:

$$\mathbb{P}(Y \in \hat{R}(X) \mid \mathcal{D}_{\text{cal}}, \mathcal{S}_{\text{cal}}) \sim \text{Beta}(k_\alpha, |\mathcal{D}_{\text{cal}}| + 1 - k_\alpha), \tag{64}$$

where $k_\alpha = \lceil (1 - \alpha)(|\mathcal{D}_{\text{cal}}| + 1) \rceil$. Moreover, $\mathbb{P}(Y \in \hat{R}(X)) = \frac{k_\alpha}{|\mathcal{D}_{\text{cal}}|+1} \geq 1 - \alpha$.

*Proof.* For the methods C-HDR, PCP, HD-PCP, and C-PCP, the conformity score $s$ is non-deterministic due to sampling uncertainty. To clarify, we define a deterministic conformity score $\bar{s} : \mathcal{X} \times \mathcal{Y} \times \mathbb{S}$, where $\mathbb{S}$ represents the space of samples for a given method.

For $j = 1, \ldots, |\mathcal{D}_{\text{cal}}|$, let $S_j = \bar{s}(X^{(j)}, Y^{(j)}, \mathcal{S}_{\text{cal}}^{(j)})$ denote the conformity score on the calibration dataset, and let $S = \bar{s}(X, Y, \mathcal{S}_{\text{test}})$ represent the conformity score for the test instance. Since $\bar{s}$ is deterministic and the tuples $(X^{(1)}, Y^{(1)}, \mathcal{S}_{\text{cal}}^{(1)}), \ldots, (X^{(|\mathcal{D}_{\text{cal}}|)}, Y^{(|\mathcal{D}_{\text{cal}}|)}, \mathcal{S}_{\text{cal}}^{(|\mathcal{D}_{\text{cal}}|)}), (X, Y, \mathcal{S}_{\text{test}})$ are i.i.d. random variables, $S_1, \ldots, S_{|\mathcal{D}_{\text{cal}}|}, S$ are also i.i.d. random variables.

Since $S_1, \ldots, S_{|\mathcal{D}_{\text{cal}}|}, S$ are indentically distributed, they share the same CDF. Using the probability integral transform, $F_S(S) \sim U(0, 1)$. Thus, $F_S(S_1), \ldots, F_S(S_{|\mathcal{D}_{\text{cal}}|})$ correspond to uniform variates $U_1, \ldots, U_{|\mathcal{D}_{\text{cal}}|}$. Since there are no ties among the scores, $F_S$ is strictly increasing, and $F_S(S_{(j)}) = U_{(j)}$ for $j = 1, \ldots, |\mathcal{D}_{\text{cal}}|$, where $S_{(j)}$ and $U_{(j)}$ are the $j$-th order statistics. Hence:

$$\mathbb{P}(Y \in \hat{R}(X) \mid \mathcal{D}_{\text{cal}}, \mathcal{S}_{\text{cal}}) = \mathbb{P}(S \leq S_{(k_\alpha)} \mid S_1, \ldots, S_{|\mathcal{D}_{\text{cal}}|}) \tag{65}$$
$$= F_S(S_{(k_\alpha)}) \tag{66}$$
$$= U_{(k_\alpha)} \tag{67}$$
$$\sim \text{Beta}(k_\alpha, |\mathcal{D}_{\text{cal}}| + 1 - k_\alpha). \tag{68}$$

The final step results from the distribution of uniform order statistics. Taking the expectation of the Beta distribution gives:

$$\mathbb{P}(Y \in \hat{R}(X)) = \mathbb{E}[\mathbb{P}(Y \in \hat{R}(X) \mid \mathcal{D}_{\text{cal}}, \mathcal{S}_{\text{cal}})] = \frac{k_\alpha}{|\mathcal{D}_{\text{cal}}| + 1} \geq 1 - \alpha. \tag{69}$$

$\square$

### E.2. Proofs of asymptotic conditional coverage

#### E.2.1. L-CP

*Proposition* 2. Assuming $|D_{\text{cal}}| \to \infty$ and $\hat{Q}(Z; X) \stackrel{\text{d.}}{=} Y|X$, L-CP achieves conditional coverage.

*Proof.* We first show that the conditional coverage of L-CP is equal to the CDF of the random variable $d_{\mathcal{Z}}(Z)$ in $\hat{q}$, i.e.,

$F_{d_{\mathcal{Z}}(Z)}(\hat{q})$. Given $x \in \mathcal{X}$, we have:

$$\mathbb{P}(Y \in \hat{R}_{\text{L-CP}}(X) \mid X = x) \tag{70}$$

$$=\mathbb{P}(Y \in \{\hat{Q}(z;x) : z \in R_{\mathcal{Z}}(\hat{q})\} \mid X = x) \tag{71}$$

$$=\mathbb{P}(\hat{Q}^{-1}(Y;X) \in R_{\mathcal{Z}}(\hat{q}) \mid X = x) \qquad (\text{Invertibility of } \hat{Q}(\cdot;X)) \tag{72}$$

$$=\mathbb{P}(Z \in R_{\mathcal{Z}}(\hat{q})) \qquad (\hat{Q}(Z;X) \overset{\text{d.}}{=} Y|X) \tag{73}$$

$$=\mathbb{P}(d_{\mathcal{Z}}(Z) \le \hat{q}) \tag{74}$$

$$=F_{d_{\mathcal{Z}}(Z)}(\hat{q}). \tag{75}$$

Marginalizing over $X$, we obtain that the marginal coverage is also equal to $F_{d_{\mathcal{Z}}(Z)}(\hat{q})$:

$$\mathbb{P}(Y \in \hat{R}_{\text{L-CP}}(X)) \tag{76}$$

$$=\mathbb{E}_X\Big[\mathbb{P}(Y \in \hat{R}_{\text{L-CP}}(X) \mid X)\Big] \tag{77}$$

$$=\mathbb{E}_X\Big[F_{d_{\mathcal{Z}}(Z)}(\hat{q})\Big] \tag{78}$$

$$=F_{d_{\mathcal{Z}}(Z)}(\hat{q}) \tag{79}$$

In the limit of $|\mathcal{D}_{\text{cal}}| \to \infty$, thanks to the Glivenko-Cantelli theorem, $\mathbb{P}(Y \in \hat{R}_{\text{L-CP}}(X)) = 1 - \alpha$ and the quantile $\hat{q}$ obtained by SCP is thus $F_{d_{\mathcal{Z}}(Z)}^{-1}(1 - \alpha)$.

Finally, we obtain that the conditional coverage is equal to $1 - \alpha$:

$$\mathbb{P}(Y \in \hat{R}_{\text{L-CP}}(X) \mid X = x) \tag{80}$$

$$=F_{d_{\mathcal{Z}}(Z)}(F_{d_{\mathcal{Z}}(Z)}^{-1}(1 - \alpha)) \tag{81}$$

$$=1 - \alpha. \tag{82}$$

$\square$

### E.2.2. C-HDR AND C-PCP

**Lemma 2.** Assuming $|D_{\text{cal}}| \to \infty$, any conformal method with conformity score $s_{\text{CDF}}$ (12) achieves conditional coverage, independently from the conformity score $s_W$ of the base method. With the additional assumption that $K \to \infty$ and $\hat{f} = f$, $s_{\text{ECDF}}$ (13) achieves conditional coverage.

*Proof.* Let $W = s_W(X, Y)$ and consider $x \in \mathcal{X}$ and $y \in \mathcal{Y}$. By the probability integral transform, $s_{\text{CDF}}(x, Y) = F_{W|X=x}(W \mid X = x) \sim \mathcal{U}(0,1)$.

Marginalizing over $X$, we obtain:

$$\mathbb{P}(Y \in \hat{R}_{\text{CDF}}(X)) = \mathbb{P}(s_{\text{CDF}}(X, Y) \le \hat{q}) \tag{83}$$

$$= \mathbb{E}_X[\mathbb{P}(s_{\text{CDF}}(X, Y) \le \hat{q} \mid X)] \tag{84}$$

$$= \mathbb{E}_X[\mathbb{P}(U \le \hat{q})] \tag{85}$$

$$= \mathbb{E}_X[\hat{q}] \tag{86}$$

$$= \hat{q}, \tag{87}$$

where $U \sim \mathcal{U}(0,1)$. In the limit of $|\mathcal{D}_{\text{cal}}| \to \infty$, thanks to the Glivenko-Cantelli theorem, $\mathbb{P}(Y \in \hat{R}_{\text{CDF}}(X)) = 1 - \alpha$ and the quantile $\hat{q}$ obtained by SCP is thus $1 - \alpha$.

Finally, we note that:

$$\mathbb{P}(Y \in \hat{R}_{\text{CDF}}(X) \mid X = x) = \mathbb{P}(s_{\text{CDF}}(X, Y) \le \hat{q} \mid X = x) \tag{88}$$

$$= \mathbb{P}(U \le 1 - \alpha) \tag{89}$$

$$= 1 - \alpha. \tag{90}$$

Assuming $\hat{f} = f$, observe that, for any $x \in \mathcal{X}$ and $y \in \mathcal{Y}$, $s_{\text{ECDF}}(x, y) \to s_{\text{CDF}}(x, y)$ as $K \to \infty$ by the law of large numbers. Thus, under these conditions, any conformal method with conformity score $s_{\text{ECDF}}$ achieves conditional coverage.

$\square$

*Proposition* 3. Assuming $|D_{\text{cal}}| \to \infty$ and $K \to \infty$, both C-HDR and C-PCP with the oracle base predictor $\hat{f} = f$ achieve conditional coverage.

*Proof.* The proof is direct by Lemma 2 with $s_W(x, y) = s_{\text{DR-CP}}(x, y)$ for C-HDR and $s_W(x, y) = s_{\text{PCP}}(x, y)$ for C-PCP.

$\square$

### E.2.3. M-CP

Consider M-CP with exact quantile estimates $\hat{l}_i(x) = Q_{Y_i}(\alpha_l \mid x)$ and $\hat{u}_i(x) = Q_{Y_i}(\alpha_u \mid x)$ where $Q_{Y_i}(\alpha \mid x)$ is the quantile function of $Y_i$ conditional to $X = x$ evaluated in $\alpha$. This section introduces two propositions where M-CP requests two different nominal coverage levels $\alpha_u - \alpha_l$, namely $\sqrt[d]{1-\alpha}$ and $1-\alpha$. The propositions show that M-CP can achieve conditional coverage under two contrasting scenarios: independence or total dependence between the dimensions of the output.

*Proposition* 4. Assuming $Y_1, \ldots, Y_d$ are conditionally independent given $X$, M-CP achieves conditional coverage if $\alpha_u - \alpha_l = \sqrt[d]{1-\alpha}$.

*Proof.* For any $x \in \mathcal{X}$ and $i \in [d]$, we first establish that the $\sqrt[d]{1-\alpha}$th quantile of the distribution of $s_i(X, Y_i)$ given $X = x$ equals 0:

$$\mathbb{P}(s_i(X, Y_i) \leq 0 \mid X = x) = \mathbb{P}(\max\{l_i(X) - Y, Y - u_i(X)\} \leq 0 \mid X = x) \tag{91}$$
$$= \mathbb{P}(l_i(X) \leq Y \wedge Y \leq u_i(X) \mid X = x) \tag{92}$$
$$= 1 - \mathbb{P}(l_i(X) > Y \vee Y > u_i(X) \mid X = x) \tag{93}$$
$$= 1 - \mathbb{P}(l_i(X) > Y \mid X = x) - \mathbb{P}(Y > u_i(X) \mid X = x) \tag{94}$$
$$= 1 - \alpha_l - (1 - \alpha_u) \tag{95}$$
$$= \alpha_u - \alpha_l \tag{96}$$
$$= \sqrt[d]{1-\alpha}. \tag{97}$$

Using (97), we show that the $1 - \alpha$th quantile of the distribution of $s(X, Y)$ given $X = x$ is 0:

$$\mathbb{P}(s_{\text{M-CP}}(X, Y) \leq 0 \mid X = x) = \mathbb{P}(s_i(X, Y_i) \leq 0, \forall i \in [d] \mid X = x) \tag{98}$$
$$= \mathbb{P}(s_1(X, Y_1) \leq 0 \wedge \cdots \wedge s_d(X, Y_d) \leq 0 \mid X = x) \tag{99}$$
$$= \mathbb{P}(s_1(X, Y_1) \leq 0 \mid X = x) \ldots \mathbb{P}(s_d(X, Y_d) \leq 0 \mid X = x) \tag{100}$$
$$= \sqrt[d]{1-\alpha}^d \tag{101}$$
$$= 1 - \alpha, \tag{102}$$

where (100) is obtained by conditional independence of $Y_1, \ldots, Y_d$ given $X$. Marginalizing over $X$, we obtain that the $1 - \alpha$th quantile of $s(X, Y)$ is 0:

$$\mathbb{P}(s_{\text{M-CP}}(X, Y) \leq 0) = \mathbb{E}_X[\mathbb{P}(s_{\text{M-CP}}(X, Y) \leq 0 \mid X)] \tag{103}$$
$$= \mathbb{E}_X[1 - \alpha] \tag{104}$$
$$= 1 - \alpha. \tag{105}$$

In the limit of $|\mathcal{D}_{\text{cal}}| \to \infty$, thanks to the Glivenko-Cantelli theorem, $\mathbb{P}(Y \in \hat{R}_{\text{M-CP}}(X)) = 1 - \alpha$ and the quantile $\hat{q}$ obtained by SCP is thus 0.

Finally, using (102) and $\hat{q} = 0$, we obtain that M-CP achieves conditional coverage:

$$\mathbb{P}(Y \in \hat{R}_{\text{M-CP}}(X) \mid X = x) = \mathbb{P}(s_{\text{M-CP}}(X, Y) \leq \hat{q} \mid X = x) = 1 - \alpha. \tag{106}$$

$\square$

*Proposition* 5. Assuming $Y_1|X \overset{\text{a.s.}}{=} \ldots \overset{\text{a.s.}}{=} Y_d|X$, M-CP achieves conditional coverage if $\alpha_u - \alpha_l = 1 - \alpha$.

*Proof.* Let $x \in \mathcal{X}$. Using (96), we first show that the $1 - \alpha$th conditional quantile of the distribution of $s_i(X, Y_i)$, for any $i \in [d]$, is 0:

$$\mathbb{P}(s_i(X, Y_i) \leq 0 \mid X = x) = \alpha_u - \alpha_l \tag{107}$$
$$= 1 - \alpha. \tag{108}$$

Using (108), we show that the $1 - \alpha$th quantile of the distribution of $s(X, Y)$ given $X$ is 0:

$$\mathbb{P}(s(X, Y) \leq 0 \mid X = x) = \mathbb{P}(s_i(X, Y_i) \leq 0, \forall i \in [d] \mid X = x) \tag{109}$$
$$= \mathbb{P}(s_1(X, Y_1) \leq 0 \wedge \cdots \wedge s_d(X, Y_d) \leq 0 \mid X = x) \tag{110}$$
$$= \mathbb{P}(s_1(X, Y_1) \leq 0 \mid X = x) \tag{111}$$
$$= 1 - \alpha, \tag{112}$$

where (111) is due to $Y_1|X \overset{\text{a.s.}}{=} \ldots \overset{\text{a.s.}}{=} Y_d|X$, which implies that, conditional to $X = x$, $l_1(X) = \cdots = l_d(X)$ and $u_1(X) = \cdots = u_d(X)$ and thus $s_1(X, Y_1) = \cdots = s_d(X, Y_d)$. Using (105), we obtain that $\hat{q} = 0$, Finally, using (112), we obtain that M-CP achieves conditional coverage:

$$\mathbb{P}(Y \in \hat{R}(X) \mid X = x) = \mathbb{P}(s(X, Y) \leq 0 \mid X = x) = 1 - \alpha. \tag{113}$$

$\square$

### E.3. Connection between sample-based and density-based methods

This section proves the connections between sample-based and density-based methods as introduced in Section 5.3. We start by restating a known lemma of conformal prediction.

**Lemma 3.** Consider a conformal prediction method with conformity score $s$. If $g : \mathbb{R} \to \mathbb{R}$ is a strictly increasing function, then the method with conformity score $g \circ s$ will produce the same prediction regions.

*Proof.* For any $x \in \mathcal{X}$, consider the prediction region created with $s$ as in Section 2.1:

$$\hat{R}(x) = \left\{ y \in \mathcal{Y} : s(x, y) \leq \text{Quantile}\left(\{s_i\}_{i \in [|\mathcal{D}_{\text{cal}}|]} \cup \{\infty\}; k_\alpha\right) \right\}. \tag{114}$$

Since $g$ is strictly increasing,

$$\hat{R}(x) = \left\{ y \in \mathcal{Y} : g(s(x, y)) \leq g\left(\text{Quantile}\left(\{s_i\}_{i \in [|\mathcal{D}_{\text{cal}}|]} \cup \{\infty\}; k_\alpha\right)\right) \right\} \tag{115}$$
$$= \left\{ y \in \mathcal{Y} : g(s(x, y)) \leq \text{Quantile}\left(\{g(s_i)\}_{i \in [|\mathcal{D}_{\text{cal}}|]} \cup \{\infty\}; k_\alpha\right) \right\}. \tag{116}$$

Since (116) corresponds to the prediction region with conformity score $g \circ s$, this shows that the two methods create the same regions. $\square$

*Proposition* 1. PCP is equivalent to DR-CP with $\hat{f} = \hat{f}_{\text{max}}$. Similarly, HD-PCP is equivalent to DR-CP with $\hat{f} = \hat{f}_{\text{max}}$ where only $\lfloor (1 - \alpha)L \rfloor$ samples with the highest density among $\{\tilde{Y}^{(l)}\}_{l \in [L]}$ are kept. Finally, C-PCP is equivalent to C-HDR with $\hat{f} = \hat{f}_{\text{max}}$.

*Proof.* In the following proof, we note $a \uparrow b$ to signify that there exists a strictly increasing function $g$ such that $a = g(b)$. Consider DR-CP with $\hat{f} = \hat{f}_{\text{max}}$. We have:

$$s_{\text{DR-CP}}(x, y) = -\hat{f}_{\text{max}}(y \mid x) \tag{117}$$
$$\uparrow -\max_{l \in [L]} f_{\mathbb{S}}(y; \tilde{Y}^{(l)}) \qquad \left(\hat{f}_{\text{max}}(y \mid x) = \max_{l \in [L]} f_{\mathbb{S}}(y; \tilde{Y}^{(l)})/C\right) \tag{118}$$
$$= \min_{l \in [L]} -f_{\mathbb{S}}(y; \tilde{Y}^{(l)}) \tag{119}$$
$$\uparrow \min_{l \in [L]} \|y - \tilde{Y}^{(l)}\|_2 \qquad \left(f_{\mathbb{S}}(y; \tilde{Y}^{(l)}) \text{ has spherical level sets centered at } \tilde{Y}^{(l)}\right) \tag{120}$$
$$= s_{\text{PCP}}(x, y). \tag{121}$$

We obtain the equivalence between the two methods by Lemma 3. The proof for HD-PCP follows the same arguments.

We now consider C-HDR with $\hat{f} = \hat{f}_{\max}$. We have:

$$s_{\text{C-HDR}}(x, y) = \frac{1}{K} \sum_{k \in [K]} \mathbb{1}(\hat{f}_{\max}(\hat{Y}^{(k)} \mid x) \geq \hat{f}_{\max}(y \mid x)) \quad \text{where } \hat{Y}^{(k)} \sim \hat{f}_{Y \mid X=x}, k \in [K]. \tag{122}$$

Developing the inequality for $k \in [K]$, we obtain:

$$\hat{f}_{\max}(\hat{Y}^{(k)} \mid x) \geq \hat{f}_{\max}(y \mid x) \tag{123}$$

$$\iff \max_{l \in [L]} f_{\mathbb{S}}(\hat{Y}^{(k)}; \hat{Y}^{(l)}) \geq \max_{l \in [L]} f_{\mathbb{S}}(y; \hat{Y}^{(l)}) \qquad \left( \hat{f}_{\max}(y \mid x) = \max_{l \in [L]} f_{\mathbb{S}}(y; \tilde{Y}^{(l)})/C \right) \tag{124}$$

$$\iff \min_{l \in [L]} -f_{\mathbb{S}}(\hat{Y}^{(k)}; \hat{Y}^{(l)}) \leq \min_{l \in [L]} -f_{\mathbb{S}}(y; \hat{Y}^{(l)}) \tag{125}$$

$$\iff \min_{l \in [L]} \|\hat{Y}^{(k)} - \tilde{Y}^{(l)}\|_2 \leq \min_{l \in [L]} \|y - \tilde{Y}^{(l)}\|_2. \qquad \left( f_{\mathbb{S}}(y; \tilde{Y}^{(l)}) \text{ has spherical level sets centered at } \tilde{Y}^{(l)} \right) \tag{126}$$

$$\tag{127}$$

Noting that (122) with (127) corresponds to the conformity score of C-PCP, we obtain the equivalence. $\square$

# F. Experimental setup

This section describes our experimental setup in more details. Computations were performed based on 2 workstations, one with with 2 A6000 GPUs and 64 CPU threads, and one with 2 A5000 GPUs and 64 CPU threads, running for 48 hours.

## F.1. Datasets

We consider a total of 13 datasets that have been used in previous studies. Since our focus is on multivariate prediction regions, we select only datasets with an output that is at least two-dimensional. Specifically, we include 6 datasets from Feldman et al., 2023, 4 datasets from Tsoumakas et al., 2011 (MULAN benchmark), 1 dataset from Wang et al., 2023b, 1 datasets from Barrio et al., 2024, and 1 dataset from Camehl et al., 2024.

Each dataset is split into training, validation, calibration, and test sets with 2048 points reserved for calibration. The remaining data is split into 55% for training, 15% for validation and 30% for testing. The preprocessing follows the setup described in Grinsztajn et al., 2022. Table 3 provides the detailed characteristics of each dataset.

Table 3: Characteristics of each dataset considered in this study.

| Source | Dataset | Nb instances | Nb features $p$ | Nb targets $d$ |
|---|---|---|---|---|
| Camehl | households | 7207 | 4 | 4 |
| Mulan | scm20d | 8966 | 60 | 16 |
| | rf1 | 9005 | 64 | 8 |
| | rf2 | 9005 | 64 | 8 |
| | scm1d | 9803 | 279 | 16 |
| Feldman | meps_21 | 15656 | 137 | 2 |
| | meps_19 | 15785 | 137 | 2 |
| | meps_20 | 17541 | 137 | 2 |
| | house | 21613 | 14 | 2 |
| | bio | 45730 | 8 | 2 |
| | blog_data | 50000 | 55 | 2 |
| Del Barrio | calcofi | 50000 | 1 | 2 |
| Wang | taxi | 50000 | 4 | 2 |

## F.2. Base predictors

We consider multiple base predictors and focus on MQF$^2$ for our main experiments (Section 6).

**MQF$^2$.** The Multivariate Quantile Function Forecaster (MQF$^2$, Kan et al., 2022) is a normalizing flow that is directly compatible with most of the methods presented since it is invertible, has an explicit PDF, and can be sampled from. `M-CP`, `CopulaCPTS` and `STDQR` require small adaptations from the original methods, as discussed below. The quantile function $\hat{Q}$ and distribution function $\hat{Q}^{-1}$ of MQF$^2$ exhibit cyclical monotonicity, meaning they are the gradient of a convex function (Hallin et al., 2021).

The main idea behind MQF$^2$ is to interpret Convex Potential Flows (Huang et al., 2020) as multivariate (vector) quantile functions, in the sense that the representation property (128) and cyclical monotonicity property (129) are satisfied (Carlier et al., 2016):

$$Y = \hat{Q}(Z; x) \qquad \forall x \in \mathcal{X} \text{ where } Z \sim \mathcal{U}(0,1)^d, \tag{128}$$

$$\left(\hat{Q}(z_1; x) - \hat{Q}(z_2; x)\right)^T (z_1 - z_2) \geq 0 \qquad \forall x \in \mathcal{X}, z_1, z_2 \in \mathcal{Z}. \tag{129}$$

When $d = 1$, this reduces to the classical univariate quantile function. In practice, we follow Kan et al., 2022 and use a quantile vector that follows a normal distribution $Z \sim \mathcal{N}(0, I)$, allowing better training.

The underlying model of MQF$^2$ is a partially input-convex neural network (PINN, Amos et al., 2017) with two hidden layers, each containing 30 units. Increasing the number of parameters did not significantly improve performance, which is partly due to the efficiency of Convex Potential Flows compared to other normalizing flows (Huang et al., 2020). While hyperparameter tuning for each dataset could enhance performance, it is not the primary focus of this paper.

MQF$^2$ is trained using maximum likelihood estimation with early stopping, with a patience of 15 epochs, where validation loss is measured every two epochs.

**Distributional Random Forests.** Distributional Random Forest (Cevid et al., 2022) is a model built upon the Random Forest algorithm, which adaptively identifies the relevant training data points for any given test point. More specifically, given a test point $x \in \mathcal{X}$, Distributional Random Forest outputs a weight $w(x^{(i)} \mid x)$ for each training point $x^{(i)}$ with $x^{(i)} \in \mathcal{D}_{\text{train}}$. This approach enables accurate estimation of any quantity of interest conditional on $x \in \mathcal{X}$. In our experiments, we estimate the conditional distribution $Y|X$ as a Gaussian mixture, with each component centered on a training point and weighted by the Distributional Random Forest.

The PDF at $y \in \mathcal{Y}$ given $x \in \mathcal{X}$ is expressed as:

$$\hat{f}(y \mid x) = \sum_{i=1}^{|\mathcal{D}_{\text{train}}|} w(x^{(i)} \mid x) \cdot \mathcal{N}(y \mid y^{(i)}, \sigma I_d),$$

where $\sigma$ is tuned by minimizing the NLL on a grid search. For the Distributional Random Forest, the minimum node size is set to 15, the forest consists of 2000 trees, and the splitting criterion is the maximum mean discrepancy (MMD).

Since this method does not operate in a latent space, we do not consider `L-CP` in combination with this base predictor. CD diagrams for this predictor are presented in Appendix G.2.

**Multivariate Gaussian Mixture Model parameterized by a hypernetwork.** As another base predictor, we consider a multivariate Gaussian Mixture Model parameterized by a hypernetwork. The hypernetwork is a multilayer perceptron (MLP) that outputs the parameters of a mixture of $M$ multivariate Gaussian distributions. Given $x \in \mathcal{X}$, for each mixture component $m \in [M]$, the hypernetwork outputs the logit $z_m(x)$ (for the categorical distribution over the mixture components), the mean $\mu_m(x)$ (component location), and the lower triangular Cholesky factor $L_m(x)$ (representing the scale of the covariance matrix).

The mixture weights $\pi_m(x)$ are obtained by applying the softmax function to the logits $z_m(x)$, ensuring they sum to 1. The covariance matrices $\Sigma_m(x)$ for each component are constructed by taking the product $L_m(x)L_m(x)^\top$, guaranteeing that they are positive semi-definite.

The PDF evaluated in $y \in \mathcal{Y}$ conditional to $x \in \mathcal{X}$ is given by:

$$\hat{f}(y \mid x) = \sum_{m=1}^{M} \pi_m(x) \cdot \mathcal{N}(y \mid \mu_m(x), \Sigma_m(x)).$$

The model is trained using maximum likelihood estimation with $M = 10$.

Similarly to Distributional Random Forests, this method does not operate in a latent space, and thus we do not consider `L-CP`. CD diagrams for this predictor are presented in Appendix G.3.

### F.3. Adaptation of conformal methods into a common framework.

To ensure a fair comparison among conformal methods, we apply the calibration step using the same base predictors. Only `M-CP`, `CopulaCPTS`, and `STDQR` require slight modifications from their original formulations.

For `M-CP` and `CopulaCPTS`, direct estimation of marginal distributions for each output $Y_i, i \in [d]$ is infeasible with $\text{MQF}^2$. Instead, we estimate the lower and upper quantiles by first sampling $\{\hat{Y}^{(l)}\}_{l \in [L]}$ from $\hat{f}_{Y|X=x}$ given $x \in \mathcal{X}$, and then computing the empirical quantiles $\hat{Y}_i^{\left(\lfloor L\frac{\alpha}{2} \rfloor\right)}$ and $\hat{Y}_i^{\left(\lfloor L(1-\frac{\alpha}{2}) \rfloor\right)}$. Sampling time is not accounted in time computations for these methods. While a more computationally efficient base predictor could be used, this approach ensures a direct comparison with other conformal methods by maintaining consistency in the base predictor.

For `STDQR`, we modify the original method by replacing the conditional variational autoencoder (CVAE) with a normalizing flow. Following recommendations for future work from Feldman et al., 2023, we exploit the property that the output is normally distributed in the latent space and replace the base predictor by a normalizing flow. This adaptation leverages the assumption that the output is normally distributed in the latent space, allowing for an exact inverse transformation and eliminating a potential source of noise. To construct a region $R_{\mathcal{Z}}$ with coverage $1 - \alpha$ in the latent space, we select the $1 - \alpha$ proportion of samples closest to the origin, ensuring correct coverage without the need for directional quantile regression. The calibration procedure remains unchanged.

### F.4. Metrics

**Marginal coverage.**   Marginal coverage is measured using

$$\text{MC} = \frac{1}{|\mathcal{D}_{\text{test}}|} \sum_{(x,y) \in \mathcal{D}_{\text{test}}} \mathbb{1}(y \in \hat{R}(x)).$$

**Region size.**   We report the mean region size

$$\text{Mean Size} = \frac{1}{|\mathcal{D}_{\text{test}}|} \sum_{(x,y) \in \mathcal{D}_{\text{test}}} |\hat{R}(x)|.$$

To avoid large regions disproportionately affecting the result, we also report the median of the region sizes

$$\text{Median Size} = \text{Quantile}(\{|\hat{R}(x)|\}_{(x,y) \in \mathcal{D}_{\text{test}}}; 0.5)$$

Computing the size of the region is challenging in high dimensions. Hence, we propose an unbiased estimator of the region size using importance sampling:

$$|\hat{R}(x)| = \int_{\mathcal{Y}} \mathbb{1}(y \in \hat{R}(x)) dy = \mathbb{E}_{\hat{Y} \sim \hat{f}(x)}\left[\frac{\mathbb{1}(\hat{Y} \in \hat{R}(x))}{\hat{f}(\hat{Y} \mid x)}\right] \approx \frac{1}{K} \sum_{k=1}^{K} \frac{\mathbb{1}(\hat{Y}^{(k)} \in \hat{R}(x))}{\hat{f}(\hat{Y}^{(k)} \mid x)}, \tag{130}$$

where $\hat{Y}^{(k)} \sim \hat{f}_{Y|X=x}, k \in [K]$. This estimator is compatible with all base predictors in Appendix F.2 since it is both possible to sample from their predictive distribution and evaluate the PDF. In Appendix F.7, we discuss the efficiency of this estimator.

**Conditional coverage.**   To ensure a robust evaluation of conditional coverage, we consider three different conditional coverage metrics, detailed in Appendix F.6. The Worst Slab Coverage (WSC, Cauchois et al., 2021) groups inputs into "slabs" and evaluates the worst obtained coverage. The coverage error conditional to $X$ (CEC-X) partitions the input space $\mathcal{X}$ and evaluates coverage on each subset. The coverage error conditional to $V = \hat{f}(\hat{Y} \mid X)$, where $\hat{Y} \sim \hat{f}_{Y|X}$, (CEC-V, Izbicki et al., 2022; Dheur et al., 2024), creates a partition based on the distribution of $V$, which is more robust to high-dimensional inputs.

**Computing time.** We report the total time required for calibration and testing the marginal coverage. Specifically, this requires evaluating conformity scores on $\mathcal{D}_{\text{cal}}$ followed by evaluating conformity scores on $\mathcal{D}_{\text{test}}$.

## F.5. Multi-Model, Multi-Dataset Comparison

In order to determine whether there are significant differences in model performance, we first apply the Friedman test (Friedman, 1940). Following the recommendations of Benavoli et al. (2016), we then conduct a pairwise post-hoc analysis using the Wilcoxon signed-rank test (Wilcoxon, 1945), coupled with Holm's alpha correction (Holm, 1979) to adjust for multiple comparisons.

The results are visualized using critical difference (CD) diagrams (Demšar, 2006). In these diagrams, models are ranked, with a lower rank (positioned further to the right) indicating better performance. A thick horizontal line connects models whose performances are not statistically different at the 0.05 significance level.

For MC and WSC, the CD diagrams report $|\text{MC} - (1 - \alpha)|$ and $|\text{WSC} - (1 - \alpha)|$, both of which should be minimized.

## F.6. Metrics of Conditional Coverage

**Worst Slab Coverage.** Introduced in Cauchois et al., 2021, the *Worst Slab Coverage* (WSC) metric quantifies the minimal coverage over all possible slabs in $\mathbb{R}^d$, where each slab contains at least a fraction $\delta$ of the total mass, with $0 < \delta \leq 1$. For a given vector $v \in \mathbb{R}^d$, the WSC associated with $v$, denoted as $\text{WSC}_v$, is defined by:

$$\text{WSC}_v = \inf_{a < b} \left\{ \hat{\mathbb{P}}_{\mathcal{D}_{\text{test}}} \left( y_i \in \hat{R}(x_i) \mid a \leq v^{\intercal} x_i \leq b \right) \text{ s.t. } \hat{\mathbb{P}}_{\mathcal{D}_{\text{test}}}(a \leq v^{\intercal} x_i \leq b) \geq \delta \right\}, \tag{131}$$

where $a, b \in \mathbb{R}$. This metric assesses conditional coverage by focusing on inputs $x_i$ that lie within a slab defined by $v$, using the inner product $v^{\intercal} x_i$ to measure similarity.

To estimate the worst-case slab, we follow the method from Cauchois et al., 2021, uniformly sampling 1,000 vectors $v_j$ from the unit sphere $\mathbb{S}^{d-1}$ and calculating:

$$\text{WSC} = \min_{v_j \in \mathbb{S}^{d-1}} \text{WSC}_{v_j}. \tag{132}$$

To mitigate overfitting on the test dataset, we partition the test set into two subsets, $\mathcal{D}_{\text{test}} = \mathcal{D}_{\text{test}}^{(1)} \cup \mathcal{D}_{\text{test}}^{(2)}$, as in Romano et al., 2020; Sesia and Romano, 2021. We identify the worst combination of $a$, $b$, and $v$ on $\mathcal{D}_{\text{test}}^{(1)}$ by minimizing the WSC metric with $\delta = 0.2$, and then evaluate conditional coverage on the separate subset $\mathcal{D}_{\text{test}}^{(2)}$.

**CEC-X.** CEC-X approximates conditional coverage by partitioning the input space $X \in \mathcal{X} \subseteq \mathbb{R}^p$. We apply the $k$-means++ clustering algorithm on the inputs $X^{(i)}$ in the validation dataset $\mathcal{D}_{\text{val}}$, creating a partition $\mathcal{A} = A_1 \cup \cdots \cup A_J$ over $\mathcal{X}$. The *Coverage Error Conditional to X* is defined as:

$$\text{CEC-X} = \frac{1}{|\mathcal{D}_{\text{test}}|} \sum_{i=1}^{|\mathcal{D}_{\text{test}}|} \sum_{j=1}^{J} \left( \hat{\mathbb{P}}_{\mathcal{D}_{\text{test}}} \left( y^{(i)} \in \hat{R}(x^{(i)}) \mid x^{(i)} \in A_j \right) - (1 - \alpha) \right)^2. \tag{133}$$

**CEC-V.** CEC-V is similar to CEC-X, but the conditioning is on the distribution of $\log V = \log \hat{f}(\hat{Y} \mid X)$, where $\hat{Y} \sim \hat{f}_{Y|X}$. Unlike CEC-X, CEC-V is more robust to high-dimensional inputs. This approach originates from the CD-split$^+$ method (Izbicki et al., 2022) and has been adapted to multivariate outputs in Dheur et al., 2024.

In practice, given an input $x$, a new feature $v_x$ is created. First, samples $v_i$ from $V \mid X = x$ are generated by sampling $y_1, \ldots, y_m \sim \hat{f}_{Y|X=x}$ and evaluating $v_i = \hat{f}(y_i \mid x)$. The resulting vector $v_x = (v_{(1)}, \ldots, v_{(m)})$ consists of the order statistics $v_{(i)}$ from $v_1, \ldots, v_m$.

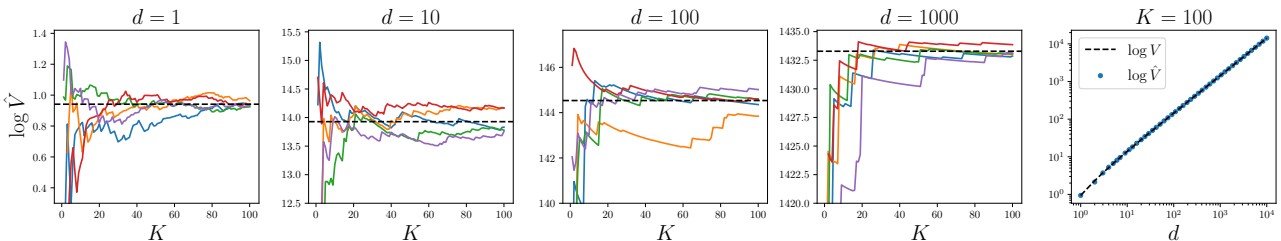

Figure 11: Panels 1 to 4: Trajectories of the log volume estimator with increasing $K$ compared to the true log volume (dashed line) for different output dimensions $d$. Panel 5: Log volume estimator with $K = 100$ compared to the true log volume (dashed line).

The $k$-means++ clustering algorithm is applied on the vectors $\log v_{X^{(i)}}$ in the validation dataset $\mathcal{D}_{\text{val}}$, and a partition $\mathcal{A}_V = A_1 \cup \cdots \cup A_J$ over $\mathbb{R}^m$ is obtained. The *Coverage Error Conditional to the distribution of $V$* is then computed according to (133), using the partition $\mathcal{A}_V$.

Dheur et al., 2024 notes that the distance function corresponding to this partitioning approach is the 2-Wasserstein distance with respect to the distribution of $V$.

### F.7. Estimator for the region size

In this section, we discuss the efficiency of the region size estimator introduced in Appendix F.4. This estimator is based on a density estimator $\hat{f}_{Y|X=x}$ and a sample $\hat{Y}^{(k)}, k \in [K]$, drawn i.i.d. from the conditional distribution $Y \mid X = x$ for any $x \in \mathcal{X}$. Specifically, the estimator is given by:

$$\hat{V}(x) = \frac{1}{K} \sum_{k=1}^{K} \frac{\mathbb{1}(\hat{Y}^{(k)} \in \hat{R}(x))}{\hat{f}(\hat{Y}^{(k)} \mid x)}.$$

While the estimator is unbiased, i.e., $\mathbb{E}[\hat{V}(x)] = |\hat{R}(x)|$, we want to study its variance. Let $I = \mathbb{1}(\hat{Y} \in \hat{R}(x))$ represent the indicator that a sample $\hat{Y}$ lies within the prediction region $\hat{R}(x)$, and let $\rho = \mathbb{P}(\hat{Y} \in \hat{R}(x))$ denote the coverage probability obtained from the samples based on our density estimator. Using the law of total variance, we obtain the following expression for the variance of $\hat{V}(x)$:

$$\mathbb{V}\left[\hat{V}(x)\right] = \frac{1}{K}\mathbb{V}\left[\frac{I}{\hat{f}(\hat{Y} \mid x)}\right]$$

$$= \frac{1}{K}\left(\mathbb{E}\left[\mathbb{V}\left[\frac{I}{\hat{f}(\hat{Y} \mid x)} \,\middle|\, I\right]\right] + \mathbb{V}\left[\mathbb{E}\left[\frac{I}{\hat{f}(\hat{Y} \mid x)} \,\middle|\, I\right]\right]\right)$$

$$= \frac{1}{K}\left(\rho\mathbb{V}\left[\frac{1}{\hat{f}(\hat{Y} \mid x)}\right] + \rho(1-\rho)\mathbb{E}\left[\frac{1}{\hat{f}(\hat{Y} \mid x)}\right]^2\right).$$

Assuming that the density estimate corresponds to the true density, i.e. $\hat{f}_{Y|X=x} = f_{Y|x}(\cdot \mid x)$ and that $\hat{R}$ achieves conditional coverage, then $\rho = 1 - \alpha$, and we obtain:

$$\mathbb{V}\left[\hat{V}(x)\right] = \frac{1}{K}\left((1-\alpha)\mathbb{V}\left[\frac{1}{f_{Y|x}(Y \mid x)}\right] + \alpha(1-\alpha)\mathbb{E}\left[\frac{1}{f_{Y|x}(Y \mid x)}\right]^2\right).$$

This indicates that the variance of our estimator only depends on the variance and expectation of the random variable $\frac{1}{f(Y|x)}$. In this case, the variance does not directly depend on the output dimension $d$.

Figure 11 shows how the estimator behaves in a scenario with a specific density estimator and prediction region with varying output dimension $d$ and an 80% coverage level. Since there is no dependence on $X$, we abbreviate the notation as follows:

$\hat{R} = \hat{R}(x)$, $\hat{f}(y) = \hat{f}(y \mid x)$, and $\hat{V} = \hat{V}(x)$ for any $x \in \mathcal{X}$. The density estimator is a standard normal distribution $\hat{f}(y) = \mathcal{N}(y; 0, I_d)$ and the prediction region is a ball $\hat{R} = \left\{ y \in \mathcal{Y} : \|y\|_2 \leq F_{\chi_d^2}^{-1}(1 - \alpha) \right\}$, where $\chi_d^2$ is the chi-squared distribution with $d$ degrees of freedom and $F_{\chi_d^2}^{-1}$ is its quantile function. It can be shown that $\mathbb{P}_{\hat{Y} \sim \hat{f}(\cdot)}(\hat{Y} \in \hat{R}) = 1 - \alpha$. In this case, the volume $V$ of $\hat{R}$ can be computed exactly.

Each of the first four panels in Figure 11 shows five trajectories for $\log \hat{V}$ as $K$ increases from 1 to 100. The true volume, $\log V$, of the prediction region is indicated by a dashed line. We observe that the estimator converges within a reasonable range of the true volume for varying output dimensions $d$. The last panel illustrates the value of $\log \hat{V}$ as a function of $d$, with $\log V$ again marked by a dashed line. From this, we observe that the estimator remains close to the true volume across different output dimensions $d$.

## G. Additional results

This section presents additional results for MQF$^2$ (Appendix G.1), Distributional Random Forests (Appendix G.2) and the Multivariate Gaussian Mixture Model (Appendix G.3). The experimental setup is described in Appendix F.

### G.1. MQF²

Figure 12 presents the marginal coverage and median region size across datasets of increasing size for MQF$^2$. In Panel 1, all methods except CopulaCPTS attain precise marginal coverage. This is expected since these methods follow the SCP algorithm (Section 2.1) and their marginal coverage conditional on the calibration dataset and samples from the calibration dataset follows a Beta distribution whose parameters only depend on the size of the calibration dataset (Appendix E.1). While CopulaCPTS attains marginal coverage, the larger variance in its marginal coverage arises because it does not follow the SCP algorithm.

In Panel 2, the median region size is normalized between 0 and 1 for each dataset in order to facilitate comparison. We observe that C-HDR often obtains the smallest median region size. The performance of the other methods can vary highly across datasets for the median region size and is better visualized in a CD diagram (see Figure 13).

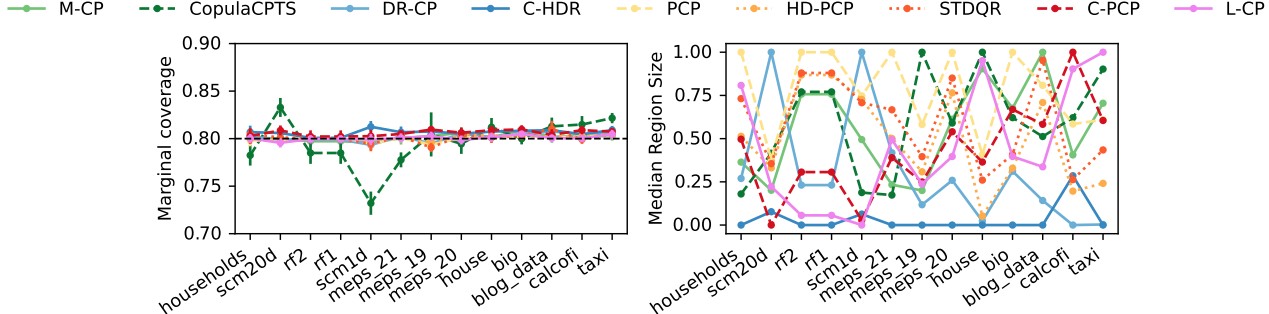

Figure 12: Marginal coverage and median region size with the base predictor MQF$^2$ across datasets sorted by size.

Figure 13 presents critical difference diagrams for three conditional coverage metrics (CEC-$X$, CEC-$Z$ and WSC), the mean region size and median region size, the total calibration and test time. Results are consistent with the results from the main text.

| Dataset | M-CP | CopulaCPTS | DR-CP | C-HDR | PCP | HD-PCP | STDQR | C-PCP | L-CP |
|---|---|---|---|---|---|---|---|---|---|
| households | $14.2_{0.48}$ | $12.3_{0.87}$ | $13.2_{0.29}$ | $\mathbf{10.6_{0.33}}$ | $20.5_{0.38}$ | $15.6_{0.39}$ | $17.8_{0.41}$ | $15.5_{0.74}$ | $18.6_{0.80}$ |
| scm20d | $67.6_{8.5}$ | $1.12\mathrm{e}{+}02_{2.1\mathrm{e}{+}01}$ | $2.33\mathrm{e}{+}02_{2.2\mathrm{e}{+}01}$ | $42.0_{7.9}$ | $1.05\mathrm{e}{+}02_{1.1\mathrm{e}{+}01}$ | $94.4_{9.4}$ | $99.4_{1.0\mathrm{e}{+}01}$ | $\mathbf{26.0_{3.1}}$ | $72.0_{1.0\mathrm{e}{+}01}$ |
| rf2 | $0.00547_{0.0027}$ | $0.00555_{0.0033}$ | $0.00215_{0.0010}$ | $\mathbf{0.000690_{0.00032}}$ | $0.00700_{0.0036}$ | $0.00617_{0.0030}$ | $0.00624_{0.0031}$ | $0.00262_{0.0012}$ | $\mathbf{0.00104_{0.00048}}$ |
| rf1 | $0.00547_{0.0027}$ | $0.00555_{0.0033}$ | $0.00215_{0.0010}$ | $\mathbf{0.000690_{0.00032}}$ | $0.00700_{0.0036}$ | $0.00617_{0.0030}$ | $0.00624_{0.0031}$ | $0.00262_{0.0012}$ | $\mathbf{0.00104_{0.00048}}$ |
| scm1d | $0.528_{0.046}$ | $0.323_{0.050}$ | $0.867_{0.078}$ | $0.239_{0.026}$ | $0.698_{0.065}$ | $0.684_{0.062}$ | $0.671_{0.069}$ | $\mathbf{0.216_{0.024}}$ | $\mathbf{0.197_{0.020}}$ |
| meps_21 | $0.185_{0.013}$ | $0.171_{0.014}$ | $0.227_{0.013}$ | $\mathbf{0.132_{0.024}}$ | $0.359_{0.021}$ | $0.246_{0.015}$ | $0.283_{0.015}$ | $0.220_{0.021}$ | $0.244_{0.052}$ |
| meps_19 | $0.214_{0.022}$ | $0.595_{0.42}$ | $0.175_{0.011}$ | $\mathbf{0.119_{0.019}}$ | $0.396_{0.059}$ | $0.266_{0.033}$ | $0.307_{0.043}$ | $0.238_{0.026}$ | $0.232_{0.043}$ |
| meps_20 | $0.371_{0.061}$ | $0.362_{0.059}$ | $0.223_{0.020}$ | $\mathbf{0.114_{0.012}}$ | $0.535_{0.050}$ | $0.436_{0.066}$ | $0.472_{0.052}$ | $0.341_{0.039}$ | $0.280_{0.028}$ |
| house | $1.17_{0.023}$ | $1.22_{0.043}$ | $\mathbf{0.664_{0.021}}$ | $\mathbf{0.651_{0.016}}$ | $0.882_{0.023}$ | $0.680_{0.018}$ | $0.799_{0.023}$ | $0.858_{0.018}$ | $1.19_{0.017}$ |
| bio | $0.303_{0.0066}$ | $0.296_{0.0092}$ | $0.257_{0.0067}$ | $\mathbf{0.218_{0.0053}}$ | $0.343_{0.0076}$ | $0.259_{0.0065}$ | $0.269_{0.0067}$ | $0.302_{0.0074}$ | $0.267_{0.0061}$ |
| blog_data | $0.170_{0.039}$ | $0.0948_{0.015}$ | $0.0374_{0.0056}$ | $\mathbf{0.0155_{0.0031}}$ | $0.141_{0.023}$ | $0.125_{0.023}$ | $0.163_{0.036}$ | $0.106_{0.021}$ | $0.0676_{0.017}$ |
| calcofi | $2.13_{0.024}$ | $2.38_{0.12}$ | $\mathbf{1.67_{0.022}}$ | $1.99_{0.026}$ | $2.33_{0.029}$ | $1.89_{0.029}$ | $1.97_{0.021}$ | $2.81_{0.042}$ | $2.70_{0.024}$ |
| taxi | $4.26_{0.068}$ | $4.72_{0.11}$ | $\mathbf{2.62_{0.029}}$ | $2.62_{0.033}$ | $4.03_{0.040}$ | $3.18_{0.030}$ | $3.63_{0.058}$ | $4.02_{0.064}$ | $4.94_{0.12}$ |

Table 4: Median region size with the base predictor $\mathrm{MQF}^2$.

| Dataset | M-CP | CopulaCPTS | DR-CP | C-HDR | PCP | HD-PCP | STDQR | C-PCP | L-CP |
|---|---|---|---|---|---|---|---|---|---|
| households | $36.9_{0.86}$ | $35.1_{1.4}$ | $\mathbf{15.7_{0.63}}$ | $40.1_{1.2}$ | $33.8_{2.1}$ | $30.1_{2.1}$ | $28.8_{2.1}$ | $62.6_{2.5}$ | $50.6_{1.4}$ |
| scm20d | $7.03\mathrm{e}{+}06_{2.5\mathrm{e}{+}06}$ | $3.82\mathrm{e}{+}07_{1.6\mathrm{e}{+}07}$ | $\mathbf{6.40\mathrm{e}{+}03_{1.9\mathrm{e}{+}03}}$ | $5.30\mathrm{e}{+}09_{2.1\mathrm{e}{+}09}$ | $1.61\mathrm{e}{+}04_{5.1\mathrm{e}{+}03}$ | $1.56\mathrm{e}{+}04_{5.1\mathrm{e}{+}03}$ | $1.59\mathrm{e}{+}04_{5.1\mathrm{e}{+}03}$ | $1.37\mathrm{e}{+}09_{1.0\mathrm{e}{+}09}$ | $2.20\mathrm{e}{+}10_{9.1\mathrm{e}{+}09}$ |
| rf1 | $1.86\mathrm{e}{+}02_{1.0\mathrm{e}{+}02}$ | $1.83\mathrm{e}{+}02_{9.6\mathrm{e}{+}01}$ | $\mathbf{15.1_{9.9}}$ | $3.50\mathrm{e}{+}02_{1.7\mathrm{e}{+}02}$ | $1.29\mathrm{e}{+}02_{8.1\mathrm{e}{+}01}$ | $1.09\mathrm{e}{+}02_{6.7\mathrm{e}{+}01}$ | $4.05\mathrm{e}{+}05_{2.1\mathrm{e}{+}05}$ | $9.85\mathrm{e}{+}02_{4.8\mathrm{e}{+}02}$ | $4.01\mathrm{e}{+}02_{1.9\mathrm{e}{+}02}$ |
| rf2 | $1.86\mathrm{e}{+}02_{1.0\mathrm{e}{+}02}$ | $1.83\mathrm{e}{+}02_{9.6\mathrm{e}{+}01}$ | $\mathbf{15.1_{9.9}}$ | $3.51\mathrm{e}{+}02_{1.7\mathrm{e}{+}02}$ | $1.29\mathrm{e}{+}02_{8.1\mathrm{e}{+}01}$ | $1.09\mathrm{e}{+}02_{6.7\mathrm{e}{+}01}$ | $4.05\mathrm{e}{+}05_{2.1\mathrm{e}{+}05}$ | $9.87\mathrm{e}{+}02_{4.8\mathrm{e}{+}02}$ | $4.02\mathrm{e}{+}02_{1.9\mathrm{e}{+}02}$ |
| scm1d | $2.37\mathrm{e}{+}05_{5.7\mathrm{e}{+}04}$ | $1.81\mathrm{e}{+}05_{5.1\mathrm{e}{+}04}$ | $78.4_{1.6\mathrm{e}{+}01}$ | $2.73\mathrm{e}{+}08_{5.3\mathrm{e}{+}07}$ | $\mathbf{57.3_{1.8\mathrm{e}{+}01}}$ | $\mathbf{43.0_{1.1\mathrm{e}{+}01}}$ | $\mathbf{43.6_{1.1\mathrm{e}{+}01}}$ | $1.48\mathrm{e}{+}08_{4.9\mathrm{e}{+}07}$ | $1.52\mathrm{e}{+}08_{2.6\mathrm{e}{+}07}$ |
| meps_21 | $1.21_{0.045}$ | $1.17_{0.046}$ | $\mathbf{0.315_{0.020}}$ | $1.44_{0.14}$ | $0.617_{0.029}$ | $0.558_{0.026}$ | $0.553_{0.025}$ | $2.07_{0.10}$ | $1.96_{0.34}$ |
| meps_19 | $1.14_{0.027}$ | $1.11_{0.031}$ | $\mathbf{0.293_{0.018}}$ | $1.29_{0.056}$ | $0.581_{0.027}$ | $0.559_{0.021}$ | $0.537_{0.021}$ | $1.96_{0.072}$ | $1.52_{0.053}$ |
| meps_20 | $1.20_{0.045}$ | $1.17_{0.038}$ | $\mathbf{0.309_{0.014}}$ | $1.30_{0.047}$ | $0.606_{0.020}$ | $0.562_{0.020}$ | $0.546_{0.018}$ | $2.01_{0.11}$ | $1.59_{0.064}$ |
| house | $1.83_{0.027}$ | $1.81_{0.038}$ | $\mathbf{0.887_{0.033}}$ | $1.09_{0.033}$ | $1.23_{0.034}$ | $0.964_{0.030}$ | $1.14_{0.030}$ | $1.44_{0.040}$ | $1.71_{0.013}$ |
| bio | $1.40_{0.39}$ | $1.42_{0.39}$ | $\mathbf{0.269_{0.010}}$ | $0.486_{0.014}$ | $0.396_{0.014}$ | $0.297_{0.0090}$ | $0.311_{0.010}$ | $1.41_{0.31}$ | $2.57_{0.77}$ |
| calcofi | $2.04_{0.023}$ | $2.22_{0.071}$ | $\mathbf{1.42_{0.018}}$ | $1.95_{0.040}$ | $2.22_{0.031}$ | $1.70_{0.022}$ | $1.78_{0.026}$ | $2.83_{0.036}$ | $2.34_{0.044}$ |
| blog_data | $0.390_{0.017}$ | $0.390_{0.016}$ | $\mathbf{0.0852_{0.0050}}$ | $0.375_{0.018}$ | $0.173_{0.0097}$ | $0.163_{0.0078}$ | $0.188_{0.0067}$ | $0.613_{0.028}$ | $0.520_{0.027}$ |
| taxi | $5.68_{0.084}$ | $6.35_{0.16}$ | $\mathbf{2.67_{0.047}}$ | $3.21_{0.049}$ | $4.55_{0.090}$ | $3.54_{0.075}$ | $3.99_{0.071}$ | $5.36_{0.090}$ | $6.51_{0.12}$ |

Table 5: Mean region size with the base predictor $\mathrm{MQF}^2$.

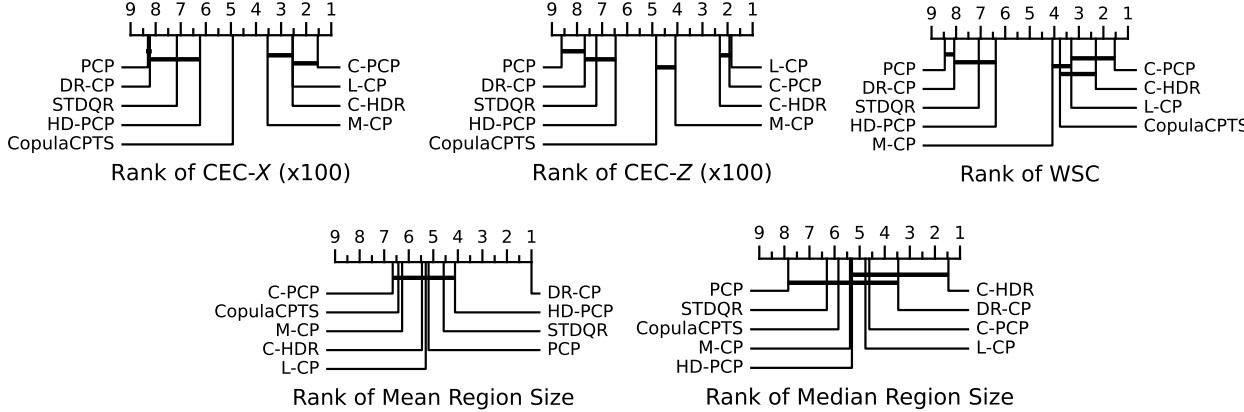

Figure 13: CD diagrams with the base predictor $\mathrm{MQF}^2$ with 10 runs per dataset and method.

### G.2. Distributional Random Forests

Figure 14 presents additional results for the base predictor Distributional Random Forests. Since this model does not rely on a latent space, results for STDQR and L-CP are not included.

In terms of conditional coverage, the results align with those of $\mathrm{MQF}^2$, with C-PCP and C-HDR outperforming DR-CP, PCP, and HD-PCP. Notably, M-CP achieves competitive conditional coverage, suggesting it pairs well with DRF-KDE. Similar to $\mathrm{MQF}^2$, all methods except for CopulaCPTS attain precise marginal coverage.

The median region size is normalized to a [0,1] range for each dataset to facilitate comparison. We observe that C-HDR generally achieves the smallest median region size, followed by DR-CP. The test time is the lowest for M-CP and CopulaCPTS

while `C-PCP` and `C-HDR` obtain the highest computation times.

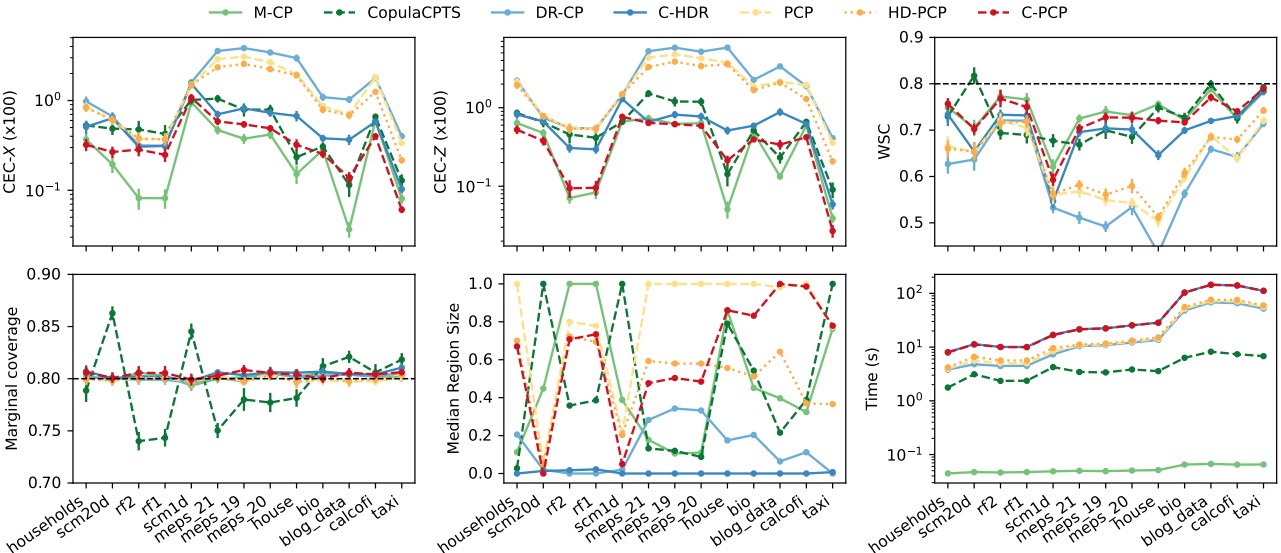

Figure 14: Conditional coverage metrics with the base predictor Distributional Random Forests across datasets sorted by size.

Figure 15 shows CD diagrams obtained with Distributional Random Forests as the base predictor. The results are consistent with Figure 14.

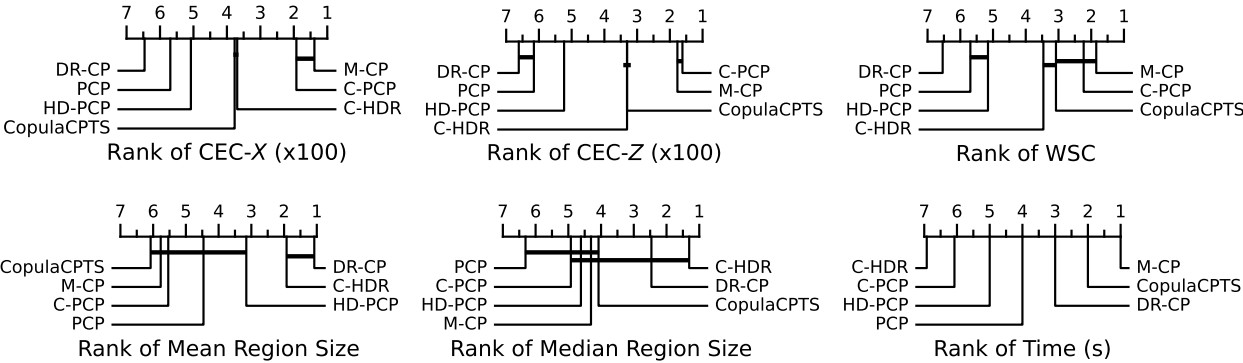

Figure 15: CD diagrams with the base predictor Distributional Random Forests based on 10 runs per dataset and method.

### G.3. Multivariate Gaussian Mixture Model

Figure 16 presents additional results for the base predictor Multivariate Gaussian Mixture Model. Similarly to Distributional Random Forests, this model does not rely on a latent space and thus results for `STDQR` and `L-CP` are not included.

The conditional coverage also aligns with $MQF^2$, `C-PCP` and `C-HDR` outperforming `DR-CP`, `PCP`, and `HD-PCP`. `M-CP` and `CopulaCPTS` achieving intermediate conditional coverage. As expected, marginal coverage is precise for all methods except CopulaCPTS.

`C-HDR` often obtains the smallest median region size, while `DR-CP` consistently attains the best mean region size.

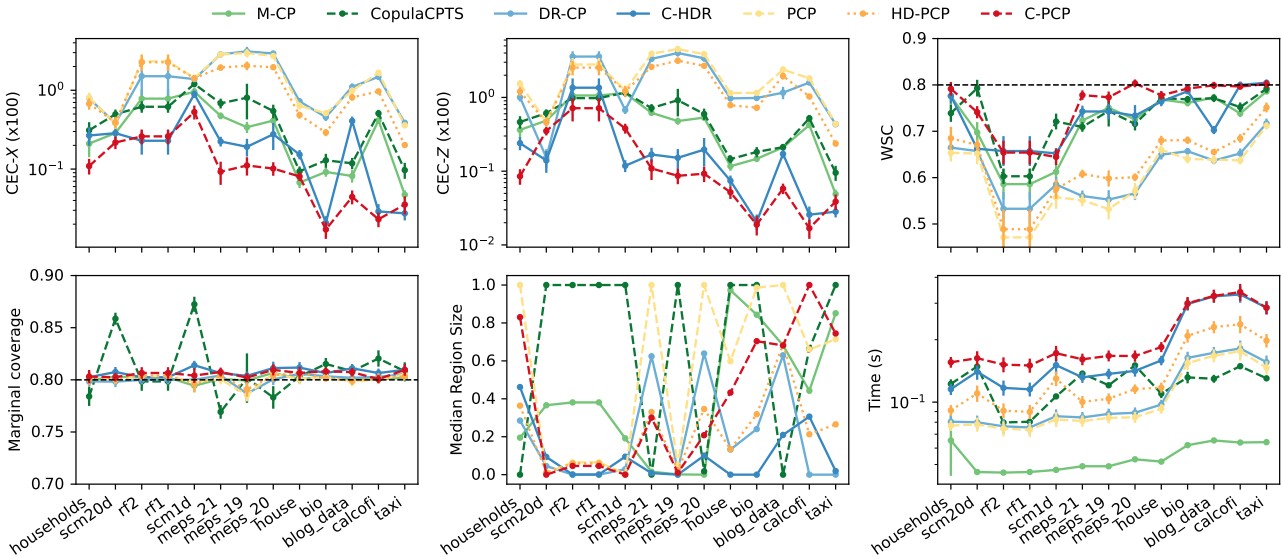

Figure 16: Conditional coverage metrics with the multivariate Gaussian mixture model base predictor across datasets sorted by size.

CD diagrams in Figure 17 are consistent with Figure 16.

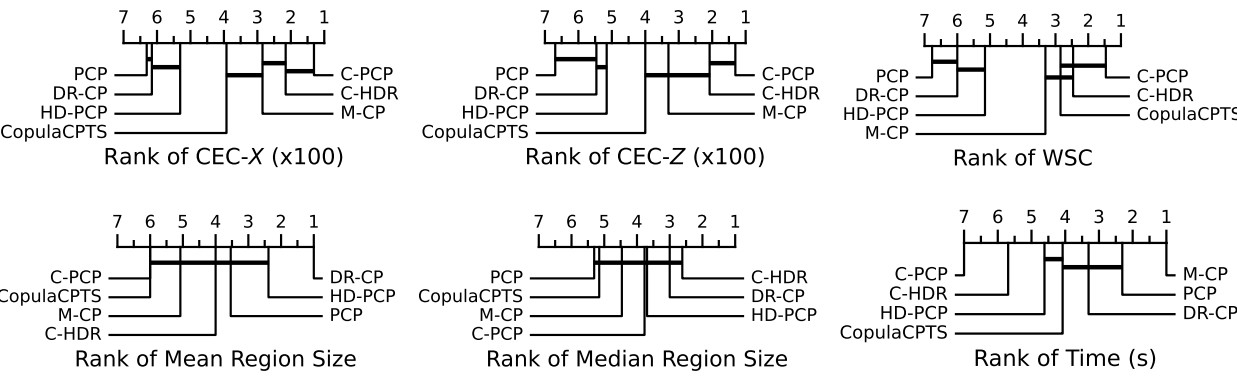

Figure 17: CD diagrams based on Multivariate Gaussian Mixture Model parameterized by a hypernetwork with $M = 10$ and 10 runs per dataset and method.

### G.4. Impact of the number of samples $K$

Figures 18 and 19 illustrate how conditional coverage, marginal coverage and region size change as a function of $K$ on all datasets. For a better comparison among datasets, the metrics CEC-$X$, CEC-$Z$, the median region size and the mean region size are normalized between 0 and 1, with results averaged over 10 runs. Furthermore, the red line indicates a linear regression fit, allowing to see the trend.

Conditional coverage metrics decreasing with $K$ indicate that conditional coverage tends to improve with an increasing number of samples. This is expected since an increasing number of Monte-Carlo samples allows a better estimation of the CDF of the scores in (13). Marginal validity is obtained with any $K$. However, small sizes of $K$ will lead to more duplicated conformity scores and thus a possibility of overcoverage. Median region sizes and mean region sizes also tend to decrease with $K$ as the CDF approximation improves.

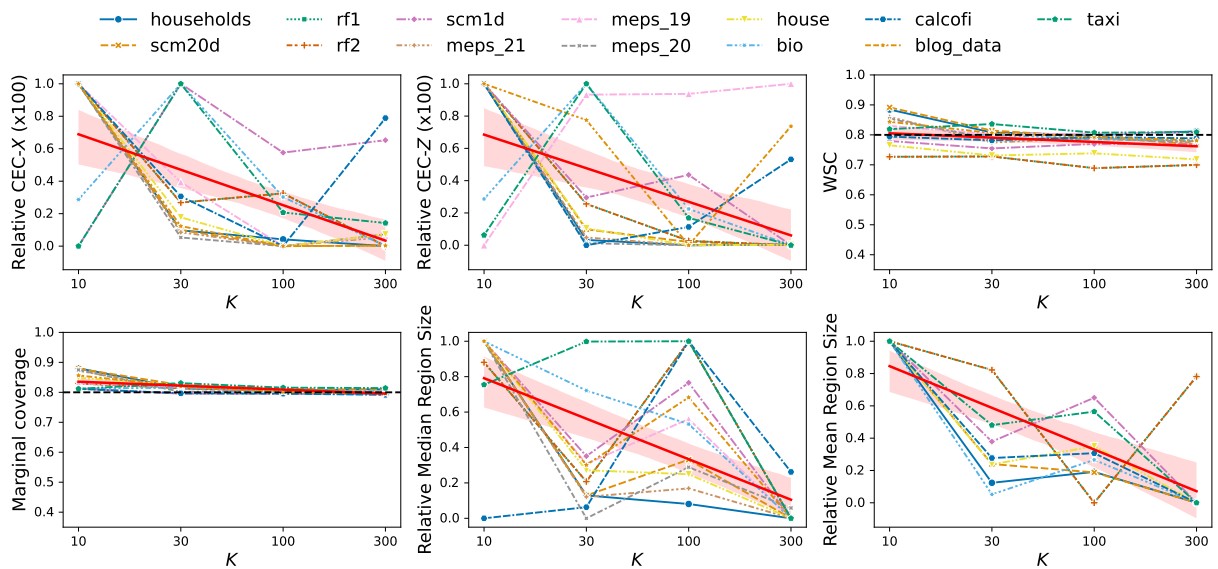

Figure 18: Evolution of conditional coverage, marginal coverage and region sizes of `C-PCP` as a function of the number of samples $K$ using the base predictor MQF$^2$. The metrics CEC-$X$, and CEC-$Z$ should be minimized, while the marginal coverage and WSC should approach $1 - \alpha$ (indicated by the dashed black line). The red line, obtained by linear regression, indicates the general trend.

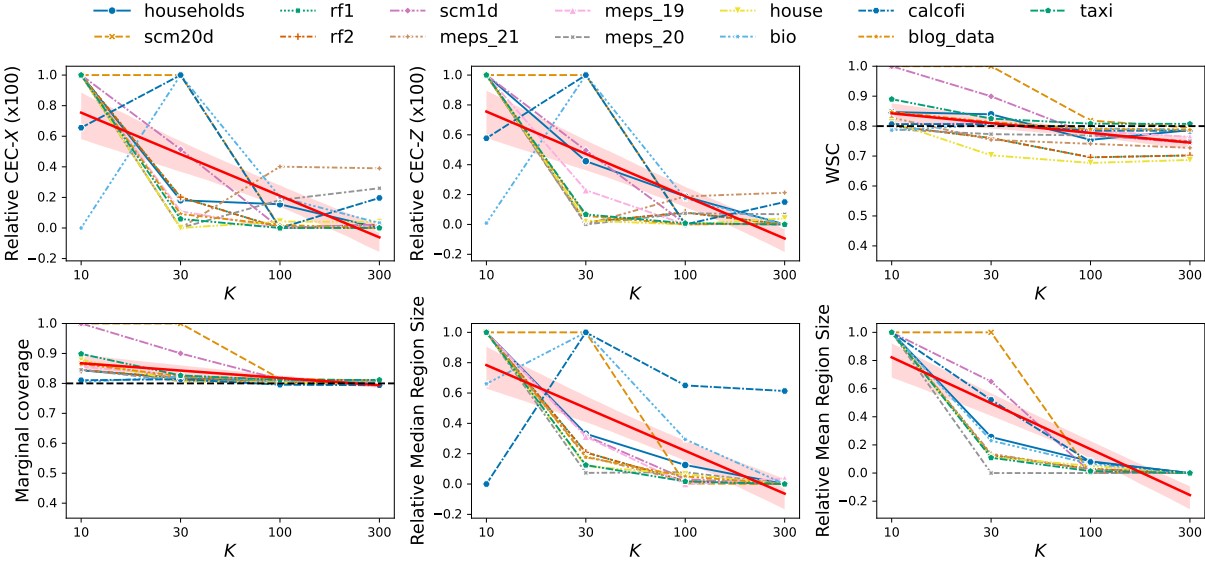

Figure 19: Reproduction of Figure 18 for `C-HDR`.

## H. Comparison with Bonferroni correction

To better understand the prediction regions produced by FWER control methods, we provide a qualitative and quantitative comparison with Bonferroni correction. We consider Bonferroni correction applied to the scores of CQR (see (16)), similarly to M-CP. Figure 20 provides an illustrative example, and Table 6 provides results on this same dataset. This shows that Bonferroni is computationally fast but produces larger regions due to the rectangular shape.

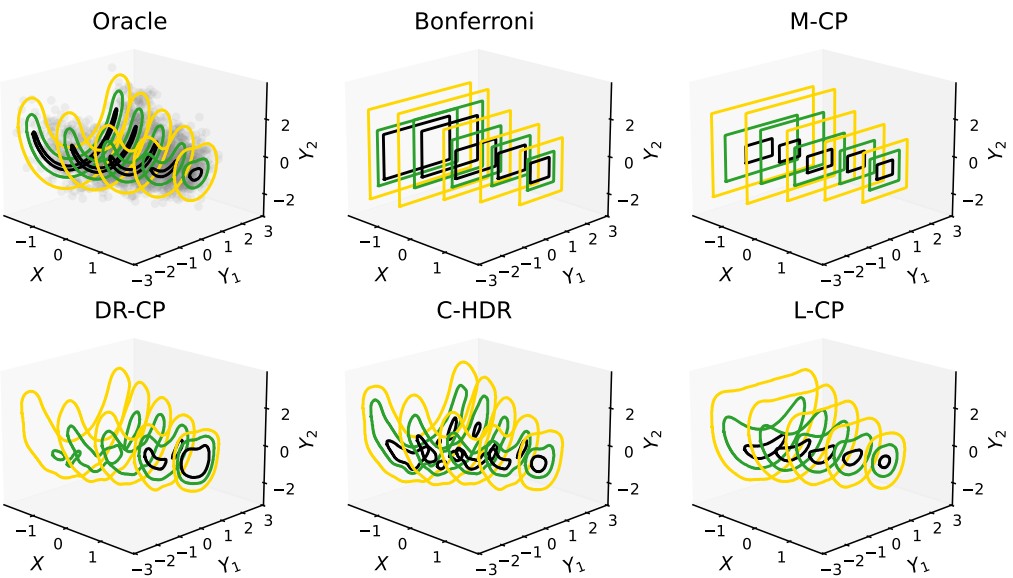

Figure 20: Prediction regions for a bivariate unimodal dataset, conditional on a unidimensional input. The black, green, and yellow contours represent regions with nominal coverage levels of 20%, 40%, and 80%, respectively. The figure is similar to Figure 2 in the main text, with Bonferroni added as a comparison. Both Bonferroni and M-CP are based on Conformal Quantile Regression (CQR) applied separately for each dimension.

Table 6: Detailed metrics for the unimodal heteroscedastic process from Figure 20, with $1 - \alpha$ fixed to 0.8.

| Method | MC | Median Size | CEC-$X$ ($\times 100$) | CEC-$Z$ ($\times 100$) | WSC | Test time |
|---|---|---|---|---|---|---|
| Bonferroni | $0.813_{0.0036}$ | $9.07_{0.15}$ | $0.0241_{0.012}$ | $0.0249_{0.0098}$ | $0.815_{0.0063}$ | $\mathbf{0.00339_{5.9e\text{-}05}}$ |
| M-CP | $\mathbf{0.801_{0.0037}}$ | $8.62_{0.074}$ | $0.0240_{0.0031}$ | $0.0157_{0.0031}$ | $\mathbf{0.796_{0.012}}$ | $0.0959_{0.058}$ |
| DR-CP | $\mathbf{0.796_{0.0019}}$ | $\mathbf{6.83_{0.042}}$ | $0.432_{0.019}$ | $0.403_{0.015}$ | $0.697_{0.0093}$ | $0.0557_{0.00075}$ |
| C-HDR | $0.809_{0.0025}$ | $6.97_{0.039}$ | $0.0129_{0.0059}$ | $0.0155_{0.0037}$ | $0.815_{0.0030}$ | $14.2_{0.11}$ |
| L-CP | $\mathbf{0.798_{0.0024}}$ | $8.06_{0.035}$ | $\mathbf{0.00586_{0.00095}}$ | $\mathbf{0.00549_{0.0014}}$ | $0.794_{0.0039}$ | $0.0584_{0.0012}$ |

## I. Comparison between C-PCP and CP²-PCP

In this section, we compare our proposed method, C-PCP, with the CP$^2$-PCP method recently proposed by Plassier et al., 2025. More generally, we also compare the methods from the CP$^2$ framework of Plassier et al., 2025 with our class of CDF-based conformity scores (Section 3.1 in the main text). In Appendix I.1, we present the more general CP$^2$ framework using our own notation for clarity, with CP$^2$-PCP as a particular case of CP$^2$. In Appendix I.2, we discuss the asymptotic properties of CP$^2$ and show the asymptotic equivalence with CDF-based methods. In Appendix I.3, we discuss the relationship between CDF-based and CP$^2$-based methods.

### I.1. The CP² framework

Let us define a family of non-decreasing nested regions $\{\mathcal{R}(x;t)\}_{t \in \mathbb{R}}$ such that $\bigcap_{t \in \mathbb{R}} \mathcal{R}(x;t) = \emptyset$ and $\bigcup_{t \in \mathbb{R}} \mathcal{R}(x;t) = \mathcal{Y}$, and $\bigcap_{t' < t} \mathcal{R}(x;t') = \mathcal{R}(x;t)$. Without loss of generality, these nested regions are expressed in terms of a conformity score $s_W(x, y) \in \mathbb{R}$ as follows:

$$\mathcal{R}(x;t) = \{y \in \mathcal{Y} : s_W(x, y) \leq t\}, \tag{134}$$

where $s_W(x, y)$ is continuous in $y$.

As the next step, we introduce a family of transformation functions $f_\tau(\lambda) : \mathbb{R} \to \mathbb{R}$ parameterized by $\tau \in \mathbb{R}$. It is assumed

that for any $\tau$, the function $\lambda \mapsto f_\tau(\lambda)$ is increasing and bijective. Let $\varphi \in \mathbb{R}$ be a constant (e.g., $\varphi = 1$). We also define the function $g_\varphi(\tau) = f_\tau(\varphi)$ and assume that $\tau \mapsto g_\varphi(\tau)$ is increasing and bijective.

As a first step towards defining CP$^2$, we construct a prediction region assuming knowledge of the conditional distribution $F_{Y|X}$. For a given input $x \in \mathcal{X}$, the prediction region is defined as:

$$\bar{R}_{\mathrm{CP}^2}(x) = \mathcal{R}(x; f_{\tau_x}(\varphi)), \tag{135}$$

where

$$\tau_x = \inf\left\{\tau : \mathbb{P}\left(Y \in \mathcal{R}(X; f_\tau(\varphi)) \mid X = x\right) \geq 1 - \alpha\right\} \tag{136}$$

implies that $\bar{R}_{\mathrm{CP}^2}(x)$ guarantees conditional coverage given $x$. Furthermore, using (134) and defining the random variable $W = s_W(X, Y)$, we can equivalently express (136) as

$$\tau_x = \inf\left\{\tau : \mathbb{P}\left(s_W(X, Y) \leq f_\tau(\varphi) \mid X = x\right) \geq 1 - \alpha\right\} \tag{137}$$

$$= \inf\left\{\tau : \mathbb{P}\left(g_\varphi^{-1}(s_W(X, Y)) \leq \tau \mid X = x\right) \geq 1 - \alpha\right\} \tag{138}$$

$$= Q_{g_\varphi^{-1}(W)|X=x}(1 - \alpha) \tag{139}$$

$$= g_\varphi^{-1}(Q_{W|X=x}(1 - \alpha)), \tag{140}$$

where we used that $g_\varphi$ is increasing and bijective, with $g_\varphi^{-1}(f_\tau(\varphi)) = \tau$. In other words, $\tau_x$ is the $1 - \alpha$ quantile of $g_\varphi^{-1}(W)$.

However, in practice, $\tau_x$ cannot be computed directly because the true conditional distribution $F_{Y|x}$ is unknown. Instead, it can be estimated using a sample $\hat{Y}^{(k)}, k \in [K]$, drawn from the estimated conditional distribution $\hat{F}_{Y|X=x}$. If $\hat{Q}_{W|X=x}(1 - \alpha)$ is the $1 - \alpha$ quantile of the empirical distribution $\frac{1}{K} \sum_{k \in [K]} \delta_{s_W(x, \hat{Y}^{(k)})}$, we can compute

$$\hat{\tau}_x = g_\varphi^{-1}(\hat{Q}_{W|X=x}(1 - \alpha)). \tag{141}$$

It should be noted that this estimated prediction region loses the exact conditional and marginal coverage properties due to the reliance on the estimated conditional distribution. The following shows how conformal prediction can restore some coverage properties.

From (134), using (135), we can write

$$\bar{R}_{\mathrm{CP}^2}(x) = \{y \in \mathcal{Y} : s_W(x, y) \leq f_{\tau_x}(\varphi)\} \tag{142}$$

$$= \left\{y \in \mathcal{Y} : f_{\tau_x}^{-1}(s_W(x, y)) \leq \varphi\right\}, \tag{143}$$

where we used the invertibility of $f_\tau$ for any $\tau \in \mathbb{R}$.

Based on (143), Plassier et al., 2025 defined the following conformity score:

$$s_{\mathrm{CP}^2}(x, y) = f_{\hat{\tau}_x}^{-1}(s_W(x, y)), \tag{144}$$

for which the corresponding prediction region $\hat{R}_{\mathrm{CP}^2}$ is given by

$$\hat{R}_{\mathrm{CP}^2}(x) = \{y \in \mathcal{Y} : s_{\mathrm{CP}^2}(x, y) \leq \hat{q}\}, \tag{145}$$

where we used (2) from the main text.

As an example, taking $f_\tau(\lambda) = \tau\lambda$ and $\varphi = 1$, the conformity score becomes:

$$s_{\mathrm{CP}^2}(x, y) = s_W(x, y)/\hat{\tau}_x, \tag{146}$$

where $\hat{\tau}_x$ is defined in (141). Finally, we obtain CP$^2$-PCP simply by replacing $s_W$ with $s_{\mathrm{PCP}}$ in (146).

## I.2. Asymptotic properties

I.2.1. ASYMPTOTIC EQUIVALENCE OF PREDICTION REGIONS

In the following, we prove that the prediction regions generated by $\mathrm{CP}^2$ (for any $f_\tau$ and $\varphi$) and CDF-based methods are identical in the oracle setting, asymptotically, as $|\mathcal{D}_{\mathrm{cal}}| \to \infty$. Specifically, for any $x \in \mathcal{X}$, both methods select the same threshold $t_{1-\alpha} = Q_W(1 - \alpha \mid X = x)$ for the prediction region $\mathcal{R}(x; t_{1-\alpha})$, which ensures a coverage level of $1 - \alpha$.

*Proposition* 6. Provided that the assumptions in Appendix I.1 hold, for any $x \in \mathcal{X}$, the prediction regions $\bar{R}_{\mathrm{CP}^2}(x)$ (for any choice of $f_\tau$ and $\varphi$) and $\hat{R}_{\mathrm{CDF}}(x)$ are equivalent.

*Proof.* Using the fact that $g_\varphi^{-1}(f_\tau(\varphi)) = \tau$ for any $\tau \in \mathbb{R}$ and that $g_\varphi$ is increasing and bijective, we can write:

$$\bar{R}_{\mathrm{CP}^2}(x) = \{y \in \mathcal{Y} : s_W(x, y) \leq f_{\tau_x}(\varphi)\} \tag{147}$$
$$= \{y \in \mathcal{Y} : g_\varphi^{-1}(s_W(x, y)) \leq \tau_x\} \tag{148}$$
$$= \{y \in \mathcal{Y} : g_\varphi^{-1}(s_W(x, y)) \leq g_\varphi^{-1}(Q_{W|X=x}(1 - \alpha))\} \tag{149}$$
$$= \{y \in \mathcal{Y} : s_W(x, y) \leq Q_{W|X=x}(1 - \alpha)\}. \tag{150}$$

Let $\bar{R}_{\mathrm{CDF}}(x)$ denote the prediction region obtained using the conformity score $s_{\mathrm{CDF}}$ as $|\mathcal{D}_{\mathrm{cal}}| \to \infty$. As shown in Section 3.1, $s_{\mathrm{CDF}}(X, Y) \sim \mathcal{U}(0, 1)$, which implies $\hat{q} = 1 - \alpha$. Therefore:

$$\bar{R}_{\mathrm{CDF}}(x) = \{y \in \mathcal{Y} : s_{\mathrm{CDF}}(x, y) \leq 1 - \alpha\} \tag{151}$$
$$= \{y \in \mathcal{Y} : F_{W|X=x}(s_W(x, y)) \leq 1 - \alpha\} \tag{152}$$
$$= \{y \in \mathcal{Y} : s_W(x, y) \leq Q_{W|X=x}(1 - \alpha)\}. \tag{153}$$

This shows that $\bar{R}_{\mathrm{CP}^2}(x) = \bar{R}_{\mathrm{CDF}}(x)$ and that the threshold $t_{1-\alpha} = Q_{W|X=x}(1 - \alpha)$ is identical for both methods.

$\square$

I.2.2. ASYMPTOTIC CONDITIONAL COVERAGE

*Proposition* 7. Provided that the assumptions in Section 5.2 of the main text hold, specifically that $\hat{F}_{Y|X=x} = F_{Y|x}$ for all $x \in \mathcal{X}$, and $|\mathcal{D}_{\mathrm{cal}}| \to \infty$, $\mathrm{CP}^2$ achieves ACC as $K \to \infty$.

*Proof.* Under these assumptions, we have $\hat{Q}_{W|X=x} = Q_{W|X=x}$, which implies $\hat{\tau}_x = \tau_x$ for all $x \in \mathcal{X}$. Hence, the prediction region for $\mathrm{CP}^2$ is given by:

$$\bar{R}_{\mathrm{CP}^2}(x) = \{y \in \mathcal{Y} : s_{\mathrm{CP}^2}(x, y) \leq \varphi\}.$$

Since this prediction region provides conditional coverage, it also ensures marginal coverage:

$$\mathbb{P}(Y \in \bar{R}_{\mathrm{CP}^2}(X)) = \mathbb{P}(s_{\mathrm{CP}^2}(X, Y) \leq \varphi) \tag{154}$$
$$= \mathbb{E}_X\left[\mathbb{P}(s_{\mathrm{CP}^2}(X, Y) \leq \varphi \mid X)\right] \tag{155}$$
$$= \mathbb{E}_X\left[1 - \alpha\right] \tag{156}$$
$$= 1 - \alpha. \tag{157}$$

Since $\hat{q}$ is the $1 - \alpha$ quantile of $s_{\mathrm{CP}^2}(X, Y)$, and as $|\mathcal{D}_{\mathrm{cal}}| \to \infty$, we have $\hat{q} = \varphi$ by definition. Therefore, since $\bar{R}_{\mathrm{CP}^2}(x)$ achieves conditional coverage (see (136)), the region $\hat{R}_{\mathrm{CP}^2}(x)$ also achieves ACC:

$$\mathbb{P}(Y \in \hat{R}_{\mathrm{CP}^2}(X) \mid X = x) = \mathbb{P}(s_{\mathrm{CP}^2}(X, Y) \leq \hat{q} \mid X = x) \tag{158}$$
$$= \mathbb{P}(s_{\mathrm{CP}^2}(X, Y) \leq \varphi \mid X = x) \tag{159}$$
$$\geq 1 - \alpha. \tag{160}$$

$\square$

### I.3. Relationship between CDF-based and CP²-based methods

A natural question is whether there exists $\{f_\tau\}_{\tau \in \mathbb{R}}$ and $\varphi \in \mathbb{R}$ (with the assumptions introduced in Appendix I.1) such that CDF-based and CP²-based methods produce the same regions. In the simple case where the distribution of the base conformity score is in a location family, Proposition 8 shows that the two methods are equivalent for a simple choice of $f_\tau$ and $\phi$. However, the proof is not easily generalizable to a location-scale family. Further development of existing classes of conformal methods with ACC and their intersections is a promising avenue for future research. Interestingly, we discuss below that answering this question would also draw links between established univariate conformal methods.

**Analogy to univariate conformal prediction.** To further clarify the distinction between CDF- and CP²-based methods, we can draw an analogy to the established univariate methods Dist-split (DS, Izbicki et al., 2020) and Conformalized Quantile Regression (CQR, Romano et al., 2019). Since CDF- and CP²-based methods calibrate one quantile intead of an interval, we only consider the right-tail version of DS and CQR:

- $s_{\text{ECDF}}$ is analogous to DS but operates in the space of conformity instead of the output space $\mathcal{Y}$. DS uses the estimated conditional CDF of the output variable, $s_{\text{DS}}(x, y) = \hat{F}_{Y|X=x}(y)$, transforming $y$ based on its rank.

- $s_{\text{CP²}}$ with difference adjustment is analogous to CQR, and also operates in the space of conformity instead of the output space $\mathcal{Y}$. Note that CP² with difference adjustment can be simplified to $s_{\text{CP²}}(x, y) = s_W(x, y) - \hat{Q}_{W|x}(1-\alpha)$. Similarly, CQR uses a score based on the difference from a single estimated conditional quantile, $s_{\text{CQR}}(x, y) = y - \hat{Q}_{Y|x}(1-\alpha)$.

Both CDF-based and CP²-based methods rely on a sample $\{\hat{Y}^{(k)}\}_{k=1}^K$ where $\hat{Y}^{(k)} \sim \hat{F}_{Y|X=x}$. The difference lies in the way they transform $s_W(x, y)$ to obtain ACC. Recall that the conformity scores $s_{\text{ECDF}}$ and $s_{\text{CP²}}$ are given by

$$s_{\text{ECDF}}(x, y) = \frac{1}{K} \sum_{k \in [K]} \mathbb{I}\left(s_W(x, \hat{Y}^{(k)}) \le s_W(x, y)\right) = \hat{F}_{W|X=x}(s_W(x, y)), \tag{161}$$

$$s_{\text{CP²}}(x, y) = f_{\hat{\tau}_x}^{-1}(s_W(x, y)) \text{ where } \hat{\tau}_x = g_\varphi^{-1}(\hat{Q}_{W|X=x}(1-\alpha)). \tag{162}$$

It is known that two conformal methods produce equal regions if and only if their conformity scores are equal after applying a strictly increasing function $\phi : \mathbb{R} \to \mathbb{R}$, i.e.:

$$s_{\text{ECDF}}(x, y) = \phi(s_{\text{CP²}}(x, y)) \quad \forall x \in \mathcal{X}, y \in \mathcal{Y}. \tag{163}$$

Given $x \in \mathcal{X}$, when $K$ is finite, the conformity score $s_{\text{ECDF}}(x, \cdot)$ is discontinuous and is thus necessarily different from the conformity score $s_{\text{CP²}}(x, \cdot)$, which is continuous. A more interesting setting is the case where $K \to \infty$ and $s_{\text{ECDF}}(x, \cdot)$ becomes continuous. We define the random variable $\hat{W} = s_W(X, \hat{Y})$, with $\hat{Y} \sim \hat{F}_{Y|X}$. Let $F_{\hat{W}|x}$ and $Q_{\hat{W}|x}$ denote the conditional CDF and QF of $\hat{W}$ given $X = x$. The conformity scores are defined as follows:

$$\bar{s}_{\text{ECDF}}(x, y) = F_{\hat{W}|x}(s_W(x, y)), \tag{164}$$

$$\bar{s}_{\text{CP²}}(x, y) = f_{\hat{\tau}_x}^{-1}(s_W(x, y)) \text{ where } \hat{\tau}_x = g_\varphi^{-1}(Q_{\hat{W}|x}(1-\alpha)). \tag{165}$$

Thus, we require that

$$f_{\tau_x}^{-1}(s_W(x, y)) = \phi(F_{\hat{W}|x}(s_W(x, y))) \quad \forall x \in \mathcal{X}, y \in \mathcal{Y} \tag{166}$$

or equivalently

$$f_{\tau_x}^{-1}(w) = \phi(F_{\hat{W}|x}(w)) \quad \forall x \in \mathcal{X}, w \in \mathbb{R}. \tag{167}$$

In Proposition 8, we show that, in the particular case where the conditional distributions $\{F_{\hat{W}|x}\}_{x \in \mathcal{X}}$ belong to a location family, there exists a simple choice of $\{f_\tau\}_{\tau \in \mathbb{R}}$, $\varphi \in \mathbb{R}$ and strictly increasing $\phi : \mathbb{R} \to \mathbb{R}$ such that the two methods are equivalent.

*Proposition* 8. Consider a scenario where all conditional distributions $\{F_{\hat{W}|x}\}_{x \in \mathcal{X}}$ belong to a location family, i.e.,

$$F_{\hat{W}|x} = F(w - \hat{\mu}_x) \text{ and } Q_{\hat{W}|x}(\alpha) = F^{-1}(\alpha) + \hat{\mu}_x, \tag{168}$$

for some continuous and strictly increasing base CDF $F$ and location parameter $\hat{\mu}_x$. The conformity scores $\bar{s}_{\text{ECDF}}$ and $\bar{s}_{\text{CP²}}$ lead to the same prediction regions.

*Proof.* We will show that there is a family of transformations $\{f_\tau\}_{\tau \in \mathbb{R}}$, $\varphi \in \mathbb{R}$ and strictly increasing $\phi : \mathbb{R} \to \mathbb{R}$ with the assumptions above such that, for any $x \in \mathcal{X}$ and $w \in \mathbb{R}$,

$$f_{\tau_x}^{-1}(w) = \phi(F_{\hat{W}|x}(w \mid X = x)) \tag{169}$$

Define the transformation function $f_\tau$ as:

$$f_\tau(\lambda) = F^{-1}(\lambda) + \tau, \tag{170}$$

where $\tau > 0$, and define $\varphi = 1 - \alpha$ and $\phi(\lambda) = \lambda$.

The inverse transformations are:

$$f_\tau^{-1}(\lambda) = F(\lambda - \tau), \tag{171}$$

and

$$g_\varphi^{-1}(w) = w - F^{-1}(\varphi). \tag{172}$$

Now, for $x \in \mathcal{X}$, compute

$$\hat{\tau}_x = F^{-1}(1 - \alpha) + \hat{\mu}_x - F^{-1}(\varphi) = \hat{\mu}_x. \tag{173}$$

Finally, we obtain the required equality

$$f_{\hat{\tau}_x}^{-1}(w) = F(w - \hat{\tau}_x) = F(w - \hat{\mu}_x) = F_{\hat{W}|x}(w). \tag{174}$$

$\square$

### I.4. Empirical comparison

We perform a direct empirical comparison between CDF-based methods (C-PCP, C-HDR) and the corresponding $CP^2$ methods ($CP^2$-PCP, $CP^2$-HPD using both linear (-L) and difference (-D) adjustments from Plassier et al., 2025). Figure 21 shows that:

- C-PCP performs comparably to CP²-PCP-L (best CP² variant for PCP).

- C-HDR performs comparably to CP²-HPD-D (best CP² variant for HPD).

- Other CP² variants (CP²-PCP-D, CP²-HPD-L) are generally outperformed by their CDF-based version.

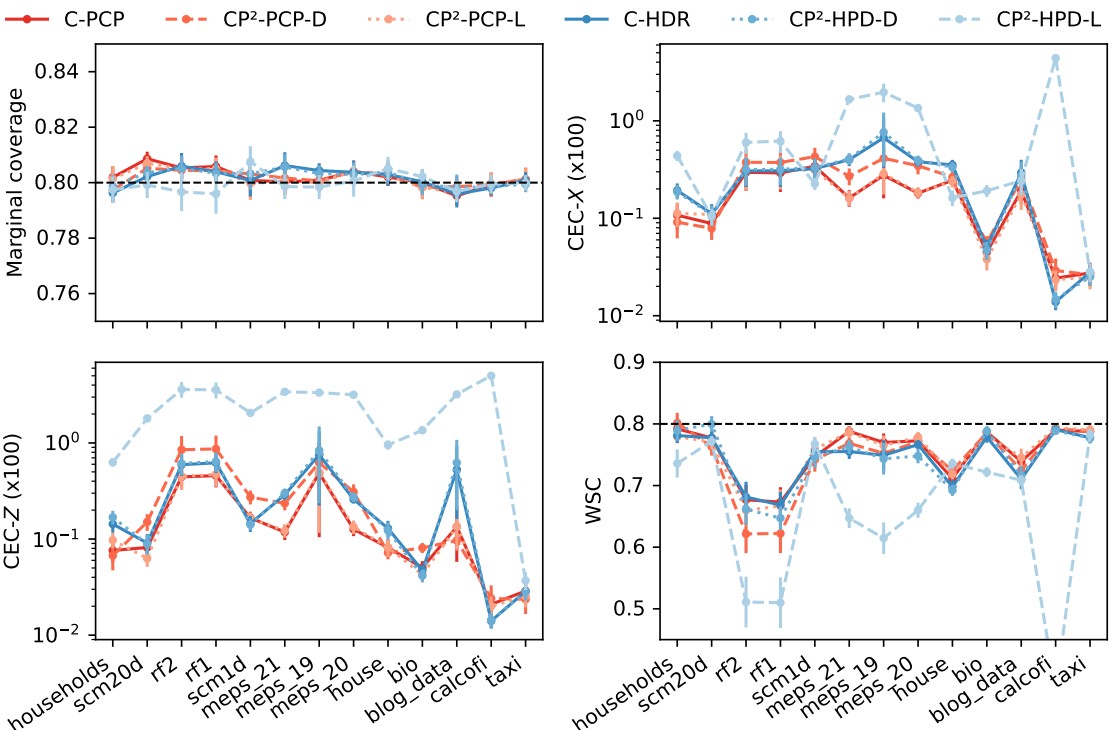

Figure 21: Comparison of CDF-base d methods and CP$^2$-based methods.

The worsened conditional coverage of CP$^2$-PCP-D is an interesting observation that was not observed in the smaller scale study of Plassier et al., 2025. In the case of CP$^2$-HPD-L, the poor conditional coverage is due to an incompatibility of the linear adjustment function with the (log-scaled) conformity score $s_{\text{DR-CP}}(x, y) = -\log \hat{f}(y|x)$, which can present negative values and thus a decreasing (instead of increasing) adjustment function $f_\tau(\lambda) = \tau\lambda$.

This shows that our simpler $s_{\text{ECDF}}$ formulation achieves the same practical benefits as $s_{\text{CP}^2}$ without the sensibility of choosing an adjustment function $f_\tau$.

## J. Results on an image dataset

To better understand the behavior of prediction regions in high-dimensional spaces, we apply conformal methods to the CIFAR-10 dataset (Krizhevsky et al., 2014), which consists of 32x32 RGB images, each labeled with one of 10 possible classes. We train a generative model conditioned on the image label, where $\mathcal{Y} = [0, 1]^{3\times32\times32}$ ($d = 3072$) represents the image space, and $\mathcal{X} = \{0, \ldots, 9\}$ ($p = 1$) represents the labels. The training, calibration, and test datasets contain 50,000, 1,500, and 1,500 images, respectively. As noted in Angelopoulos and Bates, 2023, this calibration dataset size is sufficient to ensure good marginal coverage.

Our generative model is a conditional Glow model (Kingma and Dhariwal, 2018) based on the implementation from Stimper et al., 2022 using a 3-level multi-scale architecture with 32 blocks per level. Like MQF$^2$ (Appendix F.2), this generative model is a normalizing flow and directly compatible with all methods presented, except M-CP. For a direct comparison with M-CP, we compute quantiles based on samples from the generative model as in Appendix F.2.

The latent space of the conditional Glow model, due to its multi-scale architecture, consists of three subspaces: $\mathcal{Z} = \mathcal{Z}_1 \times \mathcal{Z}_2 \times \mathcal{Z}_3$, where $\mathcal{Z}_1 = \mathbb{R}^{48\times4\times4}$, $\mathcal{Z}_2 = \mathbb{R}^{12\times8\times8}$, and $\mathcal{Z}_3 = \mathbb{R}^{6\times16\times16}$. As the distance function $d_\mathcal{Z}$ in the latent space, we use the maximum norm across the three spaces to penalize high norms in any of them: $d_\mathcal{Z}(z) = \max\{\|z_1\|_2, \|z_2\|_2, \|z_3\|_2\}$, where $z = z_1 \times z_2 \times z_3$.

Table 7 presents the metrics introduced in Appendix F.4. All methods achieve marginal coverage despite the high dimensionality of $\mathcal{Y}$, which is expected as the marginal coverage distribution conditional on the calibration dataset is

independent of $d$ (Appendix E.1). The median of the logarithm of the region size is the smallest for `C-HDR` and `DR-CP`, which matches results on tabular datasets. The mean region size is not reported because it becomes infinity with machine precision. Instead, we report the mean of the logarithm of the region size, with similar conclusions to the median region size.

Regarding conditional coverage, as in other experiments, `L-CP`, `C-HDR`, `C-PCP`, and `M-CP` exhibit the smallest CEC-$X$ and CEC-$Z$ values, indicating superior conditional coverage. The `WSC` metric supports similar conclusions, with `DR-CP` and `PCP` being the least calibrated.

Table 7: Results obtained with a conditional Glow model on CIFAR-10 with $1 - \alpha = 0.9$.

| Dataset | Method | MC | Median Log Size | Mean Log Size | CEC-$X$ ($\times 100$) | CEC-$Z$ ($\times 100$) | WSC | Time (s) |
|---|---|---|---|---|---|---|---|---|
| cifar10 | M-CP | $\mathbf{0.900_{0.0035}}$ | $-7.10\text{e+}03_{5.7}$ | $-7.05\text{e+}03_{1.3\text{e+}01}$ | $0.111_{0.028}$ | $0.201_{0.033}$ | $0.855_{0.012}$ | $\mathbf{0.465_{0.16}}$ |
| | DR-CP | $\mathbf{0.903_{0.0042}}$ | $\mathbf{-8.30\text{e+}03_{1.4\text{e+}01}}$ | $-8.33\text{e+}03_{1.4\text{e+}01}$ | $0.152_{0.030}$ | $0.325_{0.033}$ | $0.861_{0.0064}$ | $47.0_{1.7\text{e+}01}$ |
| | C-HDR | $\mathbf{0.902_{0.0041}}$ | $\mathbf{-8.33\text{e+}03_{1.9\text{e+}01}}$ | $\mathbf{-8.40\text{e+}03_{1.9\text{e+}01}}$ | $\mathbf{0.0533_{0.010}}$ | $\mathbf{0.0629_{0.020}}$ | $\mathbf{0.903_{0.0030}}$ | $453_{8.7\text{e+}01}$ |
| | PCP | $\mathbf{0.899_{0.0038}}$ | $-7.11\text{e+}03_{4.5}$ | $-7.06\text{e+}03_{1.3\text{e+}01}$ | $0.342_{0.070}$ | $0.195_{0.030}$ | $0.825_{0.0098}$ | $203_{3.5\text{e+}01}$ |
| | HD-PCP | $\mathbf{0.899_{0.0038}}$ | $-7.12\text{e+}03_{4.8}$ | $-7.06\text{e+}03_{1.2\text{e+}01}$ | $0.359_{0.075}$ | $0.198_{0.030}$ | $0.819_{0.0098}$ | $406_{6.9\text{e+}01}$ |
| | STDQR | $\mathbf{0.898_{0.0046}}$ | $-7.11\text{e+}03_{4.6}$ | $-7.06\text{e+}03_{1.2\text{e+}01}$ | $0.357_{0.071}$ | $0.205_{0.033}$ | $0.828_{0.015}$ | $204_{3.5\text{e+}01}$ |
| | C-PCP | $\mathbf{0.900_{0.0036}}$ | $-7.08\text{e+}03_{4.9}$ | $-7.04\text{e+}03_{1.1\text{e+}01}$ | $0.118_{0.021}$ | $\mathbf{0.0877_{0.023}}$ | $0.880_{0.0067}$ | $408_{6.9\text{e+}01}$ |
| | L-CP | $\mathbf{0.900_{0.0033}}$ | $-7.19\text{e+}03_{7.0}$ | $-7.15\text{e+}03_{1.1\text{e+}01}$ | $\mathbf{0.0668_{0.0086}}$ | $0.190_{0.027}$ | $0.877_{0.011}$ | $47.4_{1.8\text{e+}01}$ |

# K. Full results

Tables 8 and 9 show the full results obtained with the setup described in Section 6. Each metric is the mean over 10 independent runs. The standard error of the mean is indicated as an index. For each dataset and metric, bold values indicate results statistically similar to the best performer ($\alpha = 0.05$) according to a Z-test.

Table 8: Full results obtained with the setup described in Section 6 (Part 1).

| Dataset | Method | MC | Median Size | CEC-$X$ ($\times 100$) | CEC-$Z$ ($\times 100$) | WSC | Test time |
|---|---|---|---|---|---|---|---|
| households | M-CP | **0.801**$_{0.0051}$ | 14.2$_{0.48}$ | 0.340$_{0.068}$ | 0.364$_{0.032}$ | 0.779$_{0.010}$ | 5.69$_{0.49}$ |
| | CopulaCPTS | 0.782$_{0.0094}$ | 12.3$_{0.87}$ | 0.524$_{0.057}$ | 0.651$_{0.058}$ | 0.745$_{0.016}$ | 8.86$_{0.77}$ |
| | DR-CP | **0.802**$_{0.0046}$ | 13.2$_{0.29}$ | 0.987$_{0.10}$ | 1.88$_{0.14}$ | 0.656$_{0.018}$ | 0.225$_{0.0092}$ |
| | C-HDR | **0.807**$_{0.0054}$ | **10.6**$_{0.33}$ | **0.209**$_{0.039}$ | 0.149$_{0.020}$ | **0.795**$_{0.010}$ | 6.12$_{0.50}$ |
| | PCP | **0.798**$_{0.0048}$ | 20.5$_{0.38}$ | 1.07$_{0.085}$ | 2.35$_{0.15}$ | 0.632$_{0.015}$ | 5.48$_{0.46}$ |
| | HD-PCP | **0.800**$_{0.0043}$ | 15.6$_{0.39}$ | 0.776$_{0.091}$ | 1.38$_{0.10}$ | 0.707$_{0.014}$ | 5.76$_{0.47}$ |
| | STDQR | **0.804**$_{0.0050}$ | 17.8$_{0.41}$ | 0.918$_{0.073}$ | 1.97$_{0.098}$ | 0.677$_{0.019}$ | 8.45$_{0.79}$ |
| | C-PCP | **0.803**$_{0.0066}$ | 15.5$_{0.74}$ | **0.179**$_{0.045}$ | **0.120**$_{0.026}$ | **0.800**$_{0.0061}$ | 11.2$_{0.95}$ |
| | L-CP | **0.800**$_{0.0034}$ | 18.6$_{0.80}$ | **0.204**$_{0.040}$ | **0.118**$_{0.018}$ | **0.788**$_{0.014}$ | **0.101**$_{0.0043}$ |
| scm20d | M-CP | **0.800**$_{0.0039}$ | 67.6$_{8.5}$ | **0.0682**$_{0.011}$ | 0.914$_{0.061}$ | 0.777$_{0.0090}$ | 8.08$_{0.17}$ |
| | CopulaCPTS | 0.833$_{0.0086}$ | 1.12e+02$_{2.1e+01}$ | 0.221$_{0.063}$ | 0.878$_{0.043}$ | **0.802**$_{0.012}$ | 10.9$_{0.21}$ |
| | DR-CP | **0.799**$_{0.0048}$ | 2.33e+02$_{2.2e+01}$ | 0.429$_{0.044}$ | 2.72$_{0.16}$ | 0.691$_{0.018}$ | 0.560$_{0.025}$ |
| | C-HDR | **0.806**$_{0.0055}$ | 42.0$_{7.9}$ | 0.159$_{0.024}$ | **0.102**$_{0.017}$ | **0.796**$_{0.0065}$ | 9.42$_{0.18}$ |
| | PCP | **0.798**$_{0.0051}$ | 1.05e+02$_{1.1e+01}$ | 0.581$_{0.045}$ | 5.28$_{0.23}$ | 0.621$_{0.016}$ | 6.27$_{0.39}$ |
| | HD-PCP | **0.799**$_{0.0049}$ | 94.4$_{9.4}$ | 0.504$_{0.047}$ | 4.78$_{0.23}$ | 0.671$_{0.011}$ | 7.11$_{0.42}$ |
| | STDQR | **0.801**$_{0.0047}$ | 99.4$_{1.0e+01}$ | 0.540$_{0.052}$ | 4.86$_{0.17}$ | 0.620$_{0.016}$ | 8.15$_{0.29}$ |
| | C-PCP | 0.809$_{0.0038}$ | **26.0**$_{3.1}$ | 0.105$_{0.020}$ | **0.0896**$_{0.014}$ | **0.789**$_{0.0066}$ | 14.3$_{0.44}$ |
| | L-CP | **0.796**$_{0.0035}$ | 72.0$_{1.0e+01}$ | 0.166$_{0.033}$ | **0.0873**$_{0.018}$ | **0.786**$_{0.0063}$ | **0.0987**$_{0.0059}$ |
| rf2 | M-CP | **0.797**$_{0.0046}$ | 0.00547$_{0.0027}$ | 0.202$_{0.030}$ | 0.968$_{0.15}$ | 0.667$_{0.018}$ | 20.7$_{2.4}$ |
| | CopulaCPTS | 0.785$_{0.010}$ | 0.00555$_{0.0033}$ | 0.299$_{0.045}$ | 1.18$_{0.17}$ | 0.635$_{0.019}$ | 29.6$_{4.0}$ |
| | DR-CP | **0.799**$_{0.0028}$ | 0.00215$_{0.0010}$ | 0.949$_{0.21}$ | 5.42$_{0.70}$ | 0.549$_{0.038}$ | 0.344$_{0.015}$ |
| | C-HDR | **0.801**$_{0.0033}$ | **0.000690**$_{0.00032}$ | **0.111**$_{0.036}$ | 0.230$_{0.047}$ | **0.732**$_{0.018}$ | 21.5$_{2.4}$ |
| | PCP | **0.801**$_{0.0022}$ | 0.00700$_{0.0036}$ | 0.863$_{0.20}$ | 5.95$_{0.48}$ | 0.538$_{0.030}$ | 17.8$_{2.4}$ |
| | HD-PCP | **0.800**$_{0.0024}$ | 0.00617$_{0.0030}$ | 0.776$_{0.19}$ | 5.58$_{0.49}$ | 0.563$_{0.029}$ | 18.3$_{2.4}$ |
| | STDQR | **0.800**$_{0.0032}$ | 0.00624$_{0.0031}$ | 0.788$_{0.19}$ | 5.67$_{0.50}$ | 0.566$_{0.025}$ | 25.7$_{4.1}$ |
| | C-PCP | **0.802**$_{0.0051}$ | 0.00262$_{0.0012}$ | **0.0925**$_{0.016}$ | **0.169**$_{0.027}$ | **0.732**$_{0.017}$ | 38.6$_{4.8}$ |
| | L-CP | **0.800**$_{0.0026}$ | **0.00104**$_{0.00048}$ | **0.107**$_{0.032}$ | 0.236$_{0.042}$ | **0.730**$_{0.0093}$ | **0.0960**$_{0.0053}$ |
| rf1 | M-CP | **0.797**$_{0.0046}$ | 0.00547$_{0.0027}$ | 0.202$_{0.030}$ | 0.968$_{0.15}$ | 0.667$_{0.018}$ | 20.9$_{2.5}$ |
| | CopulaCPTS | 0.785$_{0.010}$ | 0.00555$_{0.0033}$ | 0.299$_{0.045}$ | 1.18$_{0.17}$ | 0.635$_{0.019}$ | 29.8$_{4.1}$ |
| | DR-CP | **0.799**$_{0.0028}$ | 0.00215$_{0.0010}$ | 0.949$_{0.21}$ | 5.42$_{0.70}$ | 0.549$_{0.038}$ | 0.335$_{0.016}$ |
| | C-HDR | **0.801**$_{0.0033}$ | **0.000690**$_{0.00032}$ | **0.111**$_{0.036}$ | 0.230$_{0.047}$ | **0.732**$_{0.018}$ | 21.7$_{2.5}$ |
| | PCP | **0.801**$_{0.0022}$ | 0.00700$_{0.0036}$ | 0.863$_{0.20}$ | 5.95$_{0.48}$ | 0.538$_{0.030}$ | 17.9$_{2.4}$ |
| | HD-PCP | **0.800**$_{0.0024}$ | 0.00617$_{0.0030}$ | 0.776$_{0.19}$ | 5.58$_{0.49}$ | 0.563$_{0.029}$ | 18.4$_{2.4}$ |
| | STDQR | **0.800**$_{0.0032}$ | 0.00624$_{0.0031}$ | 0.788$_{0.19}$ | 5.67$_{0.50}$ | 0.566$_{0.025}$ | 25.7$_{4.1}$ |
| | C-PCP | **0.802**$_{0.0051}$ | 0.00262$_{0.0012}$ | **0.0925**$_{0.016}$ | **0.169**$_{0.027}$ | **0.732**$_{0.017}$ | 38.8$_{4.9}$ |
| | L-CP | **0.800**$_{0.0026}$ | **0.00104**$_{0.00048}$ | **0.107**$_{0.032}$ | 0.236$_{0.042}$ | **0.730**$_{0.0093}$ | **0.0976**$_{0.0057}$ |
| scm1d | M-CP | **0.796**$_{0.0027}$ | 0.528$_{0.046}$ | 1.02$_{0.060}$ | 2.42$_{0.094}$ | 0.636$_{0.017}$ | 85.6$_{2.4e+01}$ |
| | CopulaCPTS | 0.732$_{0.011}$ | 0.323$_{0.050}$ | 1.72$_{0.20}$ | 3.49$_{0.27}$ | 0.582$_{0.017}$ | 87.8$_{2.4e+01}$ |
| | DR-CP | **0.793**$_{0.0036}$ | 0.867$_{0.078}$ | 1.50$_{0.087}$ | 5.17$_{0.20}$ | 0.559$_{0.020}$ | 0.584$_{0.025}$ |
| | C-HDR | 0.812$_{0.0046}$ | 0.239$_{0.026}$ | **0.452**$_{0.062}$ | **0.114**$_{0.015}$ | **0.761**$_{0.010}$ | 87.0$_{2.4e+01}$ |
| | PCP | **0.795**$_{0.0054}$ | 0.698$_{0.065}$ | 1.77$_{0.12}$ | 8.11$_{0.23}$ | 0.516$_{0.013}$ | 5.53$_{0.34}$ |
| | HD-PCP | **0.795**$_{0.0053}$ | 0.684$_{0.062}$ | 1.75$_{0.11}$ | 7.96$_{0.22}$ | 0.530$_{0.017}$ | 6.41$_{0.38}$ |
| | STDQR | **0.795**$_{0.0064}$ | 0.671$_{0.069}$ | 1.78$_{0.13}$ | 8.07$_{0.23}$ | 0.502$_{0.017}$ | 6.87$_{0.23}$ |
| | C-PCP | **0.803**$_{0.0053}$ | **0.216**$_{0.024}$ | 0.456$_{0.066}$ | 0.154$_{0.028}$ | **0.751**$_{0.0044}$ | 91.1$_{2.4e+01}$ |
| | L-CP | **0.799**$_{0.0045}$ | **0.197**$_{0.020}$ | 0.463$_{0.059}$ | **0.108**$_{0.017}$ | 0.731$_{0.014}$ | **0.102**$_{0.0056}$ |
| meps_21 | M-CP | **0.800**$_{0.0051}$ | 0.185$_{0.013}$ | 0.926$_{0.096}$ | 0.775$_{0.099}$ | 0.701$_{0.010}$ | 1.35e+02$_{1.7e+01}$ |
| | CopulaCPTS | 0.778$_{0.0064}$ | 0.171$_{0.014}$ | 0.957$_{0.13}$ | 0.693$_{0.099}$ | 0.684$_{0.011}$ | 1.61e+02$_{1.9e+01}$ |
| | DR-CP | **0.803**$_{0.0023}$ | 0.227$_{0.013}$ | 3.75$_{0.16}$ | 4.38$_{0.52}$ | 0.531$_{0.012}$ | 0.228$_{0.011}$ |
| | C-HDR | **0.807**$_{0.0046}$ | **0.132**$_{0.024}$ | 0.437$_{0.045}$ | 0.260$_{0.041}$ | 0.745$_{0.013}$ | 1.35e+02$_{1.7e+01}$ |
| | PCP | **0.801**$_{0.0031}$ | 0.359$_{0.021}$ | 3.17$_{0.13}$ | 3.75$_{0.44}$ | 0.550$_{0.0078}$ | 78.4$_{8.3}$ |
| | HD-PCP | **0.802**$_{0.0024}$ | 0.246$_{0.015}$ | 2.09$_{0.15}$ | 2.18$_{0.28}$ | 0.601$_{0.010}$ | 78.7$_{8.3}$ |
| | STDQR | **0.802**$_{0.0022}$ | 0.283$_{0.015}$ | 2.60$_{0.12}$ | 2.97$_{0.36}$ | 0.582$_{0.011}$ | 96.0$_{9.9}$ |
| | C-PCP | **0.805**$_{0.0026}$ | 0.220$_{0.021}$ | **0.165**$_{0.044}$ | **0.0851**$_{0.025}$ | **0.775**$_{0.0065}$ | 2.13e+02$_{2.3e+01}$ |
| | L-CP | **0.801**$_{0.0034}$ | 0.244$_{0.052}$ | 0.770$_{0.13}$ | 0.422$_{0.11}$ | 0.685$_{0.026}$ | **0.125**$_{0.0073}$ |
| meps_19 | M-CP | **0.803**$_{0.0027}$ | 0.214$_{0.022}$ | 0.702$_{0.049}$ | 0.622$_{0.086}$ | 0.709$_{0.0095}$ | 1.44e+02$_{2.3e+01}$ |
| | CopulaCPTS | **0.804**$_{0.022}$ | 0.595$_{0.42}$ | 1.13$_{0.26}$ | 0.926$_{0.29}$ | 0.721$_{0.030}$ | 1.77e+02$_{2.8e+01}$ |
| | DR-CP | 0.795$_{0.0028}$ | 0.175$_{0.011}$ | 3.91$_{0.18}$ | 3.98$_{0.73}$ | 0.501$_{0.013}$ | 0.224$_{0.011}$ |
| | C-HDR | **0.807**$_{0.0039}$ | **0.119**$_{0.019}$ | 0.380$_{0.036}$ | 0.245$_{0.039}$ | 0.753$_{0.013}$ | 1.44e+02$_{2.3e+01}$ |
| | PCP | 0.794$_{0.0033}$ | 0.396$_{0.059}$ | 2.95$_{0.23}$ | 3.51$_{0.53}$ | 0.542$_{0.013}$ | 99.9$_{1.7e+01}$ |
| | HD-PCP | 0.796$_{0.0032}$ | 0.266$_{0.033}$ | 1.98$_{0.14}$ | 2.05$_{0.35}$ | 0.583$_{0.0090}$ | 1.00e+02$_{1.7e+01}$ |
| | STDQR | 0.791$_{0.0032}$ | 0.307$_{0.043}$ | 2.63$_{0.23}$ | 2.95$_{0.49}$ | 0.557$_{0.014}$ | 1.17e+02$_{1.8e+01}$ |
| | C-PCP | 0.810$_{0.0021}$ | 0.238$_{0.026}$ | **0.128**$_{0.016}$ | **0.0757**$_{0.024}$ | **0.797**$_{0.0088}$ | 2.44e+02$_{3.9e+01}$ |
| | L-CP | **0.803**$_{0.0033}$ | 0.232$_{0.043}$ | 0.679$_{0.13}$ | 0.415$_{0.13}$ | 0.702$_{0.022}$ | **0.123**$_{0.0069}$ |

Table 9: Full results obtained with the setup described in Section 6 (Part 2).

| Dataset | Method | MC | Median Size | CEC-$X$ (×100) | CEC-$Z$ (×100) | WSC | Test time |
|---|---|---|---|---|---|---|---|
| meps_20 | M-CP | $\mathbf{0.806}_{0.0042}$ | $0.371_{0.061}$ | $0.868_{0.10}$ | $0.455_{0.12}$ | $0.702_{0.014}$ | $1.93e{+}02_{2.1e{+}01}$ |
| | CopulaCPTS | $\mathbf{0.794}_{0.0091}$ | $0.362_{0.059}$ | $0.963_{0.12}$ | $0.497_{0.12}$ | $0.692_{0.016}$ | $2.30e{+}02_{2.4e{+}01}$ |
| | DR-CP | $\mathbf{0.805}_{0.0036}$ | $0.223_{0.020}$ | $3.52_{0.11}$ | $2.79_{0.76}$ | $0.530_{0.0098}$ | $0.227_{0.010}$ |
| | C-HDR | $\mathbf{0.805}_{0.0044}$ | $\mathbf{0.114}_{0.012}$ | $0.439_{0.10}$ | $0.122_{0.036}$ | $0.745_{0.010}$ | $1.94e{+}02_{2.1e{+}01}$ |
| | PCP | $\mathbf{0.801}_{0.0036}$ | $0.535_{0.050}$ | $2.83_{0.091}$ | $2.42_{0.67}$ | $0.544_{0.0088}$ | $1.20e{+}02_{1.4e{+}01}$ |
| | HD-PCP | $\mathbf{0.804}_{0.0039}$ | $0.436_{0.066}$ | $1.89_{0.13}$ | $1.34_{0.37}$ | $0.622_{0.012}$ | $1.20e{+}02_{1.4e{+}01}$ |
| | STDQR | $\mathbf{0.803}_{0.0048}$ | $0.472_{0.052}$ | $2.45_{0.15}$ | $1.84_{0.50}$ | $0.575_{0.016}$ | $1.40e{+}02_{1.6e{+}01}$ |
| | C-PCP | $\mathbf{0.806}_{0.0041}$ | $0.341_{0.039}$ | $\mathbf{0.186}_{0.061}$ | $\mathbf{0.0484}_{0.010}$ | $\mathbf{0.792}_{0.013}$ | $3.13e{+}02_{3.1e{+}01}$ |
| | L-CP | $\mathbf{0.799}_{0.0033}$ | $0.280_{0.028}$ | $0.662_{0.073}$ | $0.259_{0.081}$ | $0.703_{0.014}$ | $\mathbf{0.127}_{0.0062}$ |
| house | M-CP | $\mathbf{0.801}_{0.0023}$ | $1.17_{0.023}$ | $0.254_{0.023}$ | $0.190_{0.019}$ | $0.730_{0.0098}$ | $56.0_{3.8e{+}01}$ |
| | CopulaCPTS | $0.812_{0.0082}$ | $1.22_{0.043}$ | $0.316_{0.035}$ | $0.276_{0.027}$ | $\mathbf{0.750}_{0.012}$ | $60.7_{3.8e{+}01}$ |
| | DR-CP | $\mathbf{0.801}_{0.0041}$ | $\mathbf{0.664}_{0.021}$ | $0.895_{0.045}$ | $1.20_{0.073}$ | $0.627_{0.011}$ | $0.283_{0.011}$ |
| | C-HDR | $0.807_{0.0039}$ | $\mathbf{0.651}_{0.016}$ | $0.388_{0.026}$ | $0.114_{0.013}$ | $0.709_{0.010}$ | $56.6_{3.8e{+}01}$ |
| | PCP | $\mathbf{0.801}_{0.0026}$ | $0.882_{0.023}$ | $0.753_{0.030}$ | $1.14_{0.038}$ | $0.643_{0.0076}$ | $17.6_{0.98}$ |
| | HD-PCP | $\mathbf{0.803}_{0.0034}$ | $0.680_{0.018}$ | $0.694_{0.033}$ | $0.789_{0.035}$ | $0.649_{0.0089}$ | $18.0_{0.99}$ |
| | STDQR | $\mathbf{0.801}_{0.0042}$ | $0.799_{0.023}$ | $0.670_{0.022}$ | $0.788_{0.038}$ | $0.649_{0.0077}$ | $19.5_{0.88}$ |
| | C-PCP | $0.809_{0.0030}$ | $0.858_{0.018}$ | $0.275_{0.026}$ | $0.0831_{0.011}$ | $0.729_{0.0091}$ | $73.7_{3.8e{+}01}$ |
| | L-CP | $\mathbf{0.802}_{0.0035}$ | $1.19_{0.017}$ | $\mathbf{0.174}_{0.020}$ | $\mathbf{0.0542}_{0.0079}$ | $\mathbf{0.756}_{0.0090}$ | $\mathbf{0.146}_{0.0067}$ |
| bio | M-CP | $0.809_{0.0021}$ | $0.303_{0.0066}$ | $0.137_{0.0093}$ | $0.253_{0.013}$ | $0.764_{0.0055}$ | $1.27e{+}02_{6.0}$ |
| | CopulaCPTS | $\mathbf{0.800}_{0.0045}$ | $0.296_{0.0092}$ | $0.137_{0.0083}$ | $0.260_{0.015}$ | $0.751_{0.0068}$ | $1.45e{+}02_{7.1}$ |
| | DR-CP | $\mathbf{0.805}_{0.0020}$ | $0.257_{0.0067}$ | $0.507_{0.028}$ | $1.14_{0.034}$ | $0.646_{0.0066}$ | $0.511_{0.020}$ |
| | C-HDR | $0.808_{0.0015}$ | $\mathbf{0.218}_{0.0053}$ | $0.0372_{0.0073}$ | $0.0360_{0.0056}$ | $\mathbf{0.794}_{0.0054}$ | $1.29e{+}02_{6.0}$ |
| | PCP | $\mathbf{0.802}_{0.0021}$ | $0.343_{0.0076}$ | $0.567_{0.029}$ | $1.32_{0.023}$ | $0.628_{0.0052}$ | $1.27e{+}02_{6.1}$ |
| | HD-PCP | $\mathbf{0.804}_{0.0016}$ | $0.259_{0.0065}$ | $0.352_{0.020}$ | $0.803_{0.020}$ | $0.673_{0.0043}$ | $1.27e{+}02_{6.1}$ |
| | STDQR | $\mathbf{0.803}_{0.0024}$ | $0.269_{0.0067}$ | $0.389_{0.019}$ | $0.912_{0.036}$ | $0.667_{0.0058}$ | $86.6_{6.5}$ |
| | C-PCP | $0.810_{0.0029}$ | $0.302_{0.0074}$ | $0.0369_{0.0063}$ | $0.0404_{0.0069}$ | $\mathbf{0.798}_{0.0052}$ | $2.54e{+}02_{1.2e{+}01}$ |
| | L-CP | $\mathbf{0.805}_{0.00093}$ | $0.267_{0.0061}$ | $\mathbf{0.0203}_{0.0045}$ | $\mathbf{0.0198}_{0.0021}$ | $0.789_{0.0039}$ | $\mathbf{0.251}_{0.013}$ |
| blog_data | M-CP | $\mathbf{0.802}_{0.0049}$ | $0.170_{0.039}$ | $0.292_{0.051}$ | $0.153_{0.072}$ | $0.736_{0.012}$ | $5.06e{+}03_{7.0e{+}02}$ |
| | CopulaCPTS | $0.813_{0.0078}$ | $0.0948_{0.015}$ | $0.313_{0.050}$ | $0.231_{0.063}$ | $0.742_{0.010}$ | $5.13e{+}03_{7.1e{+}02}$ |
| | DR-CP | $0.808_{0.0014}$ | $0.0374_{0.0056}$ | $1.06_{0.098}$ | $1.50_{0.43}$ | $0.644_{0.0059}$ | $0.515_{0.026}$ |
| | C-HDR | $0.809_{0.0030}$ | $\mathbf{0.0155}_{0.0031}$ | $0.237_{0.068}$ | $\mathbf{0.0611}_{0.019}$ | $\mathbf{0.751}_{0.013}$ | $5.06e{+}03_{7.0e{+}02}$ |
| | PCP | $\mathbf{0.801}_{0.0033}$ | $0.141_{0.023}$ | $0.938_{0.081}$ | $1.52_{0.36}$ | $0.643_{0.0052}$ | $5.74e{+}02_{7.9e{+}01}$ |
| | HD-PCP | $\mathbf{0.803}_{0.0038}$ | $0.125_{0.023}$ | $0.794_{0.075}$ | $0.945_{0.22}$ | $0.660_{0.0080}$ | $5.75e{+}02_{7.9e{+}01}$ |
| | STDQR | $\mathbf{0.810}_{0.0072}$ | $0.163_{0.036}$ | $0.805_{0.074}$ | $0.881_{0.18}$ | $0.678_{0.012}$ | $5.84e{+}02_{8.0e{+}01}$ |
| | C-PCP | $\mathbf{0.804}_{0.0045}$ | $0.106_{0.021}$ | $\mathbf{0.163}_{0.049}$ | $0.113_{0.056}$ | $\mathbf{0.764}_{0.012}$ | $5.63e{+}03_{7.3e{+}02}$ |
| | L-CP | $\mathbf{0.801}_{0.0023}$ | $0.0676_{0.017}$ | $0.327_{0.088}$ | $\mathbf{0.0624}_{0.023}$ | $0.722_{0.012}$ | $\mathbf{0.258}_{0.012}$ |
| calcofi | M-CP | $\mathbf{0.803}_{0.0023}$ | $2.13_{0.024}$ | $0.433_{0.015}$ | $0.446_{0.016}$ | $0.734_{0.0069}$ | $26.4_{1.1}$ |
| | CopulaCPTS | $0.815_{0.0075}$ | $2.38_{0.12}$ | $0.480_{0.048}$ | $0.492_{0.048}$ | $0.746_{0.0096}$ | $29.6_{1.2}$ |
| | DR-CP | $\mathbf{0.805}_{0.0027}$ | $\mathbf{1.67}_{0.022}$ | $1.44_{0.040}$ | $1.56_{0.039}$ | $0.654_{0.0061}$ | $0.529_{0.023}$ |
| | C-HDR | $\mathbf{0.805}_{0.0018}$ | $1.99_{0.026}$ | $\mathbf{0.0294}_{0.012}$ | $\mathbf{0.0187}_{0.0037}$ | $0.794_{0.0053}$ | $27.7_{1.2}$ |
| | PCP | $\mathbf{0.802}_{0.0026}$ | $2.33_{0.029}$ | $1.64_{0.042}$ | $1.79_{0.041}$ | $0.638_{0.0034}$ | $26.5_{1.2}$ |
| | HD-PCP | $\mathbf{0.802}_{0.0033}$ | $1.89_{0.029}$ | $0.980_{0.033}$ | $1.05_{0.030}$ | $0.683_{0.0050}$ | $27.3_{1.2}$ |
| | STDQR | $\mathbf{0.799}_{0.0034}$ | $1.97_{0.021}$ | $1.13_{0.031}$ | $1.23_{0.033}$ | $0.676_{0.0080}$ | $26.4_{0.99}$ |
| | C-PCP | $0.809_{0.0030}$ | $2.81_{0.042}$ | $\mathbf{0.0332}_{0.0093}$ | $\mathbf{0.0253}_{0.0048}$ | $\mathbf{0.806}_{0.0050}$ | $52.9_{2.3}$ |
| | L-CP | $\mathbf{0.800}_{0.0020}$ | $2.70_{0.024}$ | $\mathbf{0.0332}_{0.019}$ | $\mathbf{0.0179}_{0.0040}$ | $0.792_{0.0035}$ | $\mathbf{0.264}_{0.012}$ |
| taxi | M-CP | $\mathbf{0.802}_{0.0032}$ | $4.26_{0.068}$ | $0.0585_{0.0034}$ | $0.0421_{0.0058}$ | $0.784_{0.0052}$ | $60.6_{8.6}$ |
| | CopulaCPTS | $0.822_{0.0040}$ | $4.72_{0.11}$ | $0.114_{0.018}$ | $0.0989_{0.019}$ | $\mathbf{0.799}_{0.0050}$ | $68.4_{9.7}$ |
| | DR-CP | $\mathbf{0.805}_{0.0024}$ | $\mathbf{2.62}_{0.029}$ | $0.383_{0.016}$ | $0.451_{0.024}$ | $0.707_{0.0048}$ | $0.539_{0.031}$ |
| | C-HDR | $0.809_{0.0030}$ | $\mathbf{2.62}_{0.033}$ | $0.0388_{0.0049}$ | $0.0441_{0.0053}$ | $\mathbf{0.793}_{0.0040}$ | $61.9_{8.6}$ |
| | PCP | $\mathbf{0.804}_{0.0016}$ | $4.03_{0.040}$ | $0.341_{0.022}$ | $0.399_{0.025}$ | $0.715_{0.0042}$ | $60.2_{8.5}$ |
| | HD-PCP | $\mathbf{0.805}_{0.0018}$ | $3.18_{0.030}$ | $0.194_{0.012}$ | $0.219_{0.012}$ | $0.750_{0.0055}$ | $61.1_{8.5}$ |
| | STDQR | $\mathbf{0.805}_{0.0035}$ | $3.63_{0.058}$ | $0.203_{0.011}$ | $0.224_{0.013}$ | $0.748_{0.0080}$ | $32.4_{1.1}$ |
| | C-PCP | $0.807_{0.0026}$ | $4.02_{0.064}$ | $\mathbf{0.0307}_{0.0053}$ | $\mathbf{0.0338}_{0.0048}$ | $\mathbf{0.802}_{0.0050}$ | $1.21e{+}02_{1.7e{+}01}$ |
| | L-CP | $\mathbf{0.805}_{0.0033}$ | $4.94_{0.12}$ | $\mathbf{0.0264}_{0.0030}$ | $\mathbf{0.0196}_{0.0035}$ | $0.796_{0.0046}$ | $\mathbf{0.243}_{0.012}$ |

