# OpenReview forum: "A Unified Comparative Study with Generalized Conformity Scores for Multi-Output Conformal Regression"
_ICML.cc/2025/Conference — ICML 2025 poster_

### Official Review · Reviewer_jr27 · 2025-02-16

**Overall Recommendation:** 2

**Summary:**

The paper addresses the problem of applying conformal prediction to multiple continuous output prediction systems. The challenge is obtaining a small average region with a reasonable computation complexity. The paper presents a unified view of the current methods and proposes two novel approaches.

**Claims And Evidence:**

yes

**Essential References Not Discussed:**

There is a closely related research direction of multiple testing in parallel in conformal prediction and family-wise error rate control. Popular methods are Bonferroni and Bonferroni-Holm  corrections of the rectangular
shaped region.

**Experimental Designs Or Analyses:**

Yes

**Methods And Evaluation Criteria:**

yes

**Other Comments Or Suggestions:**

no

**Other Strengths And Weaknesses:**

The topic is important, and this paper is well-written and provides an up-to-date comparative review of current methods. It can be a good review paper for people interested in the topic.

After comparing many methods it is not clear what the conclusion is.  The conclusion section is short and non-informative.

The presentation could be improved. The notation is sometimes too dense, making it challenging to follow.

The introduction could provide a clearer motivation for the need for multi-output conformal methods.

**Questions For Authors:**

How is it related to family-wise error rate control methods? Popular methods are Bonferroni and Bonferroni-Holm corrections of the rectangular. I was surprised that a review paper didn't even mention this line of research.

While the tabular datasets are informative, additional experiments on real-world high-dimensional or unstructured data (e.g., images, text) would better demonstrate the practical impact of the proposed methods.

For the methods DR-CP, C-HDR, and others, It is not clear how you know the density f(x|y) from the data.

**Relation To Broader Scientific Literature:**

The main contribution is a unified treatment of previously proposed methods.

**Theoretical Claims:**

yes

---

> ### Author Rebuttal · Authors · 2025-04-01
>
> Thank you for your feedback and suggestions for improvement.
>
> **Missing relation to family-wise error rate (FWER) control methods:**
>
> Thank you for raising this point. We do briefly mention multiplicity control approaches in Appendix A (citing Timans et al., 2024 [59]). FWER methods (like Bonferroni or Holm corrections on univariate CPs) are indeed a related direction for constructing multivariate regions, typically by taking Cartesian products of univariate regions $\hat{R}\_i$. Their strength lies in leveraging well-understood univariate CP. However, our work focuses on methods that explicitly model the joint distribution $F\_{Y|X}$ (using e.g., multivariate NFs, DRFs, GMMs, or generative sampling) to construct potentially non-rectangular regions that can better capture output dependencies and often result in smaller region sizes (as seen empirically). While FWER methods are valuable, especially when only marginal predictions are available, the methods we survey and propose aim for potentially tighter regions by directly modeling the multivariate structure. We have revised Appendix A to better clarify this distinction.
>
> **Conclusion unclear/short:**
>
> We agree that the conclusion was too brief and lacked practical guidance. We have expanded it to provide clear, actionable takeaways based on our unified study and experiments, summarizing the trade-offs:
> *   **If only univariate margin models are available/desired:** Use M-CP or CopulaCPTS (fast, but hyperrectangular; may poorly capture dependencies).
> *   **To minimize *average* region size (potentially sacrificing conditional coverage):** Use DR-CP (if density $\hat{f}$ available), else STDQR (if invertible latent model Q available), else PCP (if only samples $\hat{Y}$ available).
> *   **To achieve good *conditional coverage* and small *median* size:** Use C-HDR (if density $\hat{f}$ available), else L-CP (if invertible Q available), else C-PCP (if only samples $\hat{Y}$ available).
> *   **For good conditional coverage with lowest *computational cost* among ACC methods:** Use L-CP (if invertible Q available).
>
> **Questions**
> 1.  *Relation to FWER:* Please see our response above regarding the distinction (modeling joint vs. combining univariate CPs, region shapes, and dependency handling). To better understand the prediction regions produced by FWER control methods, we also provide a qualitative and quantitative comparison with Bonferroni correction on our dataset from Figure 1 in [this linked PDF](https://pdfhost.io/v/gqCAunzaRU_rebuttal-jr27). The method is fast computationally but produces larger regions due to the rectangular shape, similarly to M-CP.
>
> 2.  *Need high-dimensional/unstructured data experiments:* We agree on the importance of high-dimensional settings. To this end, we included an experiment on CIFAR-10 image generation (Appendix I), where the output is a 3x32x32 image ($d$=3072). We used a conditional Glow model [34, 54]. The results (Table 3) largely confirm our findings from tabular data: C-PCP, L-CP, and C-HDR demonstrate superior conditional coverage (WSC, CEC) compared to other methods even in this high-dimensional setting. We agree this experiment deserves more visibility and will ensure it is also mentioned in the introduction.
>
> 3. *How is the density $f(y|x)$ known?*
> Crucially, we do not assume the true density $f_{Y|X}$ is known. All density-based methods (DR-CP, C-HDR, HD-PCP) rely on an estimate $\hat{f}(y|x)$ learned from data. In our experiments, $\hat{f}$ is provided by models like normalizing flows (MQF², Glow), DRF+KDE, or GMMs (detailed in Appendix F.2). Conformal prediction takes this potentially imperfect estimate $\hat{f}$ and calibrates the resulting prediction regions (using the calibration set) to provide valid finite-sample marginal coverage guarantees, regardless of how accurate $\hat{f}$ is. Methods like C-HDR, L-CP, and C-PCP additionally provide asymptotic conditional coverage guarantees as long as the estimator $\hat{f}$ (or sampler $\hat{Y}$, or latent map $\hat{Q}$) converges to the true data generating process.

---

### Official Review · Reviewer_g7mK · 2025-03-12

**Overall Recommendation:** 4

**Summary:**

This paper provides an overview of multi-output conformal regression methods, putting them in a unified setting, and propose two new approaches based on scores CDF, that generalizes some previous methods, as well as a latent-based approach, that generalizes other families of approaches.

## update after rebuttal

I thank the authors for their replies. Overall the paper was quite enjoyable to me, and while as raised by other reviewers there are certainly ways to get further results, I do believe it makes a nice contribution with respect to the current literature on mutli-output conformal regression.

**Claims And Evidence:**

Yes, the claims made about the new method, notably in terms of conditional coverage, are supported both by theoretical proofs (appendices E.2., that seem correct as far as I could tell with the time I could spend on it), as well as by extensive experiments (appendix F, notably).

**Essential References Not Discussed:**

Most references I could think of were present in the paper, that appears to be very well educated in the field.

**Experimental Designs Or Analyses:**

I check the experimental designs, and all appeared sound to me. Graphs and results appear to be in-line with expectations.

**Methods And Evaluation Criteria:**

The methods and evaluations criteria are classical metrics and arguments from conformal prediction papers dealing with multi-output regression, as well as some new unbiased metrics regarding the region size (discussed in appendix F). Overall the used methods appear to be sound.

**Other Comments Or Suggestions:**

No specific comment, except that authors should double check references (some do not compile as expected, e.g., [37] and [61])

**Other Strengths And Weaknesses:**

Overall, this is a quite enjoyable paper that wraps up many method of the literature and provides in addition some interesting new ones. I see no major weaknesses in the paper. My only regret is that this probably should have been a journal paper rather than a 6 pages paper with 30 pages of supplementaries, but I guess this is how things are done nowadays.

**Questions For Authors:**

No specific questions to the authors, this is a nice work overall.

**Relation To Broader Scientific Literature:**

Yes, in the sense that providing reliable multi-output regression regions is a generally interesting problem.

**Theoretical Claims:**

I checked the claims as best as I could, and they appear correct to me. However, given the average size of ICML submissions (say, 30 pages with supplementaries included) and the number of reviews required by the confrence, my checks were not as thorough as they would have been for, say, a journal paper.

---

> ### Author Rebuttal · Authors · 2025-04-01
>
> Thank you for the thorough, positive review and your support for acceptance. We appreciate you finding the paper enjoyable and the methods sound. We will correct the reference issues you kindly pointed out in the final version.

---

### Official Review · Reviewer_ReUg · 2025-03-13

**Overall Recommendation:** 3

**Summary:**

This paper performs a unified comparative study of existing conformal methods with different multivariate base models for constructing multivariate prediction regions. It generalizes two classes of conformity scores from the univariate to the multivariate case. Moreover, it conducts large-scale experiments comparing the different multi-output conformal methods.

**Claims And Evidence:**

Yes, the multivariate conformity scores introduced ensure asymptotic conditional coverage while maintaining exact finite-sample marginal coverage. The proofs are provided.

**Essential References Not Discussed:**

Not that I know of.

**Experimental Designs Or Analyses:**

I read the descriptions of the experiments in the paper and did not see major issues.

**Methods And Evaluation Criteria:**

Yes, the comparative study and the large-scale experiments are evaluated on a wide range of conformal methods and datasets.

**Other Comments Or Suggestions:**

This work performs an extensive study on multivariate conformal methods and introduces two multivariate conformity scores. This paves the way for a research direction which is relatively underexplored.

**Other Strengths And Weaknesses:**

Strengths:
- The paper is well-written
- This work studies multivariate conformal prediction, which is relatively underexplored. This can provide a valuable direction for future research.
- Two multivariate conformity scores, which generalizes existing univariate conformity scores, are introduced. Theoretical analyses are provided.
- This work gives a detailed and unified comparative study for existing multivariate conformal models.
- The experiments cover a wide range of methods
- The methods are implemented using a unified code base, ensuring fairness

Weaknesses:
- The proposed scores seem not to have finite-sample conditional guarantees

**Questions For Authors:**

No

**Relation To Broader Scientific Literature:**

This paper generalizes two commonly used classes of conformity scores to the multivariate case. This is an important contribution as the classes of multivariate conformity scores are underexplored in the literature. Moreover, this paper provides an extensive comparison of existing conformal methods.

**Theoretical Claims:**

I did not go through the proofs for the theoretical claims.

---

> ### Author Rebuttal · Authors · 2025-04-01
>
> Thank you for your feedback and recognizing the value of our work.
>
> **Proposed scores seem not to have finite-sample conditional guarantees:**
>
> You are correct. Our focus is on developing flexible conformal methods applicable to complex, modern generative models (density, sample, or latent-based) with minimal assumptions on the underlying data distribution or the base predictor's accuracy. Under such weak, distribution-free assumptions, achieving exact finite-sample conditional coverage is generally impossible, as proven by Barber et al. (2019) [5]. Instead, we provide asymptotic conditional coverage (ACC) guarantees (Appendix E.2) for C-PCP, L-CP, and C-HDR. This means that as the base model converges to the true distribution and the calibration set size increases, these methods provably achieve the desired conditional coverage level. Our experiments (Fig 3) further empirically validate that these methods also achieve superior empirical conditional coverage in finite samples compared to methods lacking ACC guarantees.

---

### Official Review · Reviewer_mLNW · 2025-03-19

**Overall Recommendation:** 2

**Summary:**

The paper considers conformal prediction for high-dimensional regression. While one can extend uni-dimensional regression algorithms to multi-dimensional ones, other algorithms that explicitly work in $\geq 1$ dimensions also exist.

This paper introduces two conformity scores, C-PCP and L-CP, exploring their connection to existing methods. Further, the paper experimentally compares different algorithms on various synthetic and real-world datasets.

**Claims And Evidence:**

This paper is a comparative study.

**Essential References Not Discussed:**

NA

**Experimental Designs Or Analyses:**

The paper evaluates the algorithms on marginal coverage, conditional coverage, set sizes, and computational time. A few questions/suggestions:

1.	Replacing Fig. 5 with a table containing mean and median set sizes would help. The actual values convey more information than relative rankings.

2.	Is there a quantitative analysis for Section 5.1?

3.	Figs. 7 and 8 are difficult to compare.

4.	Tables 3, 4, and 5 should bold the statistically significant values (accounting for the standard deviations).

**Methods And Evaluation Criteria:**

The paper uses a variety of synthetic and real-world datasets. Furthermore, the paper compares multiple algorithms for its study. The paper mentions that CP$^{2}$-PCP is similar to C-PCP (lines 195-197, column 1). What is the reason for not including it?

**Other Comments Or Suggestions:**

1.	The existing methods that C-PCP and L-CP build on should be discussed in detail in the main paper. For example, details of directional quantile regression [Feldman et al., 2023].

2.	The non-conformity score need not be deterministic (lines 83-84, column 2). As long as the training data is independent of the calibration and test data, marginal coverage is guaranteed.

3.	The conformity score by Sadinle et al. [2016] is for classification. Even though one can use it for regression, one should explicitly mention that.

4.	The citations need to be corrected. For example, Sadinle et al. [2016] was in the Journal of the American Statistical Association in 2019.

Typo:

1.	"...$D_{cal} \rightarrow \infty$..." $\rightarrow$ "...$|D_{cal}| \rightarrow \infty$..." (line 162, column 2)

**Other Strengths And Weaknesses:**

Strengths:

1.	The paper considers conformal prediction for high-dimensional regression, which is of practical importance.

2.	The paper compares different conformity scores on various datasets. This comparison includes empirical results and some theoretical links.

Weaknesses:

1.	The novelty seems limited to the proposed conformity scores C-PCP and L-CP.

2.	It is unclear if and when the proposed conformity scores perform better than the existing ones. C-HDR seems to perform the best.

**Questions For Authors:**

1.	What are the challenges with extending conformal prediction to high-dimensional regression (Section 4)?

2.	What causes the absence of contours in Fig. 2 (lines 260-264, column 1)? The method should still construct a set.

3.	I am confused about the claim of the introduction of new classes of conformity scores (lines 124-126, column 2). Don't these classes already exist? The paper also provides existing methods for both classes.

4.	Is there a practical takeaway from Proposition 1?

5.	Do C-PCP and L-CP use Euclidean distance or a different distance?

6.	Does the analysis change if the setup switches to exchangeable data (Section 2)?

7.	What do the acronyms C-PCP and L-CP stand for?

**Relation To Broader Scientific Literature:**

The paper compares different conformal prediction methods for high-dimensional regression. It also introduces 2 new conformity scores.

**Theoretical Claims:**

I checked the correctness of Proposition 1.

---

> ### Author Rebuttal · Authors · 2025-04-01
>
> Thank you for your valuable feedback.
>
> **The paper mentions that CP²-PCP is similar to C-PCP. What is the reason for not including it?**
>
> CP²-PCP was proposed concurrently with our submission period and published very recently (ICLR 2025). Hence, we were unable to include it in our empirical comparisons. However, recognizing its relevance, we have added an in-depth discussion in Appendix H comparing CP²-PCP with our CDF-based scores (including C-PCP) and highlighting the similarities and differences in their approaches to achieving conditional validity.
>
> **The novelty seems limited to the proposed conformity scores C-PCP and L-CP**
>
> In addition to the proposed conformity scores C-PCP and L-CP, this work is the first (to our knowledge) to categorize systematically, implement (in a unified codebase for fairness), and empirically compare a broad range of multi-output conformal methods (marginal, density, sample, latent-based) within a single framework. This synthesis reveals crucial trade-offs (Table 1), motivates our proposed methods (C-PCP/L-CP each address specific gaps), and establishes theoretical connections, which were previously unexplored.
>
> **Unclear if/when proposed scores perform better; C-HDR seems best**
>
> C-HDR indeed performs very well, especially for region size when density estimation is accurate and feasible. However, C-PCP and L-CP offer distinct practical advantages in specific scenarios:
> *   **C-PCP:** Requires **no explicit density estimation or likelihood evaluation**, only samples from a generative model. This makes it applicable to models like diffusion models or GANs where densities might be intractable or unavailable, situations where C-HDR cannot be used.
> *   **L-CP:** Achieves asymptotic conditional coverage (ACC) with **significantly lower computational cost** (Figure 6, often >100x faster) compared to C-HDR and C-PCP, as it avoids per-instance sampling or complex density calculations.
> Both C-PCP and L-CP achieve asymptotic conditional coverage (ACC), unlike DR-CP/PCP/HD-PCP/STDQR/M-CP/CopulaCPTS. Their region sizes are competitive, generally only surpassed by C-HDR (if applicable) and DR-CP (which lacks ACC guarantees). Table 1 and the Conclusion summarize these trade-offs.
>
> **Suggestions**
>
> Thank you, these are excellent suggestions for improving clarity. We have implemented them as described in [this linked PDF](https://pdfhost.io/v/VUnLttqXdJ_rebuttal) and will incorporate other comments in the final version.
>
> **Questions:**
> 1.  *Challenges in high-dim CP:* Primarily, the lack of a natural ordering (unlike 1D) makes simple extensions of univariate methods (like CQR directly) difficult. This motivates methods based on joint density (DR-CP, C-HDR), samples (PCP, C-PCP), or latent spaces (L-CP, STDQR) to capture complex dependencies and define multivariate regions.
> 2.  *DR-CP empty contours (Fig 2):* This occurs when, for a given $x$ and a desired coverage level (e.g., $x=1$ and $1-\alpha$ = 0.2), no region in the output space achieves a density value $\hat{f}(y|x)$ high enough to be less than or equal to the constant threshold $-\hat{q}$. This visually demonstrates DR-CP's lack of conditional coverage. Methods with asymptotic conditional coverage (C-HDR, C-PCP, L-CP) adapt their thresholds or regions based on x and avoid this issue asymptotically.
> 3. *Claim of new classes:* We introduce generalizations of existing univariate concepts to the multivariate setting, forming broader classes:
> - CDF-based: Generalizes the univariate HPD-split/C-HDR score [31] to work with any base multivariate conformity score $s_W$ (Eq. 11), leading specifically to C-PCP when $s_W=s_\text{PCP}$.
> - Latent-based: Generalizes univariate Distributional CP [13] to multivariate outputs using any invertible conditional generative model Q and any latent distance function $d_\mathcal{Z}$ (Eq. 14), leading to L-CP. Feldman et al. [20] also uses a latent space but performs CP on grid samples mapped to the *output* space, not directly in the latent space like L-CP.
> 4. *Practical takeaway from Prop 1:* An interesting practical takeaway follows from the fact that DR-CP and C-HDR are linked in the same way as PCP and C-PCP. Since DR-CP can be shown to asymptotically have the smallest average region size while C-HDR empirically has a smaller median region size, similar observations are expected for PCP and C-PCP. This is verified empirically: PCP has a smaller average region size across all base predictors, while C-PCP has a smaller median region size. We have added this insight to Section 5.3.
> 5. *Distance for C-PCP/L-CP:* In our experiments, PCP (and thus C-PCP) uses Euclidean distance, and L-CP uses the Euclidean norm in the latent space. Both could be generalized to other metrics.
> 6.  *i.i.d. vs exchangeable data:* Thank you for pointing this out, the analysis remains valid. We will relax this assumption.
> 7. *Acronyms:* C-PCP = CDF-based Probabilistic Conformal Prediction. L-CP = Latent-based Conformal Prediction.

---

### Official Review · Reviewer_zeNu · 2025-03-26

**Overall Recommendation:** 3

**Summary:**

This paper reviews latest developments of conformal prediction methods in multi-output regression tasks.

**Claims And Evidence:**

Yes. The paper did a comprehensive overview and detailed analysis of various methods, categorized them into different variants, and compared the results both numerically and visually.

**Essential References Not Discussed:**

NA

**Experimental Designs Or Analyses:**

The experiments are well designed.

**Methods And Evaluation Criteria:**

Yes. The paper used popular datasets and common evaluation criteria.

**Other Comments Or Suggestions:**

No further suggestions.

**Other Strengths And Weaknesses:**

Strength:
1. The paper is a comprehensive review of an important topic in conformal prediction.

Weakness:
1. Many CP methods rely on conditional density estimation techniques and can integrate with various approaches. Were the experiments conducted to compare the performance of CP methods under different conditional density estimation variants to ensure a comprehensive evaluation?
2. Specifically, while the original ST-DQR paper used conditional VAE, this paper employs conditional normalizing flows. Did the authors perform a comparison between these two approaches?
3. More broadly, for a fair comparison, CP methods should be evaluated based on their optimal performance when paired with different conditional density estimation methods. Were such evaluations conducted to ensure fairness and completeness?

**Questions For Authors:**

NA

**Relation To Broader Scientific Literature:**

The work is important and provides an overview of the multi-output conformal regression methods. It can lead to future developments in this field.

**Theoretical Claims:**

The theoretical claims are well supported by the proof in appendices.

---

> ### Author Rebuttal · Authors · 2025-04-01
>
> Thank you for your positive evaluation and constructive feedback. We address your questions below.
>
> **Comparison of CP under different conditional density estimation (CDE) variants:**
>     We agree that evaluating CP methods across different CDEs is important for robustness. We performed extensive experiments (detailed in Appendix G) using a diverse set of CDE models: two normalizing flows (MQF² [33], conditional Glow [54]), Distributional Random Forests [12] (with KDE), and a GMM hypernetwork [23, 8]. Our key findings regarding the relative performance and properties of the compared conformal methods remained stable across these different base predictors, demonstrating the robustness of our conclusions.
>
> **Comparison between our normalizing flow approach and the VAE approach of STDQR**
>
> Our primary goal was a fair comparison among conformal methods. To achieve this, we standardized the base predictor architecture where appropriate. For STDQR [20], we replaced the original CVAE with a conditional normalizing flow (specifically, MQF² [33]). This ensures that comparisons between STDQR, L-CP, and other methods are not confounded by differences in the underlying generative model architecture (VAE vs. Flow). This adaptation was also motivated by the exact invertibility and direct density evaluation offered by flows, eliminating the noise associated with VAE sampling and the need for directional quantile regression in the latent space, as discussed in Appendix F.3 and recommended by Feldman et al. [20]. While a direct empirical comparison between the two models is outside the scope of this comparative study, our approach ensures a more direct comparison of the conformalization strategies themselves.
>
> **Pairing between CP methods and conditional density estimation methods**
>
> This is a valid point. Ideally, each CP method could be paired with its optimal CDE. However, for a unified comparative study, this would introduce confounding factors, making it difficult to isolate the performance differences attributable to the CP methods themselves versus the CDE pairings. Our approach was to select a set of strong, representative CDEs (as listed above) and apply them consistently across all applicable CP methods. This ensures a fair comparison of the conformal methods within our unified framework. Appendix G shows that the relative rankings and conclusions are largely consistent across these CDEs, suggesting that our findings are not overly sensitive to a specific CDE choice within this representative set.

---

### Decision · Program_Chairs · 2025-05-01

**Decision:**

Accept (poster)

**Comment:**

This paper provides a highly valuable and thorough contribution to the field of multi-output conformal prediction by systematically categorizing and unifying existing methods. Its primary strength lies not in introducing highly novel techniques but in offering a clear, well-structured framework that synthesizes prior work, making it easier for researchers to understand the landscape of this important area. The authors go beyond a simple survey by conducting detailed empirical evaluations across diverse datasets, supported by rigorous theoretical analysis. This combination of breadth and depth ensures the work serves as both a useful reference and a practical guide for future research.

The writing is clear and polished, making the paper accessible even to those less familiar with multi-output conformal prediction. The inclusion of a unified codebase further enhances the quality and impact of the work, promoting reproducibility and providing a practical tool for researchers and practitioners alike. The authors’ efforts to standardize comparisons across methods and conduct experiments under consistent conditions demonstrate a commitment to scientific rigor and fairness.

While the proposed methods, C-PCP and L-CP, are not highly novel, they are thoughtfully integrated into the broader framework, and the authors provide an honest and balanced discussion of their trade-offs relative to existing approaches, such as C-HDR. Reviewer concerns around novelty, finite-sample guarantees, and other comparisons were addressed comprehensively in the rebuttal, with the authors clarifying key points and improving the conclusion to include actionable insights and practical takeaways.

Overall, this paper exemplifies high standards of academic writing, organization, and execution. It fills an important gap in the literature by systematically analyzing and categorizing multi-output conformal prediction methods while providing a valuable resource for both new and experienced researchers in the field.